# Survival Instinct in Offline Reinforcement Learning

**Anqi Li**
University of Washington

**Dipendra Misra**
Microsoft Research

**Andrey Kolobov**
Microsoft Research

**Ching-An Cheng**
Microsoft Research

## Abstract

We present a novel observation about the behavior of offline reinforcement learning (RL) algorithms: on many benchmark datasets, offline RL can produce well-performing and safe policies even when trained with "wrong" reward labels, such as those that are zero everywhere or are negatives of the true rewards. This phenomenon cannot be easily explained by offline RL's return maximization objective. Moreover, it gives offline RL a degree of robustness that is uncharacteristic of its online RL counterparts, which are known to be sensitive to reward design. We demonstrate that this surprising robustness property is attributable to an interplay between the notion of *pessimism* in offline RL algorithms and certain implicit biases in common data collection practices. As we prove in this work, pessimism endows the agent with a *survival instinct*, i.e., an incentive to stay within the data support in the long term, while the limited and biased data coverage further constrains the set of survival policies. Formally, given a reward class — which may not even contain the true reward — we identify conditions on the training data distribution that enable offline RL to learn a near-optimal and safe policy from any reward within the class. We argue that the survival instinct should be taken into account when interpreting results from existing offline RL benchmarks and when creating future ones. Our empirical and theoretical results suggest a new paradigm for offline RL, whereby an agent is "nudged" to learn a desirable behavior with imperfect reward but purposely biased data coverage. Please visit our website https://survival-instinct.github.io for accompanied code and videos.

## 1 Introduction

In offline reinforcement learning (RL), an agent optimizes its performance given an offline dataset. Despite being its main objective, we find that return maximization is not sufficient for explaining some of its empirical behaviors. In particular, *in many existing benchmark datasets, we observe that offline RL can produce surprisingly good policies even when trained on utterly wrong reward labels.*

In Fig. 1, we present such results on the `hopper` task from D4RL [1], a popular offline RL benchmark, using a state-of-the-art offline RL algorithm ATAC [2]. The goal of an RL agent in the `hopper` task is to move forward as fast as possible while avoiding falling down. We trained ATAC agents on the original datasets and on three modified versions of each dataset, with "wrong" rewards: *1) zero*: assigning a zero reward to all transitions, *2) random*: labeling each transition with a reward sampled uniformly from $[0, 1]$, and *3) negative*: using the negation of the true reward. Although these wrong rewards contain no information about the underlying task or are even misleading, the policies learned from them in Fig. 1 (left) often perform significantly better than the behavior (data collection) policy and the behavior cloning (BC) policy. They even outperform policies trained with the true reward (denoted as *original* in Fig. 1) in some cases. This is puzzling, since RL is notorious for being sensitive to reward mis-specification [3–6]: in general, maximizing the wrong rewards with RL leads to sub-optimal performance that can be worse than simply performing BC. In addition, these wrong-reward policies demonstrate a "safe" behavior, which keeps the `hopper` from falling down for a longer period than other comparators in Fig. 1 (right). This is yet another peculiarity hard to link to

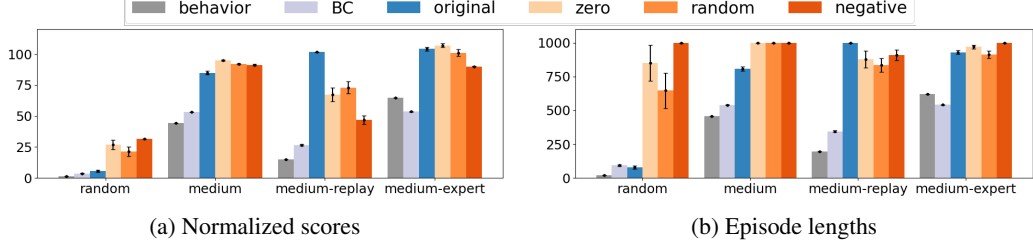

| (a) Normalized scores | (b) Episode lengths |

Figure 1: On the `hopper` task from D4RL [1], ATAC [2], an offline RL algorithm, can produce high-performance and safe policies even when trained on wrong rewards.

return maximization, as none of the wrong rewards encourage the agent to stay alive. As we will show empirically in Section 4, these effects are not unique to ATAC or the `hopper` task. They occur with multiple offline RL algorithms, including ATAC [2], PSPI [7], IQL [8], CQL [9] and the Decision Transformer (DT) [10], on dozens of datasets from D4RL [1] and Meta-World [11] benchmarks.

This robustness of offline RL is not only counter-intuitive but also cannot be explained fully by the literature. Offline RL theory [2, 7, 12, 13] provides performance guarantees only when the data reward is the true reward. Although offline imitation learning (IL) [14–16] makes no assumptions about reward, it only shows that the learned policy can achieve performance comparable to that of the behavior policy, not beyond. Robust offline RL [17–21] shows that specialized algorithms can perform well when the size of data perturbation is small. In contrast, we demonstrate that standard offline RL algorithms can produce good policies even when we completely change the reward. Constrained offline RL algorithms [22–24] can learn safe behaviors when constraint violations are both explicitly labeled and optimized. However, in the phenomena we observe, no safety signal is given to off-the-shelf, unconstrained offline RL algorithms. Recently, [25] observes a similar robustness phenomena of offline RL, but does not provide a complete explanation.[1] In Appendix B, we discuss the gap between the related work and our findings in more detail.

In this work, we provide an explanation for this seemingly surprising observation. In theory, we prove that this robustness property is attributed to the interplay between the use of pessimism in offline RL algorithms and the implicit bias in typical data collection processes. Offline RL algorithms often use pessimism to avoid taking actions that lead to unknown future events. We show that this risk-averse tendency bakes a *"survival instinct"* into the agent, an incentive to stay within the data coverage in the long term. On the other hand, the limited coverage of offline data further constrains the set of *survival policies* (policies that remain in the data support in the long term). When this set of survival policies correlates with policies that achieve high returns w.r.t. the true reward (as in the example in Fig. 1), robust behavior emerges.

Our theoretical and empirical results have two important implications. First and foremost, offline RL has a survival instinct that leads to inherent robustness and safety properties that online RL does not have. Unlike online RL, offline RL is *doubly robust*: as long as the data reward is correct or the data has a positive implicit bias, a pessimistic offline RL agent can perform well. Moreover, offline RL is safe as long as the data only contains safe states; thus safe RL in the offline setup can be achieved without specialized algorithms. Second, because of the existence of the survival instinct, the data coverage has a profound impact on offline RL. While a large data coverage improves the best policy that can be learned by offline RL with the true reward, it can also make offline RL more sensitive to imperfect rewards. In other words, collecting a large set of diverse data might not be necessary or helpful (see Section 4). This goes against the common wisdom in the RL community that data should be as exploratory as possible [26–28].

We emphasize that survival instinct's implications should be taken into account when interpreting results on existing offline RL benchmarks as well as when designing future ones. We should treat the evaluation of offline RL algorithms differently from online RL algorithms. We suggest evaluating the performance of an offline RL algorithm by training it with wrong rewards in addition to the true reward so that we can isolate the performance due to return maximization from the compounded effects of survival instinct and implicit data bias. We propose to use this performance as a score to *quantify* the data bias (see Eq. (3) in Section 4.1) in practice.

---

[1]We learned about this paper after the submission. Our theoretical results not only validate their conjecture on the expert dataset, but also predict how this robustness can happen in data collected by suboptimal policies.

We believe that our findings shed new light on RL applicability and research. To practitioners, we demonstrate that offline RL does not always require the correct reward to succeed. This opens up the possibility of using offline RL in domains where obtain high-quality rewards is challenging. Research-wise, the existence of the survival instinct raises the question of how to design data collection or data filtering procedures that would help offline RL to leverage this instinct in order to improve RL's performance with incorrect or missing reward labels. While in this paper we focus on positive data biases, we caution that in practice the data bias might be negatively correlated with a user's intention. In that case, running offline RL even with the right reward would not lead to the right behavior.

## 2 Background

The goal of offline RL is to solve an unknown Markov decision process (MDP) from offline data. Typically, an offline dataset is a collection of tuples, $\mathcal{D} \coloneqq \{(s, a, r, s')|(s, a) \sim \mu(\cdot, \cdot), r = r(s, a), s' \sim P(\cdot|s, a)\}$, where $r$ is the reward, $P$ captures the MDP's transition dynamics, and $\mu$ denotes the state-action data distribution induced by some data collection process. Modern offline RL algorithms adopt pessimism to address the issue of policy learning when $\mu$ does not have the full state-action space as its support. The basic idea is to optimize for a performance lower bound that penalizes actions leading to out-of-support future states. Such a penalty can take the form of behavior regularization [8, 29, 30], negative bonuses to discourage visiting less frequent state-action pairs [12, 31, 32], pruning less frequent actions [13, 33], adversarial training [2, 7, 14, 34, 35] or value penalties in modified dynamic programming [9, 36].

We are interested in settings where the offline RL agent learns not from $\mathcal{D}$ itself but from its corrupted version $\tilde{\mathcal{D}}$ with a wrong reward $\tilde{r}$, i.e., $\tilde{\mathcal{D}} \coloneqq \{(s, a, \tilde{r}, s')|(s, a, s') \in \mathcal{D}, \tilde{r} = \tilde{r}(s, a) \in [-1, 1]\}$. We assume $\tilde{r}$ is from a reward class $\tilde{\mathcal{R}}$ — which may not necessarily contain the true reward $r$ — and we wish to explain why offline RL can learn good behaviors from the corrupted dataset $\tilde{\mathcal{D}}$ (e.g., as in Fig. 1). To this end, we introduce notations and assumptions we will use in the paper.

**Notation**   We focus on the setting of infinite-horizon discounted MDPs. We denote the task MDP that the agent aims to solve as $\mathcal{M} = (\mathcal{S}, \mathcal{A}, r, P, \gamma)$, where $\mathcal{S}$ is the state space, $\mathcal{A}$ is the action space, and $\gamma \in [0, 1)$ is the discount. Without loss of generality, we assume $r : \mathcal{S} \times \mathcal{A} \to [0, 1]$. Let $\Delta(\mathcal{U})$ denote the set of probability distributions over a set $\mathcal{U}$. We denote a policy as $\pi : \mathcal{S} \to \Delta(\mathcal{A})$. For a reward function $r$, we define a policy $\pi$'s state value function as $V_r^\pi(s) \coloneqq \mathbb{E}_{\pi, P}[\sum_{t=0}^\infty \gamma^t r(s_t, a_t)|s_0 = s]$ and the state-action value function as $Q_r^\pi(s, a) \coloneqq r(s, a) + \gamma \mathbb{E}_{s' \sim P(\cdot|s, a)}[V_r^\pi(s')]$. Solving MDP $\mathcal{M}$ requires learning a policy that maximizes the return at an initial state distribution $d_0 \in \Delta(\mathcal{S})$, that is, $\max_\pi \mathbb{E}_{s \sim d_0}[V_r^\pi(s)]$. We denote the optimal policy as $\pi^*$ and the optimal value functions as $V_r^*$ and $Q_r^*$. Given $d_0$, we define the average state-action visitation of a policy $\pi$ as $d^\pi(s, a) \coloneqq (1 - \gamma)\mathbb{E}_{\pi, P}[\sum_{t=0}^\infty \gamma^t d_t^\pi(s, a)]$, where $d_t^\pi(s, a)$ denotes the probability of visiting $(s, a)$ at time $t$ when running $\pi$ starting at an initial state sampled from $d_0$. Note that $(1 - \gamma)\mathbb{E}_{s \sim d_0}[V_r^\pi(s)] = \mathbb{E}_{(s,a) \sim d^\pi}[r(s, a)]$. For a function $f : \mathcal{S} \times \mathcal{A} \to \mathbb{R}$, we use the shorthand $f(s, \pi) = \mathbb{E}_{a \sim \pi|s}[f(s, a)]$; similarly for $f : \mathcal{S} \to \mathbb{R}$, we write $f(p) = \mathbb{E}_{s \sim p}[f(s)]$, e.g., $V_r^\pi(d_0) = \mathbb{E}_{s \sim d_0}[V_r^\pi(s)]$. For a distribution $p$, we use supp$(p)$ to denote its support.

**Assumption**   We make the typical assumption in offline RL that the data distribution $\mu$ assigns positive probabilities to these state-actions visited by running the optimal policy $\pi^*$ starting from $d_0$.

**Assumption 1** (Single-policy Concentrability). We assume $\sup_{s \in \mathcal{S}, a \in \mathcal{A}} \frac{d^{\pi^*}(s, a)}{\mu(s, a)} < \infty$.

This is a standard assumption in the offline RL literature, which in the worst case is a necessary condition to have no regret w.r.t. $\pi^*$ [37]. There are generalized notions [7, 34] of this kind, which are weaker but requires other smoothness assumptions. We note that Assumption 1 does not assume that $\mu$ is the average state-action visitation frequency of a single behavior policy, nor does it assume $\mu$ has a full coverage of all states and actions or all policy distributions [27, 38].

## 3 Why Offline RL can Learn Right Behaviors from Wrong Rewards

In this section, we provide conditions under which offline RL's aforementioned robustness w.r.t. misspecified rewards emerges. Our main finding is summarized in the theorem below.

**Theorem 1.** *(Informal) Under Assumption 1 and certain regularity assumptions, if an offline RL algorithm Algo is set to be sufficiently pessimistic and the data distribution $\mu$ has a positive bias, for any data reward $\tilde{r} \in \tilde{\mathcal{R}}$, the policy $\hat{\pi}$ learned by Algo from the dataset $\tilde{\mathcal{D}}$*

*has performance guarantee* $V_r^{\pi^*}(d_0) - V_r^{\hat{\pi}}(d_0) \leq O(\iota)$ *as well as safety guarantee* $(1 - \gamma)\sum_{t=0}^{\infty} \gamma^t \mathrm{Prob}\left(\exists \tau \in [0,t], s_\tau \notin \mathrm{supp}(\mu)|\hat{\pi}\right) \leq O(\iota)$ *for a small* $\iota$ *that decreases to zero as the degree of pessimism and dataset size increase.*

In other words, the robustness originates from an *interplay* between pessimism and an implicit positive bias in data. Here is the insight on why Theorem 1 is true. Because of pessimism, offline RL endows agents with a "survival instinct" — it implicitly solves a constrained MDP (CMDP) problem [39] that enforces the policy to stay within the data support. When combined with a training data distribution that has a positive bias (e.g., all policies staying within data support are near-optimal) such a survival instinct results in robustness to reward misspecification.

Overall, Theorem 1 has two important implications:

1. Offline RL is *doubly robust*: it can learn near optimal policies so long as either the reward label is correct or the data distribution is positively biased;
2. Offline RL is *intrinsically safe* when data are safe, regardless of reward labeling, without the need of explicitly modeling safety constraints.

In the remaining of this section, we provide details and discussion of the statements above. The complete theoretical statements and proofs can be found in Appendix D.

### 3.1 Intuitions for Survival Instinct and Positive Data Bias

We first use a grid world example to build some intuitions. Fig. 2 shows a goal-directed problem, where the true reward is +1 and -1 upon touching the key and the lava, respectively. The offline data is suboptimal and does not have full support. All data trajectories that touched the lava were stopped early, while others were allowed to continue until the end of the episode. The goal state (key) is an absorbing state, where the agent can stay *forever* beyond the episode length. We use the wrong rewards in Fig. 1 to train PEVI [12], a finite-horizon, tabular offline RL method. PEVI performs dynamic programming similar to value iteration, but with a pessimistic value initialization and an instantaneous pessimism-inducing penalty of $O(-1/\sqrt{n(s,a)})$, where $n(s,a)$ is the empirical count in data. The penalties ensure that the learned value lower bounds the true one.

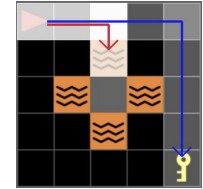

Figure 2: A grid world. BC (red); offline RL with wrong rewards (blue). The opacity indicates the frequency of a state in the data (more opaque means more frequent). Offline RL with the three wrong rewards produces the same policy.

We see the PEVI agent learned with any wrong reward in Fig. 1 is able to solve the problem despite data imperfection, while the BC agent that mimics the data directly fails. The main reasons are: *1)* There is a *data bias* whereby longer trajectories end closer to the goal (especially, the longest trajectories are the ones that reach the goal, since the goal is an absorbing state). We call this a *length bias* (see Section 3.3). This positive data bias is due to bad trajectories (touching the lava or not reaching the goal) being cut short or timing out. *2)* Pessimism in PEVI gives the agent an *algorithmic bias* (i.e., survival instinct) that favors longer data trajectories. Because of the pessimistic value initialization, PEVI treats trajectories shorter than the episode length as having the lowest return. As a result, by maximizing the pessimistically estimated values, PEVI learns good behaviors despite wrong rewards by leveraging the survival instinct and positive data bias *together*. We now make this claim more general.

### 3.2 Survival Instinct

Survival instinct is a pessimism induced behavior that offline RL algorithms tend to favor policies leading to *in-support trajectories*. We formally characterize this risk aversion behavior by the concept of constrained MDP (CMDP) [39], which we define below.

**Definition 1.** *Let* $f, g : \mathcal{S} \times \mathcal{A} \to [-1,1]$. *A CMDP* $\mathcal{C}(\mathcal{S}, \mathcal{A}, f, g, P, \gamma)$ *is a constrained optimization problem:* $\max_\pi V_f^\pi(d_0)$ *s.t.* $V_g^\pi(d_0) \leq 0$. *Let* $\pi^\dagger$ *denote its optimal policy. For* $\delta \geq 0$, *we define the set of* $\delta$-*approximately optimal policies* $\Pi_{f,g}^\dagger(\delta) := \{\pi : V_f^{\pi^\dagger}(d_0) - V_f^\pi(d_0) \leq \delta, V_g^\pi(d_0) \leq \delta\}$.

We prove that when trained on $\tilde{D}$, offline RL, because of its pessimism, implicitly solves the CMDP below even when the algorithm does not explicitly model any constraints:

$$\mathcal{C}_\mu(\tilde{r}) := \mathcal{C}(\mathcal{S}, \mathcal{A}, \tilde{r}, c_\mu, P, \gamma), \tag{1}$$

where $c_\mu(s,a) := \mathbb{1}[\mu(s,a) = 0]$ indicates whether $(s,a)$ is out of the support of $\mu^2$ i.e., the

constraint in Eq. (1) enforces the agent's trajectories to stay within the data support. Note the constraint in this CMDP is feasible because of Assumption 1.

**Proposition 1** (Survival Instinct). *(Informal) Under certain regularity conditions on $\mathcal{C}_\mu(\tilde{r})$, the policy learned by an offline RL algorithm Algo with the offline dataset $\tilde{D}$ is $\iota$-approximately optimal with respect to $\mathcal{C}_\mu(\tilde{r})$, for some small $\iota$ which decreases as the algorithm becomes more pessimistic.*

Proposition 1 says if an offline RL algorithm is sufficiently pessimistic, then the learned policy is approximately optimal to $\mathcal{C}_\mu(\tilde{r})$. The learned policy has not only small regret with respect to the data reward $\tilde{r}$, but also small chances of escaping the support of the data distribution $\mu$.

We highlight that the survival instinct in Proposition 1 is a *long-term* behavior. Such a survival behavior cannot be achieved by myopically taking actions in the data support in general (such as BC).[3] For instance, when some trajectories generating the data are truncated (e.g., due to early-stopping or intervention for safety reasons), taking in-support actions may still lead to out-of-support states in the future, as the BC agent in Section 3.1.

**Proof Sketch**    The key to prove Proposition 1 is to show that an offline RL algorithm by pessimism has small regret for not just $\tilde{r}$ but a set of reward functions consistent with $\tilde{r}$ on data but different outside of data, including the Lagrange reward of (1) (i.e., $\tilde{r} - \lambda c_\mu$, with a large enough $\lambda \geq 0$). We call this property admissibility and we prove in Appendix F that many existing offline RL algorithms are admissible, including model-free algorithms ATAC [2], VI-LCB [31] PPI/PQI [13], PSPI [7], as well as model-algorithms, ARMOR [34, 40], MOPO [32], MOReL [33], and CPPO [14]. By Proposition 1 these algorithms have survival instinct. Please see Appendix D for details.

### 3.3   Positive Data Bias

In addition to survival instinct, another key factor is an implicit positive bias in common offline datasets. Typically these data manifest meaningful behaviors. For example, in collecting data for goal-oriented problems, data recording is normally stopped if the agent fails to reach the goal within certain time limit. Another example is problems (like robotics or healthcare) where wrong decisions can have detrimental consequences, i.e., problems that safe RL studies. In these domains, the data are collected by qualified policies only or under an intervention mechanism to prevent catastrophic failures. Such a one-sided bias can creates an effect that staying within data support would lead to meaningful behaviors. Below we formally define the positive data bias; Later in Section 4 (Fig. 5), we provide empirical estimates of the degree of positive data bias.

**Definition 2** (Positive Data Bias). A distribution $\mu$ is $\frac{1}{\epsilon}$-*positively biased* w.r.t. a reward class $\tilde{\mathcal{R}}$ if

$$\max_{\tilde{r} \in \tilde{\mathcal{R}}} \max_{\pi \in \Pi^\dagger_{\tilde{r}, c_\mu}(\delta)} V_r^{\pi^*}(d_0) - V_r^\pi(d_0) \leq \epsilon + O(\delta) \tag{2}$$

for all $\delta \geq 0$, where $\Pi^\dagger_{\tilde{r}, c_\mu}(\delta)$ denotes the set of $\delta$-approximately optimal policy of $\mathcal{C}_\mu(\tilde{r})$.

We measure the degree of positive data bias based on how bad a policy can perform in terms of the true reward $r$ when approximately solving the CMDP $\mathcal{C}_\mu(\tilde{r})$ in (1) defined with the wrong reward $\tilde{r} \in \tilde{\mathcal{R}}$. If the data distribution $\mu$ is $\infty$-positively biased, then any approximately optimal policy to $\mathcal{C}_\mu(\tilde{r})$ (which includes the policies learned by offline RL, as shown earlier) can achieve high return in the true reward $r$. We also can view the degree of positiveness as reflecting whether $\tilde{r}$ provides a similar ranking as $r$, *among policies within the support of $\mu$*. When there is a positive bias, offline RL can learn with $\tilde{r}$ to perform well under $r$, even when $\tilde{r}$ is not aligned with $r$ globally. This is in contrast to online RL, which requires global alignment due to the exploratory nature of online RL.

**Examples**    We provide a few examples of positive data bias. (See Appendix D.4 for proofs.)

- The distribution $d^\pi$ induced by any policy $\pi$ is $\infty$-positively biased for any rewards resulting from potential-based reward shaping [41] since it provides the same ranking for all policies.
- For an IL setup, the data distribution $\mu$ is $\infty$-positively biased for any rewards $\tilde{r} : \mathcal{S} \times \mathcal{A} \to \mathbb{R}$ if the data is generated by the optimal policy (i.e., $\mu = d^{\pi^*}$). This is because following the optimal

---

[2]The use of an indicator function is not crucial here. We can extend the current analysis here to other costs that are zero in the support and strictly positive out of the support.

[3]BC requires a stronger condition, e.g., the data are all complete trajectories without intervention or timeouts.

policy is the only way to stay within the data support. In Section 4, we empirically show that offline RL algorithms can learn good policies with wrong rewards on D4RL `expert` datasets, which are collected by near-optimal policies.

- A positive bias happens when longer trajectories in the data have smaller optimality gap. This is a generalized formal definition of the *length bias* mentioned in Section 3.1. This condition is typically satisfied when intervention is taken in the data collection process (despite the data collection policy being suboptimal), as in the motivating example in Fig. 1. Later in Section 4, we will investigate deeper into this kind of length bias empirically.

**Remark** We highlight that the positive data bias assumption in Definition 2 is *different* from assuming that the data is collected by expert policies, which is typical in the IL literature. Positive data bias assumption can hold in cases when data is generated by highly suboptimal policies, which we observe in the `hopper` example from Section 1. On the other hand, there are also cases where IL can learn well, while positive data bias does not exist (e.g., when learning from data collected by a stochastic expert policy that covers highly suboptimal actions with small probabilities).

### 3.4 Summary: Offline RL is Doubly Robust and Intrinsically Safe

We have discussed the survival instinct from pessimism and the potential positive data bias in common datasets. In short, survival instinct enables offline RL algorithms to learn policies that benefit from a favorable inductive bias of staying within the support of a positive data distribution. As a result, offline RL becomes robust to reward mis-specification (namely, Theorem 1).

Below we discuss some direct implications of Theorem 1. First, we can view this phenomenon as a doubly robust property of offline RL. We borrow the name "doubly robust" from the offline policy *evaluation* literature [42] to highlight the robustness of offline RL to reward mis-specification as an offline policy *optimization* approach.

**Corollary 1.** *Under Assumption 1, offline RL can learn a near optimal policy, as long as the reward is correct,* or *the data has a positive implicit bias.*

Theorem 1 also implies that offline RL is an intrinsically safe learning algorithm, unlike its counterpart online RL where additional penalties or constraints need to be explicitly modelled [43–47].

**Corollary 2.** *If $\mu$ only covers safe states and there exists a safe policy staying within the support of $\mu$, then the policy of offline RL only visits safe states with high probability (see Theorem 1).*

We should note that covering safe states is a very mild assumption in the safe RL literature [45, 48], e.g., compared with all (data) states have a safe action. It does not require the data collection policies that generate $\mu$ are safe. This condition can be easily satisfied by filtering out unsafe states in post processing. The existence of a safe policy is also mild and common assumption. In Section 4.2, we validate this inherent safety property on offline SafetyGymnasium [49], an offline safe RL benchmark.

## 4 Experiments

We conduct two group of experiments. In Section 4.1, we conduct large scale experiment, showing that multiple offline RL algorithms can be robust to reward mis-specification on a variety of datasets. In Section 4.2, we experimentally validate the inherent safety of offline RL algorithms stated in Corollary 2. We show that admissible offline RL algorithms, without modifications, can achieve state-of-the-art performance on an offline safe RL benchmark [49].

### 4.1 Robustness to Mis-specified Rewards

We empirically study the performance of offline RL algorithms under wrong rewards in a variety of tasks. We use the same set of wrong rewards as in Fig. 1. *1)* zero: the zero reward, *2)* random: labeling each transition with a reward value randomly sampled from Unif$[0, 1]$, and *3)* negative: the negation of true reward. We consider five offline RL algorithms, ATAC [2], PSPI [7], IQL [8], CQL [9] and decision transformer (DT)[4] [10]. We deliberately choose offline RL algorithms to cover those that are provably pessimistic [2, 7] and those that are popular among practitioners [8, 9], as well as an unconventional offline RL algorithm [10]. We consider a variety of tasks from D4RL [1] and Meta-World [11] ranging from safety-critical tasks (i.e., the agent dies when reaching bad states),

---

[4]We condition the decision transformer on the return of a trajectory sampled randomly from trajectories that achieve top 10% of returns in terms of data reward.

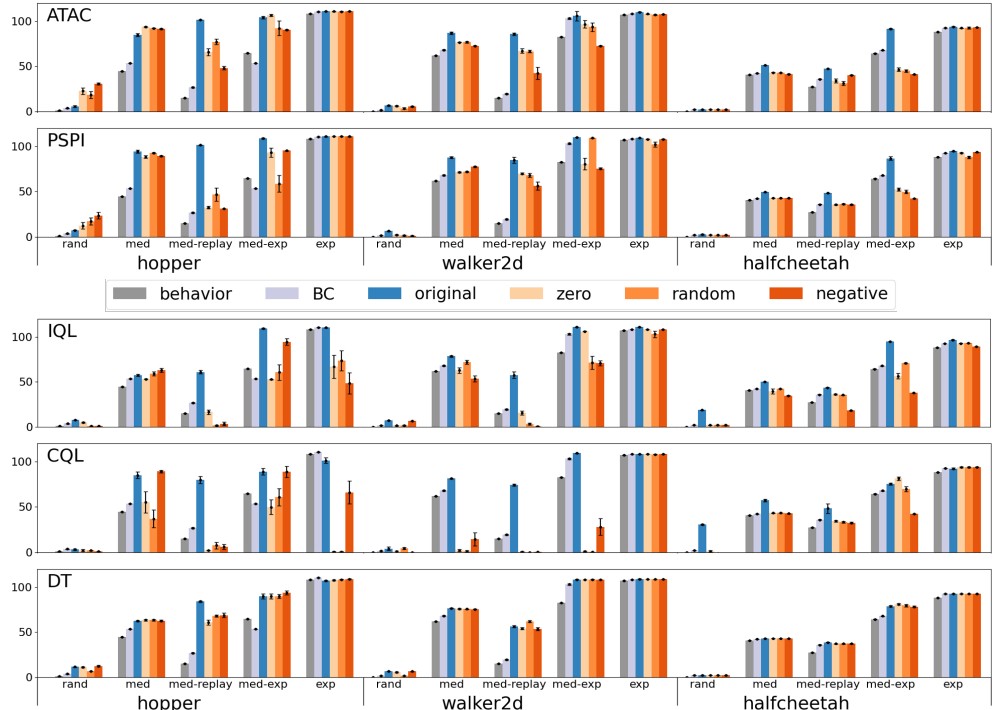

Figure 3: Normalized scores for locomotion tasks from D4RL [1]. The mean and standard error for normalized scores are computed across 10 random seeds. For each random seed, we evaluate the final policy of each algorithm over 50 episodes.

goal-oriented tasks, and tasks that belong to neither. We train a total of around 16k offline RL agents (see Appendix C.8). Please see Appendix C for details.

**Messages**    We would like to convey three main messages through our experiments. First, implicit data bias *can* exist naturally in a wide range of datasets, and offline RL algorithms that are sufficiently pessimistic can leverage such a bias to succeed when given wrong rewards. Second, offline RL algorithms, regardless of how pessimistic they are, become sensitive to reward when the data does not possess a positive bias. Third, offline RL algorithms without explicit pessimism, e.g., IQL [8], can sometimes still be pessimistic enough to achieve good performance under wrong rewards.

**Remark on negative results**    We consider "negative" results, i.e., when offline RL *fails* under wrong rewards, as important as the positive ones. Since they tell us *how* positive data bias can be broken or avoided. We hope our study can provide insights to researchers who are interested in actively incorporating positive bias in data collection, as well as who hope to design offline RL benchmarks specifically with or without positive data bias.

### 4.1.1   Locomotion Tasks from D4RL

We evaluate offline RL algorithms on three locomotion tasks, `hopper`, `walker2d`[5] and `halfcheetah`, from D4RL [1], a widely used offline RL benchmark. For each task, we consider five datasets with different qualites: `random`, `medium`, `medium-replay`, `medium-expert`, and `expert`. We refer readers to [1] for the construction of these datasets. We measure policy performance in D4RL normalized scores [1]. We provide three baselines 1) behavior: normalized score directly computed from the dataset, 2) BC: the behavior cloning policy, and 3) original: the policy produced by the offline RL algorithm using the original dataset (with true reward). The normalized scores for baselines and offline RL algorithms with wrong rewards are presented in Fig. 3. The exact numbers can be found in tables in Appendix C.

**Positive bias exists in some D4RL datasets.**    We visualize the length bias of D4RL datasets in Fig. 4. Here are our main observations. *1)* Most datasets for `hopper` and `walker2d`, with the

---

[5]We remove terminal transitions from `hopper` and `walker2d` datasets when learning with wrong rewards. In Appendix C, we include a ablation study on the effect of removing terminals when using true reward.

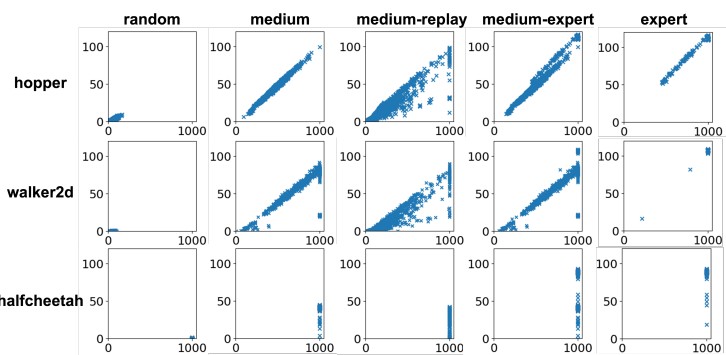

Figure 4: A visualization of length bias in datasets from D4RL [1]. Each plot corresponds to a dataset for a task (row) with a dataset (column). Each trajectory in a dataset is represented as a data point with the $x$-coordinate being its episode length and $y$-coordinate being its normalized score.

exception of the medium-replay datasets, have a strong length bias, where longer trajectories have higher returns. This length bias is due to the safety-critical nature of `hopper` and `walker2d` tasks, as the trajectories get terminated when reaching bad states. The length bias is especially salient in `hopper-medium`, where the normalized score is almost proportional to episode length. *2)* The medium-replay datasets for `hopper` and `walker2d` have more diverse behavior, so the bias is smaller. *3)* All `halfcheetah` datasets do not have length bias, as they all have the same length of 1000. *4)* `hopper-expert` dataset has a length bias, while `walker2d` and `halfcheetah-expert` datasets do not have an obvious length bias. However, it is worth noting that all `expert` datasets have an IL-type positive bias as discussed in Section 3.3.

**Offline RL can learn good policies with wrong rewards on datasets with strong length bias.** On datasets with strong length bias (`hopper-random`, `hopper-medium` and `walker2d-medium`), we observe that ATAC and PSPI with wrong rewards generally produce well-performing policies, in a few cases even out-performing the policies learned from the true reward (original in Fig. 4). DT is mostly insensitive to reward quality. IQL and CQL with wrong rewards can sometimes achieve good performance; among the two, we find IQL to be more robust to wrong rewards and CQL with wrong rewards almost fails completely on all `walker2d` datasets.

**Offline RL can learn good policies with wrong rewards on `expert` datasets**  In Section 3, we point out that the data has a positive bias when it is generated by the optimal policy. In Fig. 3, we observe that all offline RL algorithms, when trained with wrong rewards, can achieve expert-level performance on `walker2d` and `halfcheetah-expert` datasets. ATAC, PSPI and DT perform well on the `hopper-expert` dataset while IQL and CQL receive lower scores when using wrong rewards.

**Offline RL needs stronger reward signals on datasets with diverse behavior policy.**  The medium-replay datasets of `hopper` and `walker2d` are generated from multiple behavior policies. Due to their diverse nature, they are a multiple ways to stay within data support in a long term. As a result, the survival instinct of offline RL by itself is not sufficient to guarantee good performance. Here algorithms with wrong rewards generally under-perform the policies trained with true reward. As datasets have a more diverse coverage, they can be less positively biased and offline RL requires a stronger reward signal to differentiate good survival policies and the bad ones. Practically speaking, when high-quality reward is not available, a diverse dataset can even hurt offline RL performance. This is contrary to the common belief that larger data support is more preferable [26–28].

**Offline RL requires good reward to perform well on datasets without length bias.**  In all four `halfcheetah` datasets, the trajectories all have the same length, as is demonstrated in Fig. 4. This means that there is no data bias. We observe that all algorithms with wrong rewards at best perform similarly as the behavior and BC policies in most cases; this is an imitation-like effect due to the survival instinct. In the `halfcheetah-medium-expert` dataset, since there are a variety of surviving trajectories, with score from 0 to near 100, we observe that the performance of the resulting policy degrades as data reward becomes more different from the true reward.

**Offline RL algorithms with provable pessimism produce safer policies.**  For `hopper` and `walker2d`, we observe that policies learned by ATAC and PSPI, regardless of the reward, can keep the agent from falling for longer. The episode lengths of IQL and CQL policies are often

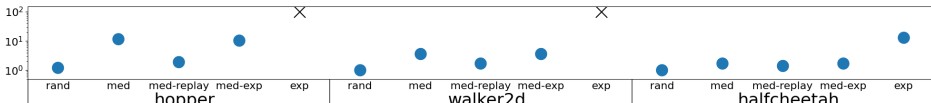

Figure 5: Estimated positive data bias of all 15 D4RL datasets given by ATAC. Datasets marked by "×", i.e., `hopper-expert` and `walker2d-expert`, have infinite positive bias.

comparable to that of the behavior policies. We provide statistics of episode lengths in Appendix C. In Section 4.2, we provide further results validating the safety properties of ATAC and PSPI.

**On empirically estimating positive data bias** Inspired by Definition 2, we propose an empirical estimate of positive data bias. Let $J_{\text{zero}}$, $J_{\text{rand}}$ and $J_{\text{neg}}$ be the return (with respect to the true reward) of an offline RL algorithm learned with zero, random, and negative rewards, respectively. Then, given an estimated optimal return $\hat{J}^{\star}$, we propose to empirically estimate the positive data bias by

$$\text{estimated positive bias} = \hat{J}^{\star}/\max\{\hat{J}^{\star}-\min\{J_{\text{zero}},J_{\text{rand}},J_{\text{neg}}\},0\}. \tag{3}$$

In Fig. 5, we visualize the estimated positive data bias of all 15 D4RL datasets given by ATAC. Estimated positive data bias of all D4RL and Meta-World datasets is given in Fig. 9.

**Remark on benchmarking offline RL** Our observations on the popular D4RL datasets raise an interesting question for benchmarking offline RL algorithms. An implicit positive data bias can give certain algorithms a hidden advantage, as they can already achieve good performance without using the reward. Without controlling data bias, it is hard to differentiate whether the performance of those algorithms are due to their ability to maximize returns or simply due to their survival instinct.

### 4.1.2 Goal-oriented Manipulation Tasks from Meta-World

We study goal-oriented manipulation tasks from the Meta-World benchmark [11]. We consider 15 goal-oriented tasks from Meta-World: the 10 tasks from the MT-10 set and the 5 testing tasks from the ML-45 set. For each task, we generate a dataset of 110 trajectories using the scripted policies provided by [11]; 10 are produced by the scripted policy, and 100 are generated by perturbing the scripted policy with a zero-mean Gaussian noise of standard deviation 0.5. In the datasets, unsuccessful trajectories receive a time-out signal after 500 steps (maximum episode length for Meta-World).

For each task, we measure the success rate of policies for 50 *new* goals unseen during training. Similar to D4RL experiments, we consider BC policies and policies trained with the true reward (original) as baselines. Since the training dataset is generated for a different set of goals, we do not compare the success rate of learned policies with the success rate in data.

**Goal-oriented problems have data bias.** Goal-oriented problems are fundamentally different from safety-critical tasks (such as `hopper` and `walker2d`). A good goal-oriented policy should reach the goal as fast as possible rather than wandering around until the end of an episode. But another form of length bias still exists here, as successful trajectories are labeled with a termination signal that indicates an unbounded length and failed trajectories have a bounded length (see Section 3.1).

**Offline RL can learn with wrong rewards on goal-oriented tasks.** The success rate of learned policies with different rewards are shown in Fig. 6. We observe that ATAC and PSPI with wrong rewards generally achieve comparable or better success rate than BC policies. The exceptions are often due to that ATAC or PSPI, even with the true reward, does not work as well as BC in those tasks, e.g. drawer-open, bin-picking and box-close. This could be caused by overfitting or optimization errors (due to challenges of dynamics programming over long horizon problems). In a number of tasks, such as peg-insert-side, push and reach, ATAC and PSPI with wrong rewards can out-perform BC by a margin. This shows that data collection for goal-oriented problems have a positive bias that offline RL algorithms can succeed without true reward. This is remarkable as when learning with random and negative rewards, unsuccessful trajectories are long and generally have significantly higher return, as reward for each step is non-negative. The offline RL algorithms need to be sufficiently pessimistic and be able to plan over a long horizon to propagate pessimism to the unsuccessful data trajectories. For IQL, there is a gentle performance decrease as the reward becomes more different than the true reward, even though policies learned with wrong rewards are not much worse than BC in many cases. CQL shows robustness to reward in a few tasks such as button-press, drawer-close and door-unlock. Interestingly, DT performs almost uniformly across all rewards, potentially because DT does not explicitly maximize returns.

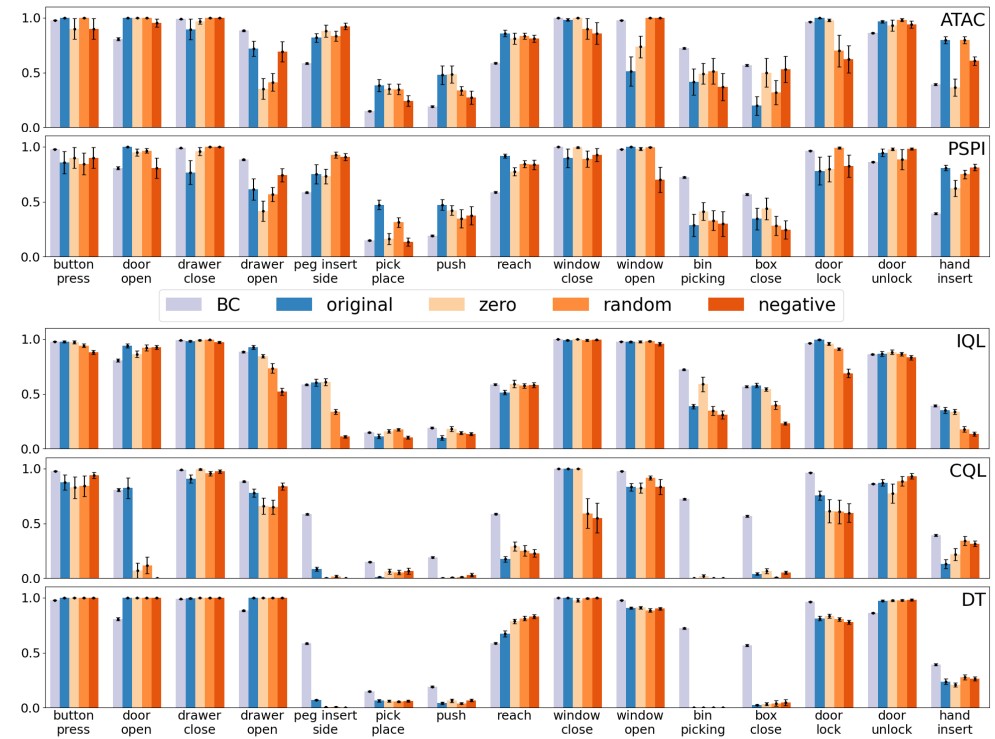

Figure 6: Success rate for goal-oriented tasks from Meta-World [11]. The average success rate and confidence interval are computed across 10 random seeds. For each seed, we evaluate final policies for 50 episodes, each with a new goal unseen in the dataset.

**Remark on BC performance** It is noticeable that BC policies in general achieve high performance, in many tasks often out-performing offline RL policies learned with true reward. This effect has also been observed in existing work on similar tasks [50]. We would like to clarify that the goal of our experiments is to study the behavior of offline RL algorithms when trained with wrong rewards, rather than showing that offline RL performs better than BC. We refer interested readers to existing studies [50, 51] on when using offline RL algorithms is or is not preferable over BC.

## 4.2 Inherent Safety Properties of Offline RL Algorithms

We show offline RL algorithms (without any modifications) can behave as a safe RL algorithm. We conduct experiments on offline SafetyGymnasium [49] using ATAC [2] and PSPI [7]. We first remove all transitions with non-zero cost and then run offline RL algorithms on the filtered datasets.

The results are summarized in Table 6 in Appendix C.6. We observe that *offline RL with this naïve data filtering strategy (using less data) can achieve comparable performance as the per-task best performing state-of-the-art offline safe RL algorithm*, which uses the full dataset and has knowledge of the cost target. Our agents in general incur low cost except for the circle tasks. We hypothesize that learning a safe policy from the circle datasets is hard, as other baselines also struggle to produce safe policies. Note that these are preliminary results, and better performance might be achievable through using a more sophisticated data filtering strategy. Please see Appendix C.6 for experiment details.

## 5 Concluding Remarks

We present unexpected results on the robustness of offline RL to reward mis-specification. We show that this property originates from the interaction between the *survival instinct* of offline RL and hidden positive biases in common data collection processes. Our findings suggest extra considerations should be taken when interpreting offline RL results and designing future benchmarks. In addition, our findings open a new space for offline RL research: the possibility of designing algorithms that proactively leverage the survival instinct to learn policies for domains where rewards are nontrivial to specify or unavailable. Please see Appendix A for a discussion on limitations and broader impacts.

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

# Contents

## A   Limitations and Broader Impacts

**Limitations**   We would like to point out that our results do not suggest that offline RL can always learn with wrong rewards (since the data may not have the needed positive bais). This is shown in our experimental results. In addition, our theoretical analysis makes certain simplifications (e.g., ignoring the optimization difficulty of finding a survival policy), which may not hold in practice. We experimented with a limited number of offline RL algorithms and benchmarks. It would be interesting to see if similar experimental results can be obtained by, e.g., model-based offline RL algorithms and on other types of dataset such as the ones which use image observations. Future work may also explore approaches to quantifying positive data bias without having to run offline RL algorithms.

**Broader Impacts**   Our discovery might shed light on a potential positive impact of offline RL. By survival instinct, offline RL may be able to train sequential-decision policies that have good behaviors without needing to collect negative data. This is different from the common belief that RL can learn to behave well only if it has seen both positive (success) and negative (failure) data. This ability to learn from one-sided data is important especially for applications where collecting negative data is unethical, e.g., learning not to produce hateful speech by first collecting hateful speech, learning not to harm patients in medical applications by first harming some, etc.

On the flip side, also by survival instinct, offline RL can be prone to existing societal (e.g., gender, racial) biases in data, and, moreover, such a bias cannot be easily corrected by relabeling data with a different reward. As a result, when using an offline RL algorithm, more strategic thinking on data collection might be needed. We encourage researchers and practitioners to collect datasets using methodology such as those proposed in [52] and provide details such as how data was collected and cleaned so that users can assess whether it is appropriate to train offline RL with these datasets.

# B  Related Work

**Offline RL**    Offline RL studies the problem of return maximization given an offline dataset. Offline RL algorithms can be broadly classified into model-based approaches, e.g., [32, 33, 14, 35, 34], and model-free approaches, e.g., [12, 29, 7, 2, 9, 8]. Since the agent needs to learn without online data collection, offline RL becomes more challenging when the offline dataset does not provide good coverage over the state-action distribution of all feasible policies. There have been two common strategies to handle insufficient coverage, behavior regularization approaches, e.g., [29, 30, 8], which restricts the learned policy to be close to the behavior policy, and value regularization approaches, e.g., [12, 13, 7, 2, 9, 34] which provide a pessimistic value estimates for the learned policy. Both approaches can be viewed as optimizing for a performance lower bound of the agent's return.

Offline RL commonly assumes compatibility between offline dataset and MDP, i.e., the offline dataset should be generated from the task MDP of interest. Our work is distinctive in the offline RL literature as we study the behavior of existing offline RL algorithms when data reward *differs* from the true MDP reward. Existing tools for analyzing offline RL are not directly applicable to our setting: policies with good performance under data reward, as guaranteed by such tools, do not necessarily attain high return under the true reward. Our novel analysis is made possible by studying the properties of offline data distribution, an aspect mostly neglected by existing work in the literature.

**Offline RL with imperfect reward**    In the offline RL literature, there is a line of work studying scenarios where the reward is missing in a subset of the dataset [53–56] or when the data reward is misspecified [57]. These approaches propose to construct a new reward function and apply offline RL algorithms on the relabeled dataset. For example, [53, 56] label the missing reward with a constant minimum reward. [55, 54] learn a pessimistic reward function. [57] optimize for a reward function which minimizes the reward difference among expert data.

Different from this line of work, we are interested in running offline RL algorithms using the misspecified reward *as it is*. We show both theoretically and empirically that offline RL algorithms can produce well-performing policies with rewards different from the true reward as long as the underlying data distribution satisfies certain conditions. It can be an interesting future work to combine these techniques with our analysis to explore the robustness of offline RL algorithms under the class of learned reward functions.

After paper submission, we learned that [25] observed a similar robustness phenomena of offline RL. However, the authors did not provide a complete explanation. They found that offline RL algorithms such as AWR [58] empirically can learn well with random or zero rewards on D4RL locomotion datasets. For the expert datasets, they conjectured that this robustness may be due to the fact that offline RL algorithms behave like imitation learning on expert data, but they did not provide details of why offline RL algorithms have this robustness behavior on other datasets. Here we provide a formal theoretical proof to show that this robustness is due to the interaction between the survival instinct of offline RL and an implicit positive data bias. Our results support [25]'s conjecture on the expert dataset; we show that expert datasets generated by an optimal policy can theoretically be proved to be infinitely positively biased, and we also empirically estimate the positive bias on the D4RL datasets in Fig. 5, which agrees with the theoretical prediction. Our results further provide conditions for positive data bias beyond the expert data scenario, such as the length bias in Section 3. These new insights show why offline RL algorithms can learn good policies with wrong rewards, on datasets collected by non-expert suboptimal policies. Finally, we present strong empirical evidence to show this robustness phenomenon happens in a large number of existing benchmarks with multiple offline RL algorithms.

**Offline IL**    Offline imitation learning (IL) can be considered as a special case of offline RL when reward is missing from the entire dataset, or when the agent has the largest uncertainty about the reward [2]. In offline IL, the learner is instead given the information on whether a transition is generated by an expert. The goal of offline IL is to produce a policy that has comparable performance with the expert. [59] learns a discriminator reward to classify expert and non-expert data, and apply offline RL algorithms with the learned reward. [60] uses the discriminator as the weight for weighted behavior cloning. [61, 16, 62] proposes DICE [63]-style algorithms to minimize the divergence between the state-action or state-(next state) occupancy measure of the learner and the expert. [15, 64] minimize state-action or state-(next state) distribution divergence under a pessimistic model. [65]

extends maximum entropy inverse RL [66] to an offline setting. [67] proposes to run offline RL under an indicator reward function on whether the data is expert data. [55] learns a pessimistic reward function where the expert data labeled as the maximum reward.

Our experiments with zero reward function is similar to an offline IL setting with the behavior policy being the expert. However, we show that offline RL algorithms can sometimes achieve significant performance gain over behavior and behavior cloning policies, which is a phenomenon that cannot be explained by existing analysis in offline IL. We provide an explanation to this puzzle: in Appendix D.4, we show that the data distribution is $\infty$-positively biased (according our definition) for offline RL, when the data is generated by the optimal policy of the true MDP. This means that an admissible offline RL algorithm can achieve near-optimal performance with *any* reward function from its corresponding admissible reward class in this case. We empirically show that in Section 4.

**Robust offline RL**    Robust offline RL [17, 68, 18, 21, 69, 19, 20, 70] aims to develop new offline RL algorithms for cases when the compatibility assumption does not hold, i.e., the offline dataset may not be generated from the task MDP. Among this line of work, [21, 19, 20, 70] study the robust MDP setting, where the dataset can be generated from any MDP within a $\delta$-ball of a nominal MDP. An adversary can choose any MDP from the $\delta$-ball when evaluating the learned policy. [17] considers when the adversary can modify $\epsilon$ fraction of transition tuples in the offline dataset, while [18] considers when up to $K$ trajectories can be added in or removed from the original dataset. [68] uses the formulation of Byzantine-robust: when multiple offline datasets are presented to the learner, among them an $\alpha$ fraction are corrupted. [69] assumes the dataset is generated from the true MDP, but the state presented to the policy during test time can be a corrupted up to $\epsilon$ distance.

Our work is fundamentally different from this line of work in that we study the *inherent* and somewhat *unintended* robustness of offline RL algorithms. We show that, despite originally designed under the compatibility assumption, offline RL algorithms show strong robustness against perturbation of reward. Another salient difference is that robust offline RL typically requires an upper bound on the size of perturbation to the true or nominal MDP, and most of proposed algorithms assume knowledge of this upper bound. However, our work does not make assumptions on the size of perturbation and, moreover, the algorithms we study are not even aware of the existence of such perturbation. Indeed, we empirically show that offline RL algorithms, without any modifications, can succeed even when data reward is a constant or the negation of the true reward. Additionally, in this work, we consider the reward perturbation while most work in robust offline RL focuses more on dynamics perturbation.

**Constrained and safe offline RL**    Our work is related to the literature of constrained and safe offline RL [22, 23, 71, 24, 72]. In constrained offline RL, the agent solves for a CMDP given an offline dataset $\{(s, a, r, c, s')\}$. The goal of constrained offline RL is to produce a policy which maximizes the expected return while ensuring that constraint violation is below a given upper bound. [22] uses an offline policy evaluation oracle to solve for the Lagrangian relaxation of the CMDP. [23] constructs a pessimistic estimation of value and constraint violation. [24, 72] proposes DICE [63]-style algorithms to solve for the optimal state-action occupancy measure. [71] proposes a variant of decision transformer [10] capable of producing policies for different constraint violation upper bounds during test time.

Our work, instead, considers a different scenario where the constraint is only given *implicitly* — through the support of the offline dataset. We assume that any transition in the dataset satisfies the constraint, which can be achieved through intervention during data collection. We show that offline RL can inherently produce approximately safe policies due to its survival instinct. Although our assumption seems more restrictive as in that we require a constraint violation-free dataset, we argue that our assumption is more practical in many safety critical scenarios: compared to executing unsafe actions, it is perhaps more reasonable to intervene before constraint violation happens. We believe that our findings open up new possibilities for applying offline RL as a safe RL oracle, which can be especially promising in an offline-online RL setup [73, 74].

## C    Experiment Details

In the experiments, we train a total of around 16k offline RL agents (see Appendix C.8 for details). In this appendix, we provide details of experiment setup, implementation of algorithms, hyperparameter

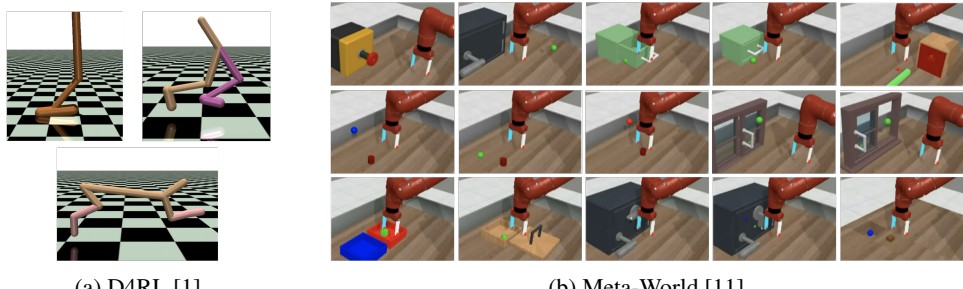

|  (a) D4RL [1] | (b) Meta-World [11] |

Figure 7: We study offline RL with wrong rewards for a variety of tasks: (a) three locomotion tasks from D4RL [1]; hopper (top left), walker2d (top right) and halfcheetah (bottom). the figures are from [75] (b) 15 goal-oriented tasks from Meta-World [11]. The 15 tasks are composed of the 10 tasks from MT-10 (top 2 rows) and the 5 testing tasks from ML-45 (bottom row).

tuning, as well as additional experimental results and ablation studies. At the end of the appendix, we include Table 13 and Table 14 which contain the actual numbers used in generating Fig. 3 and Fig. 6.

## C.1 Benchmark Datasets

In the experiments, we consider three tasks (15 datasets) from D4RL [1], 15 datasets collected from the Meta-World [11] benchmark, and 16 datasets from SafetyGymnasium [49]. The D4RL and Meta-World tasks are visualized in Fig. 7. Below we provide more details on these tasks.

**D4RL** We consider three locomotion tasks from D4RL [1], `hopper`, `walker2d` and `halfcheetah`[6] (see Fig. 7). For each task, we use five datasets of different behavior policy quality: `random`, `medium`, `medium-replay`, `medium expert`, and `expert`.

**hopper:** The goal of `hopper` is to move forward as fast as possible without falling down. An episode terminates when either of the three happens: *1)* the height of the `hopper` is below a threshold, *2)* the body of the `hopper` is too tilted, or *3)* the velocity of any joint is too large. The original true reward function (which has been used to train online RL agents) is

$$r_{\text{hopper}} := \mathbb{1}[\texttt{hopper is alive}] + \frac{1}{\Delta t}\Delta x - 0.001\|\tau\|^2,$$

where $\Delta x$ is the displacement in the forward direction, and $\tau$ is the applied torque. The true reward function is designed such that the `hopper` can receive bonus when staying alive and moving forward. We show that, for a few `hopper` datasets, offline RL agent can succeed without such a carefully designed reward, and even with the negation of the true reward.

**walker2d:** The `walker2d` task is similar to `hopper`, except that the `walker2d` agent has a higher degree of freedom. We observe similarly that offline RL can learn good behaviors without a reward that is specifically designed to encourage staying alive and moving forward.

**halfcheetah:** The `halfcheetah` task differs from the two tasks above since episodes here always have the same length and never terminates early. The true reward of `halfcheetah` is

$$r_{\text{halfcheetah}} := \frac{1}{\Delta t}\Delta x - 0.1\|\tau\|^2.$$

Since the `halfcheetah` datasets (with the exception of the `expert` dataset) do not have a positive bias, we find that offline RL agent is sensitive to reward quality.

**Meta-World** We consider 15 goal-oriented manipulation tasks from [11], see Fig. 7. We use the 10 tasks from the MT-10 set and the 5 testing tasks from the ML-45 set. Each Meta-World task comes with a hand-crafted reward function that encourage progress toward task completion, e.g., when the end-effector grasps the target object, when the object is placed at the goal, etc. We refer readers to [11] for the specific reward functions used for each tasks. We make a few modifications to the Meta-World environment: *1)* we shift and scale the true reward such that the agent receives a

---

[6]We use the v2 version of all three tasks.

reward of range $[-1, 0]$ rather than $[0, 10]$ of the original environment, *2)* we terminate an episode and mark the last state with a terminal flag when the goal is achieved. We show that, data collected by goal-oriented tasks can have a positive bias, and an offline RL agent can make use of such bias to learn the right behavior even with wrong rewards. We would like to clarify that the data generated by the original Meta-World environment (with no terminal signals) does not have a positive bias in general, since all trajectories, success or failure, would have a same length of $500$.

**SafetyGymnasium**  We experiment with 16 datasets from offline SafetyGymnasium [49]. The 16 datasets are given by two embodiments, `point` and `car`, with 4 tasks, `button`, `circle`, `goal`, and `push`, each with 2 difficulty levels.

It is worth noting that [49] focuses on learning from a dataset which contains *both* safe and unsafe behaviors, while our results (Corollary 2) focus on offline RL's ability to produce safe policies given a dataset that *only* contains safe states. Therefore, we use a naïve filtering strategy of removing all transitions with non-zero cost and then run offline RL algorithms on the filtered datasets.

Many algorithms benchmarked in [49] use a cost target, which sets the maximum total cost allowed by safety. [49] uses three different cost targets, $\{20, 40, 80\}$, and reports the normalized total reward and cost for each algorithm averaged over the three targets. By contrast, standard offline RL algorithms do not use such a cost target. We report results of offline RL algorithms according to an "effective cost target" of $34.29$, which is similar to how BC-All results are presented in [49].

## C.2  Implementation Details

We use the implementation of ATAC and PSPI from https://github.com/chinganc/lightATAC. We make a small modification to the ATAC and PSPI policy class. Instead of a $\tanh$-Gaussian policy, we use a scaled $\tanh$-Gaussian policy, i.e., $c \cdot \tanh(\mathcal{N}(\mu, \Sigma))$ with $c \geq 1$. We find scaled $\tanh$-Gaussian policies to be more numerically stable when the data has a lot of on-boundary actions, i.e., actions that takes the value of either $-1$ or $1$ on any dimension. For PSPI, we let the critic $f$ to minimize for $f(s, \pi)$ for all states from the dataset (rather than all initial states). We find it to provide better results when the effective horizon given by the discount factor is smaller than the actual episode length. We use the implementation of IQL and BC from https://github.com/gwthomas/IQL-PyTorch/, and the implementation of CQL from https://github.com/young-geng/CQL. For decision transformer, we use the implementation from https://github.com/kzl/decision-transformer. We feed the decision transformer with wrong rewards (consistent with data reward) during testing.

## C.3  Hyperparameters

We use the default values (ones provided by the original paper or by the implementation we use) for most of the hyperparameters for all algorithms. For each algorithm, we tune one or two hyperparameters (with 4 combinations at most) which affect the degree of pessimism[7]. The values of hyperparamters of all algorithms are given in Table 1–4, where the choices of hyperparameters used in tuning are highlighted in blue. The tuned hyperparameter values for all experiments and all algorithms are provided in Table 9–12.

For D4RL and Meta-World experiments, we tune hyperparameter per each dataset and per reward label. During tuning, we evaluate each combination of hyperparameter(s) for 2 random seeds for D4RL and 3 random seeds for Meta-World. We choose the best-performing values and report the results of these hyperparameters over 10 *new* random seeds (not including the tuning seeds). We find the D4RL scores for the true reward to be close to what is reported in the original papers.

For SafetyGymnasium, we follow procedures from [49] and report results of the best hyperparameters over 3 random seeds. We take the safe agent with the highest cumulative reward if there is at least one safe agent, and take the agent with the lowest cumulative cost if there are no safe agents.

---

[7]For decision transformer, we do not tune hyperparameters since it has no notion of pessimism.

| Hyperparameter | Value |
|---|---|
| Bellman consistency coefficient $\beta$ | $\{0.1, 1, 10, 100\}$ |
| Number of hidden layers | 3 |
| Number of hidden units | 256 |
| Nonlinearity | ReLU |
| Policy distribution | scaled $\tanh$-Gaussian |
| Action scale $c$ | 1.0 for D4RL and SafetyGymnasium |
| | 1.2 for Meta-World |
| Fast learning rate $\eta_{\text{fast}}$ | $5 \times 10^{-4}$ |
| Slow learning rate $\eta_{\text{slow}}$ | $5 \times 10^{-7}$ |
| Minibatch size $|\mathcal{D}_{\text{mini}}|$ | 256 |
| Gradient steps | $10^6$ |
| Warmstart (BC) steps | $10^5$ |
| Temporal difference loss weight $w$ | 0.5 |
| Target smoothing coefficient $\tau$ | 0.005 |
| Discount factor $\gamma$ | 0.99 for D4RL and SafetyGymnasium |
| | 0.998 for Meta-World |
| Minimum target value $V_{\text{min}}$ | $\frac{2}{1-\gamma} \min(-1, \tilde{r}_{\text{min}})$ |
| Maximum target value $V_{\text{max}}$ | $\frac{2}{1-\gamma} \max(1, \tilde{r}_{\text{max}})$ |

Table 1: Hyperparameters for ATAC and PSPI

| Hyperparameter | Value |
|---|---|
| Inverse temperature $\beta$ | $\{0.3, 3\}$ |
| Expectile $\tau$ | $\{0.7, 0.9\}$ |
| Number of hidden layers | 3 |
| Number of hidden units | 256 |
| Nonlinearity | ReLU |
| Policy distribution | Gaussian |
| Learning rate $\eta$ | $3 \times 10^{-4}$ |
| Minibatch size $|\mathcal{D}_{\text{mini}}|$ | 256 |
| Gradient steps | $10^6$ |
| Target smoothing coefficient $\alpha$ | 0.005 |
| Discount factor $\gamma$ | 0.99 for D4RL |
| | 0.998 for Meta-World |

Table 2: Hyperparameters for IQL

## C.4 Episode Lengths for `hopper` and `walker2d` Tasks

We show the episode lengths for `hopper` and `walker2d` tasks in Fig. 8. We observe that policies learned by ATAC and PSPI can consistently keep the agent from falling down for a significant longer period than the behavior policies. The only exception is when running PSPI on `hopper-medium-expert` dataset with random reward. This is potentially due to the fact that we use only 2 seeds for hyperparameter tuning. The average normalized scores and episode lengths during tuning are $101.1$ and $925.0$, respectively, higher than the final values of $58.8$ and $546.0$. We observe that DT policies also tend to keep the agent alive for a longer number of steps than behavior policies.

For IQL and CQL, it is relatively rare for them to achieve long episode lengths when trained with wrong rewards. IQL and CQL, when using negative reward, can keep the agent from falling for longer in `hopper-medium` and `hopper-medium-expert` datasets. We would like to note that our hyperparameters are tuned based on normalized scores rather than episode lengths. The statistics of episode lengths are included in Table 5.

## C.5 Effect of Removing Terminal Transitions

For `hopper` and `walker2d` tasks, we remove the terminal transitions, i.e., the transitions where the `hopper` or walker falls, from the datasets with modified rewards. This is to satisfy the safety

| Hyperparameter | Value |
|---|---|
| Critic pessimism coefficient $\alpha$ | $\{0.1, 1, 10, 100\}$ |
| Number of hidden layers | $2^*$ |
| Number of hidden units | 256 |
| Nonlinearity | ReLU |
| Policy distribution | $\mathrm{tanh}$-Gaussian$^*$ |
| Learning rate $\eta$ | $3 \times 10^{-4}$ |
| Minibatch size $|\mathcal{D}_{\mathrm{mini}}|$ | 256 |
| Gradient steps | $10^6$ |
| Target smoothing coefficient $\tau$ | 0.005 |
| Discount factor $\gamma$ | 0.99 for D4RL |
| | 0.998 for Meta-World |

Table 3: Hyperparameters for CQL. $^*$We include an ablation study of CQL with 3 hidden layers and scaled $\mathrm{tanh}$-Gaussian policy, i.e., the same architecture as ATAC and PSPI, in Appendix C.7.

| Hyperparameter | Value |
|---|---|
| Number of hidden layers | 3 |
| Number of hidden units | 128 |
| Nonlinearity | ReLU |
| Context length $K$ | 20 |
| Dropout | 0.1 |
| Warmup steps | $2 \times 10^2$ |
| Learning rate $\eta$ | $5 \times 10^{-4}$ |
| Minibatch size $|\mathcal{D}_{\mathrm{mini}}|$ | 256 |
| Gradient steps | $7 \times 10^4$ for D4RL |
| | $5 \times 10^3$ for Meta-World |
| Gradient norm clip | 0.25 |
| Weight decay | $10^{-4}$ |
| Return-to-goal conditioning | Return randomly sampled from the top $10\%$ |
| | of trajectory returns in the data, w.r.t. the data reward |

Table 4: Hyperparameters for the decision transformer (DT)

condition in Corollary 2. A terminal signal implicitly adds an absorbing[8], but in this case, unsafe, state into the data distribution $\mu$. This means that staying within data support in a long term would not provide safety, as the agent can stay within support by falling. For D4RL datasets in particular, we need to remove terminal transitions (rather than just the terminal flag) since terminal transitions in the original datasets do not contain valid next states.[9]

This, however, means that there are two underlying variables: *1)* the data reward, and *2)* whether terminal transitions are removed, when comparing offline RL with wrong rewards and offline RL on the original dataset. To separate the effect of two variables, we study the performance of offline RL algorithms with the original true reward but the terminal transitions removed. The results[10] are listed in the No-terminal column in Table 13. We observe that ATAC and PSPI with the original true reward show similar results whether the terminal transitions are removed. IQL and CQL with true reward, interestingly, does significantly worse when the terminal transitions are removed. This is potentially due to the increased instability of target network [76, 77] when there are no terminal signals.

### C.6 Results on Safety Gymnasium Datasets

In Table 6, we present results on `point` datasets and `car` from Safety Gymnasium [49]. For baselines, we select the best none-BC Algorithms from [49]: the best-performing offline safe RL agent selected

---

[8]This is also why we mark goal completion as terminal in Meta-World experiments. For goal-oriented tasks, terminal signal upon task completion implicitly creates an absorbing goal state in the data distribution.

[9]The authors of D4RL datasets set the next state to be a place-holder value when a transition is marked as terminal.

[10]We do not study decision transformer (DT) in this ablation since DT does not use terminal signals.

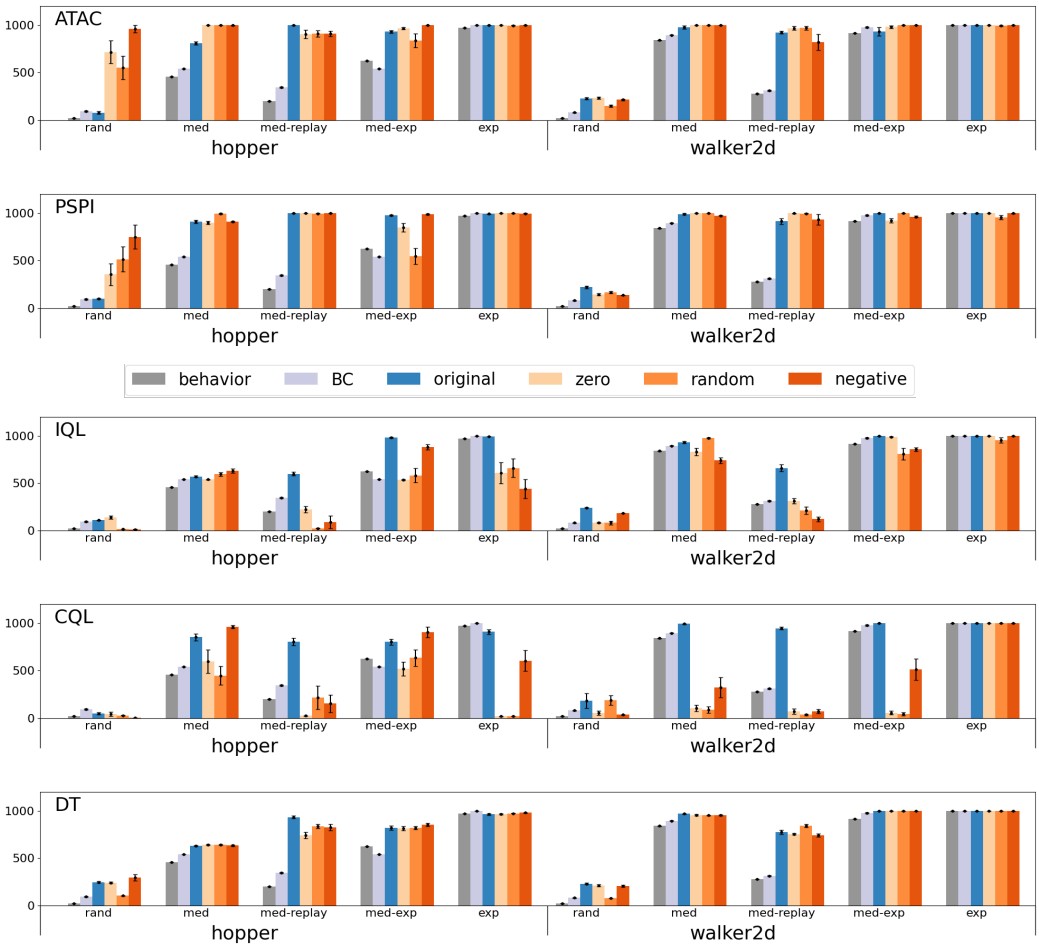

Figure 8: Episode lengths for `hopper` and `walker2d` datasets from D4RL. The mean and standard error of episode lengths are computed across 10 random seeds. For each random seed, we evaluate the final policy of each algorithm over 50 episodes. Note that the hyperparameters are tuned based on normalized scores rather than episode lengths.

using the same criterion (safe agent with highest reward if there is at least one safe agent, and agent with lowest cost otherwise). We observe that ATAC and PSPI with a naïve filtering strategy can achieve comparable performance with state-of-the-art offline safe RL algorithms benchmarked in [49].

## C.7 Ablation Study on CQL Architecture

Our results are generated by CQL with the architecture in Table 3, which is given by the original implementation. In this study, we examine the performance of CQL when using the same architecture as ATAC and PSPI, i.e., with 3 hidden layers and scaled $\tanh$-Gaussian. The normalized scores for D4RL datasets and success rate for Meta-World tasks are included in Table 7 and Table 8, respectively. We find the performance of CQL with 3 hidden layers and scaled $\tanh$-Gaussian policy, denoted as $CQL_3$, to be similar with the performance of CQL with the original architecture.

## C.8 Compute Usage

For ATAC, PSPI, IQL, CQL and BC, each run takes around $5.5$ hours on an NC4as_T4_v3 Azure virtual machine for D4RL, Meta-World, and SafetyGymnasium experiments. For D4RL experiments (including ablation studies), we use a total of $5$ (algorithms) $\times$ $15$ (D4RL datasets) $\times$ $5$ (reward types) $\times$ $4$ (hyperparameters) $\times$ $2$ (seeds) $=$ $3000$ runs for hyperparameter tuning and

| Dataset | Behavior | BC | Algo | Original | Zero | Random | Negative | No-terminal |
|---|---|---|---|---|---|---|---|---|
| hopper random | 22.1 | 93.2±5.0 | ATAC | 79.3±13.0 | 714.2±120.0 | 552.7±123.4 | 958.8±39.1 | 567.2±144.9 |
| | | | PSPI | 99.2±6.2 | 354.5±113.0 | 513.9±131.0 | 749.1±124.5 | 517.3±142.9 |
| | | | IQL | 108.7±4.0 | 137.1±18.0 | 12.8±0.1 | 12.1±2.7 | 14.5±1.0 |
| | | | CQL | 48.4±12.5 | 41.8±20.9 | 28.4±3.8 | 6.0±0.0 | 16.7±10.1 |
| | | | DT | 245.2±8.2 | 239.7±8.6 | 103.2±2.0 | 295.1±32.8 | – |
| hopper medium | 457.2 | 540.5±1.2 | ATAC | 808.4±15.3 | 999.7±0.3 | 998.9±1.0 | 997.6±1.9 | 840.2±19.8 |
| | | | PSPI | 906.8±16.9 | 897.7±16.2 | 993.0±6.4 | 907.8±6.0 | 833.9±29.7 |
| | | | IQL | 571.2±8.5 | 538.7±4.0 | 594.8±20.3 | 631.4±21.7 | 446.4±65.8 |
| | | | CQL | 851.1±35.9 | 594.7±123.9 | 447.2±97.4 | 962.2±17.0 | 503.7±123.0 |
| | | | DT | 630.4±6.7 | 641.1±8.2 | 638.8±6.1 | 633.4±9.1 | – |
| hopper medium replay | 197.0 | 345.4±7.0 | ATAC | 1000.0±0.0 | 904.5±44.3 | 909.4±35.3 | 909.6±29.5 | 1000.0±0.0 |
| | | | PSPI | 1000.0±0.0 | 1000.0±0.0 | 995.9±3.9 | 1000.0±0.0 | 1000.0±0.0 |
| | | | IQL | 598.5±19.8 | 222.5±33.2 | 22.3±4.2 | 88.1±65.5 | 100.9±53.6 |
| | | | CQL | 801.9±38.2 | 24.9±2.6 | 217.5±123.7 | 153.8±91.8 | 116.4±93.2 |
| | | | DT | 934.5±13.6 | 742.3±33.4 | 836.2±20.6 | 825.6±31.2 | – |
| hopper medium expert | 622.2 | 541.8±1.1 | ATAC | 930.1±13.1 | 965.3±10.5 | 836.5±71.7 | 998.3±1.6 | 885.1±36.3 |
| | | | PSPI | 974.5±6.1 | 848.0±44.6 | 546.0±84.4 | 988.9±4.2 | 969.8±5.3 |
| | | | IQL | 980.4±5.1 | 535.4±3.4 | 581.2±74.7 | 883.2±28.5 | 735.3±62.8 |
| | | | CQL | 800.8±31.4 | 516.6±72.7 | 633.1±87.9 | 901.9±55.7 | 489.1±103.0 |
| | | | DT | 817.4±22.5 | 813.8±22.0 | 817.8±17.0 | 852.1±17.4 | – |
| hopper expert | 972.8 | 998.4±0.6 | ATAC | 998.8±0.5 | 997.3±1.1 | 995.3±2.1 | 998.8±1.2 | 999.9±0.1 |
| | | | PSPI | 991.5±1.5 | 998.8±0.9 | 998.4±0.8 | 995.8±1.7 | 992.1±3.0 |
| | | | IQL | 993.3±1.7 | 607.7±112.9 | 658.7±97.1 | 439.6±102.0 | 238.3±77.3 |
| | | | CQL | 906.7±25.9 | 19.3±1.9 | 19.9±1.4 | 603.2±110.0 | 17.7±1.2 |
| | | | DT | 963.1±6.8 | 966.0±4.9 | 972.9±2.9 | 979.4±2.9 | – |
| walker2d random | 20.4 | 81.2±2.1 | ATAC | 227.6±10.2 | 231.0±11.0 | 149.6±10.4 | 214.8±9.2 | 189.0±19.8 |
| | | | PSPI | 221.1±13.7 | 143.8±10.5 | 165.7±9.8 | 135.8±1.2 | 135.9±0.5 |
| | | | IQL | 240.5±3.8 | 83.7±4.9 | 79.9±21.4 | 182.3±1.6 | 62.8±23.0 |
| | | | CQL | 182.9±79.5 | 56.0±23.1 | 188.7±50.9 | 34.7±5.1 | 37.2±10.2 |
| | | | DT | 228.4±7.6 | 210.9±10.2 | 75.2±1.2 | 203.3±11.1 | – |
| walker2d medium | 839.6 | 895.3±2.4 | ATAC | 976.9±17.6 | 999.7±0.3 | 997.8±1.5 | 999.8±0.2 | 990.2±5.9 |
| | | | PSPI | 989.1±7.2 | 998.2±1.7 | 1000.0±0.0 | 973.0±4.9 | 991.7±4.7 |
| | | | IQL | 930.1±12.3 | 831.1±37.3 | 876.9±7.4 | 739.1±30.7 | 517.4±11.2 |
| | | | CQL | 991.0±3.0 | 105.3±34.7 | 86.9±32.7 | 321.5±107.8 | 130.4±35.0 |
| | | | DT | 968.7±3.3 | 955.6±7.5 | 951.5±4.7 | 954.9±6.9 | – |
| walker2d medium replay | 276.3 | 312.7±5.8 | ATAC | 923.1±16.4 | 967.8±21.5 | 968.4±19.8 | 817.4±83.6 | 985.8±10.0 |
| | | | PSPI | 914.4±30.4 | 997.1±2.7 | 995.3±3.2 | 931.8±53.2 | 997.3±1.2 |
| | | | IQL | 659.9±37.5 | 313.5±27.5 | 211.1±37.1 | 119.5±25.1 | 41.8±3.4 |
| | | | CQL | 945.1±14.9 | 71.3±27.5 | 35.1±6.9 | 72.8±23.3 | 87.5±26.7 |
| | | | DT | 776.1±21.3 | 755.5±10.7 | 841.9±14.8 | 740.8±17.8 | – |
| walker2d medium expert | 912.8 | 976.9±3.5 | ATAC | 929.8±44.4 | 979.4±15.9 | 999.4±0.6 | 1000.0±0.0 | 960.3±36.0 |
| | | | PSPI | 1000.0±0.0 | 918.9±21.3 | 960.2±8.3 | 960.2±8.3 | 987.9±9.8 |
| | | | IQL | 999.6±0.4 | 989.0±2.9 | 808.8±60.9 | 856.7±21.7 | 334.1±12.5 |
| | | | CQL | 998.4±1.5 | 55.8±20.4 | 44.3±17.4 | 514.5±112.3 | 12.6±2.2 |
| | | | DT | 1000.0±0.0 | 999.1±0.9 | 1000.0±0.0 | 999.5±0.4 | – |
| walker2d expert | 999.0 | 1000.0±0.0 | ATAC | 1000.0±0.0 | 1000.0±0.0 | 993.6±3.6 | 1000.0±0.0 | 1000.0±0.0 |
| | | | PSPI | 1000.0±0.0 | 1000.0±0.0 | 951.7±22.1 | 997.4±1.7 | 1000.0±0.0 |
| | | | IQL | 1000.0±0.0 | 1000.0±0.0 | 955.3±29.0 | 1000.0±0.0 | 1000.0±0.0 |
| | | | CQL | 1000.0±0.0 | 1000.0±0.0 | 1000.0±0.0 | 1000.0±0.0 | 998.3±1.6 |
| | | | DT | 1000.0±0.0 | 1000.0±0.0 | 1000.0±0.0 | 1000.0±0.0 | – |

Table 5: Episode lengths for hopper and walker2d datasets. The mean and standard error of episode lengths are computed across 10 random seeds. For each random seed, we evaluate the final policy of each algorithm over 50 episodes.

6 (algorithms) × 15 (D4RL datasets) × 5 (reward types) × 10 (seeds) = 4500 runs for generating the final results. For Meta-World experiments, we use 5 (algorithms) × 15 (Meta-World datasets) × 4 (reward types) × 4 (hyperparameters) × 3 (seeds) = 3600 runs for hyperparameter tuning and 6 (algorithms) × 15 (Meta-World datasets) × 4 (reward types) × 10 (seeds) = 3600 runs for generating the final results. SafetyGymnasium experiments require a total of 2 (algorithms) × 16 (SafetyGymnasium datasets) × 4 (hyperparameters) × 3 (seeds) = 384 runs. This amount to 15084 runs or 82962 hours of training on NC4as_T4_v3 virtual machines.

For DT, we do not tune any parameters at large scale. Each run takes around 2.5 hours on an NC6s_v2 or ND6s Azure virtual machine. We train a total of 30 (D4RL and Meta-World datasets) × 4 (reward types) × 10 (seeds) = 1200 DT agents.

| Task | ATAC | | PSPI | | Best None-BC Algorithm from [49] | | |
|---|---|---|---|---|---|---|---|
| | reward ↑ | cost ↓ | reward ↑ | cost ↓ | reward ↑ | cost ↓ | algorithm |
| PointButton1 | 0.08 | 0.82 | 0.10 | 1.04 | 0.13 | 1.35 | COpiDICE [24] |
| PointButton2 | 0.14 | 0.89 | 0.19 | 1.29 | 0.15 | 1.51 | COpiDICE [24] |
| PointCircle1 | 0.40 | 2.79 | 0.34 | 2.39 | 0.59 | 0.69 | CDT [71] |
| PointCircle2 | 0.61 | 4.45 | 0.55 | 2.84 | 0.64 | 1.05 | CDT [71] |
| PointGoal1 | 0.58 | 0.94 | 0.59 | 0.90 | 0.71 | 0.98 | BCQ-Lag [23] |
| PointGoal2 | 0.55 | 2.30 | 0.45 | 2.55 | 0.40 | 1.31 | CPQ [23] |
| PointPush1 | 0.18 | 0.51 | 0.20 | 0.85 | 0.33 | 0.86 | BCQ-Lag [23] |
| PointPush2 | 0.10 | 0.68 | 0.10 | 0.81 | 0.23 | 0.99 | BCQ-Lag [23] |
| CarButton1 | -0.06 | 1.25 | -0.08 | 1.11 | 0.21 | 1.60 | CDT [71] |
| CarButton2 | -0.11 | 1.22 | -0.08 | 1.04 | 0.13 | 1.58 | CDT [71] |
| CarCircle1 | 0.66 | 5.51 | 0.64 | 5.92 | 0.60 | 1.73 | CDT [71] |
| CarCircle2 | 0.64 | 5.96 | 0.64 | 5.99 | 0.66 | 2.53 | CDT [71] |
| CarGoal1 | 0.50 | 0.99 | 0.41 | 0.78 | 0.47 | 0.78 | BCQ-Lag [23] |
| CarGoal2 | 0.24 | 1.00 | 0.24 | 1.05 | 0.25 | 0.91 | COptiDICE [24] |
| CarPush1 | 0.33 | 0.96 | 0.32 | 0.75 | 0.31 | 0.40 | CDT [71] |
| CarPush2 | 0.10 | 0.95 | 0.11 | 0.81 | 0.09 | 1.07 | COptiDICE [24] |

Table 6: Results on `point` and `car` datasets from offline SafetyGymnasium [49] over 3 random seeds. Cumulative reward and cost are normalized as described in [49]. An agent is considered safe if the cumulative cost is no larger than 1. Unsafe agent with cumulative cost more than 1 (total cost 34.29) is shown in gray.

| Dataset | Algo | Original | Zero | Random | Negative | No-terminal |
|---|---|---|---|---|---|---|
| hopper-random | CQL | 3.0±0.7 | 2.0±1.1 | 1.8±0.2 | 0.7±0.0 | 1.2±0.5 |
| | $CQL_3$ | 11.2±2.2 | 2.0±0.4 | 7.7±3.5 | 3.9±2.9 | 10.1±3.2 |
| hopper-medium | CQL | 85.0±3.7 | 55.0±11.7 | 36.9±9.8 | 89.0±1.5 | 46.3±11.5 |
| | $CQL_3$ | 48.0±12.2 | 61.7±12.2 | 17.3±9.8 | 75.8±6.9 | 32.8±10.3 |
| hopper-medium-replay | CQL | 79.7±3.7 | 1.9±0.1 | 7.6±3.7 | 5.8±2.7 | 4.4±2.8 |
| | $CQL_3$ | 100.3±1.6 | 1.8±0.0 | 1.8±0.0 | 1.6±0.2 | 1.8±0.0 |
| hopper-medium-expert | CQL | 88.8±3.6 | 49.7±8.1 | 60.7±9.4 | 88.6±6.1 | 47.4±11.0 |
| | $CQL_3$ | 104.7±1.4 | 40.6±10.7 | 17.3±9.0 | 48.2±8.7 | 30.0±9.5 |
| walker2d-random | CQL | 4.0±1.8 | 0.9±0.9 | 4.3±1.4 | -0.4±0.0 | 0.2±0.4 |
| | $CQL_3$ | 4.8±0.2 | 1.4±0.8 | 3.3±1.2 | 0.9±0.6 | -0.2±0.1 |
| walker2d-medium | CQL | 81.4±0.3 | 2.0±0.9 | 1.2±0.7 | 14.5±7.4 | 2.1±0.8 |
| | $CQL_3$ | 81.0±0.4 | 5.0±4.7 | -0.2±0.0 | 13.3±7.7 | 0.1±0.3 |
| walker2d-medium-replay | CQL | 74.1±1.2 | 0.5±0.3 | -0.2±0.1 | 0.2±0.3 | 0.4±0.4 |
| | $CQL_3$ | 76.8±1.3 | -0.2±0.1 | 0.1±0.1 | 0.1±0.3 | 0.3±0.3 |
| walker2d-medium-expert | CQL | 109.3±0.3 | 0.8±0.5 | 0.1±0.2 | 28.0±9.1 | -0.2±0.0 |
| | $CQL_3$ | 108.0±0.6 | -0.2±0.1 | -0.3±0.1 | 26.8±11.0 | -0.2±0.0 |
| halfcheetah-random | CQL | 30.7±0.6 | 1.0±0.8 | -1.4±0.3 | -5.4±0.3 | – |
| | $CQL_3$ | 31.3±0.6 | 0.5±0.4 | 1.9±0.1 | -7.0±0.4 | – |
| halfcheetah-medium | CQL | 57.2±1.5 | 43.2±0.1 | 43.5±0.1 | 42.7±0.1 | – |
| | $CQL_3$ | 65.1±0.6 | 43.1±0.1 | 43.3±0.1 | 42.2±0.1 | – |
| halfcheetah-medium-replay | CQL | 48.2±5.4 | 34.2±0.7 | 33.1±0.9 | 32.4±1.0 | – |
| | $CQL_3$ | 47.8±5.2 | 38.3±1.0 | 37.7±1.1 | 32.6±1.2 | – |
| halfcheetah-medium-expert | CQL | 75.2±1.0 | 81.2±2.1 | 69.6±2.8 | 42.4±0.2 | – |
| | $CQL_3$ | 88.0±1.7 | 75.5±1.7 | 80.0±2.5 | 42.6±0.1 | – |

Table 7: D4RL normalized scores for CQL with the original architecture in Table 3 (CQL) and with the same architecture as ATAC and PSPI ($CQL_3$), i.e., with 3 hidden layers and scaled $\tanh$-Gaussian. We find that CQL performs similarly under the two choices of architecture.

| Dataset | Algo | Original | Zero | Random | Negative |
|---|---|---|---|---|---|
| button-press | CQL | 0.88±0.07 | 0.83±0.10 | 0.84±0.09 | 0.94±0.03 |
| | CQL$_3$ | 0.78±0.10 | 0.90±0.06 | 0.73±0.11 | 0.68±0.11 |
| door-open | CQL | 0.82±0.09 | 0.07±0.07 | 0.12±0.08 | 0.00±0.00 |
| | CQL$_3$ | 0.47±0.15 | 0.29±0.14 | 0.00±0.00 | 0.38±0.15 |
| drawer-close | CQL | 0.91±0.04 | 0.99±0.01 | 0.96±0.02 | 0.97±0.02 |
| | CQL$_3$ | 0.99±0.01 | 0.93±0.03 | 0.97±0.02 | 0.98±0.01 |
| drawer-open | CQL | 0.78±0.04 | 0.66±0.07 | 0.65±0.07 | 0.84±0.03 |
| | CQL$_3$ | 0.86±0.03 | 0.79±0.09 | 0.78±0.06 | 0.80±0.05 |
| peg-insert-side | CQL | 0.08±0.02 | 0.00±0.00 | 0.02±0.01 | 0.00±0.00 |
| | CQL$_3$ | 0.10±0.02 | 0.05±0.03 | 0.08±0.02 | 0.03±0.02 |
| pick-place | CQL | 0.01±0.00 | 0.06±0.02 | 0.05±0.02 | 0.07±0.03 |
| | CQL$_3$ | 0.00±0.00 | 0.00±0.00 | 0.00±0.00 | 0.00±0.00 |
| push | CQL | 0.00±0.00 | 0.01±0.00 | 0.01±0.00 | 0.03±0.02 |
| | CQL$_3$ | 0.03±0.01 | 0.00±0.00 | 0.00±0.00 | 0.01±0.01 |
| reach | CQL | 0.17±0.03 | 0.29±0.04 | 0.25±0.05 | 0.23±0.03 |
| | CQL$_3$ | 0.35±0.05 | 0.35±0.03 | 0.37±0.05 | 0.22±0.06 |
| window-close | CQL | 1.00±0.00 | 1.00±0.00 | 0.59±0.13 | 0.55±0.13 |
| | CQL$_3$ | 0.97±0.03 | 0.75±0.12 | 0.77±0.12 | 0.98±0.02 |
| window-open | CQL | 0.83±0.03 | 0.82±0.05 | 0.92±0.02 | 0.83±0.07 |
| | CQL$_3$ | 0.84±0.07 | 0.94±0.02 | 0.93±0.02 | 0.77±0.09 |
| bin-picking | CQL | 0.00±0.00 | 0.02±0.02 | 0.00±0.00 | 0.00±0.00 |
| | CQL$_3$ | 0.00±0.00 | 0.00±0.00 | 0.00±0.00 | 0.00±0.00 |
| box-close | CQL | 0.04±0.01 | 0.07±0.02 | 0.01±0.00 | 0.05±0.01 |
| | CQL$_3$ | 0.03±0.02 | 0.00±0.00 | 0.01±0.00 | 0.01±0.00 |
| door-lock | CQL | 0.75±0.04 | 0.61±0.10 | 0.61±0.11 | 0.60±0.08 |
| | CQL$_3$ | 0.63±0.06 | 0.64±0.09 | 0.57±0.10 | 0.62±0.09 |
| door-unlock | CQL | 0.87±0.03 | 0.77±0.09 | 0.89±0.04 | 0.93±0.03 |
| | CQL$_3$ | 0.92±0.02 | 0.73±0.11 | 0.71±0.11 | 0.76±0.11 |
| hand-insert | CQL | 0.13±0.04 | 0.22±0.06 | 0.34±0.04 | 0.32±0.03 |
| | CQL$_3$ | 0.04±0.03 | 0.08±0.03 | 0.18±0.03 | 0.08±0.02 |

Table 8: Meta-World success rate for CQL with the original architecture in Table 3 (CQL) and with the same architecture as ATAC and PSPI (CQL$_3$), i.e., with 3 hidden layers and scaled $\tanh$-Gaussian. We find that CQL performs similarly under the two choices of architecture.

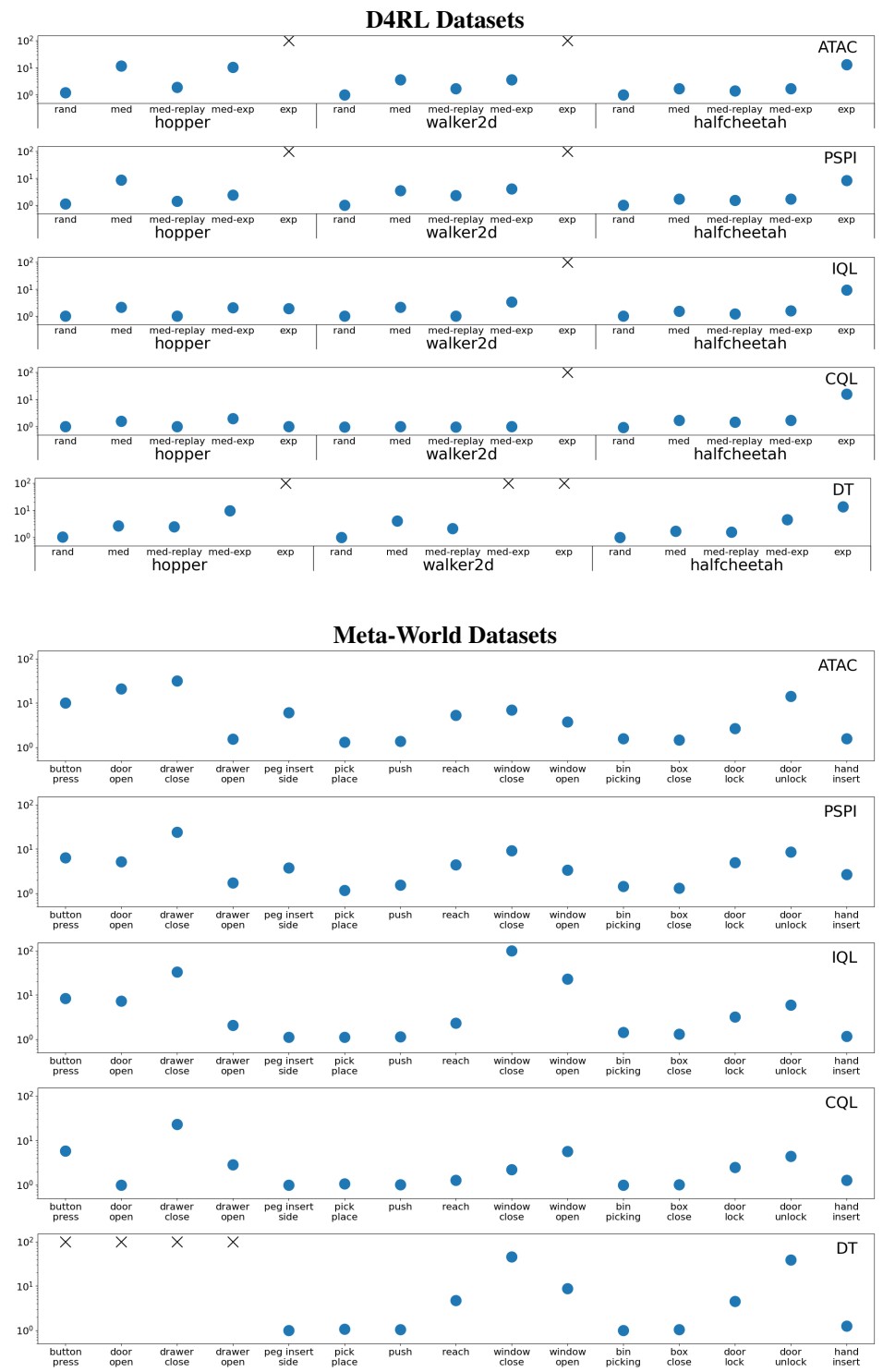

Figure 9: Estimated positive data bias of D4RL and Meta-World datasets given all 6 offline RL algorithms, ATAC, PSPI, IQL, CQL, and DT. Datasets marked by "×" have infinite positive bias.

| Dataset | Original | Zero | Random | Negative | No-terminal |
|---|---|---|---|---|---|
| hopper-random | $\beta = 100.0$ | $\beta = 100.0$ | $\beta = 10.0$ | $\beta = 1.0$ | $\beta = 10.0$ |
| hopper-medium | $\beta = 10.0$ | $\beta = 100.0$ | $\beta = 100.0$ | $\beta = 1.0$ | $\beta = 100.0$ |
| hopper-medium-replay | $\beta = 100.0$ | $\beta = 1.0$ | $\beta = 1.0$ | $\beta = 0.1$ | $\beta = 100.0$ |
| hopper-medium-expert | $\beta = 1.0$ | $\beta = 10.0$ | $\beta = 10.0$ | $\beta = 1.0$ | $\beta = 1.0$ |
| hopper-expert | $\beta = 0.1$ | $\beta = 1.0$ | $\beta = 1.0$ | $\beta = 0.1$ | $\beta = 0.1$ |
| | | | | | |
| walker2d-random | $\beta = 10.0$ | $\beta = 0.1$ | $\beta = 1.0$ | $\beta = 0.1$ | $\beta = 100.0$ |
| walker2d-medium | $\beta = 10.0$ | $\beta = 1.0$ | $\beta = 1.0$ | $\beta = 0.1$ | $\beta = 100.0$ |
| walker2d-medium-replay | $\beta = 100.0$ | $\beta = 10.0$ | $\beta = 10.0$ | $\beta = 10.0$ | $\beta = 100.0$ |
| walker2d-medium-expert | $\beta = 10.0$ | $\beta = 10.0$ | $\beta = 1.0$ | $\beta = 0.1$ | $\beta = 10.0$ |
| walker2d-expert | $\beta = 1.0$ | $\beta = 100.0$ | $\beta = 0.1$ | $\beta = 0.1$ | $\beta = 1.0$ |
| | | | | | |
| halfcheetah-random | $\beta = 0.1$ | $\beta = 0.1$ | $\beta = 0.1$ | $\beta = 0.1$ | – |
| halfcheetah-medium | $\beta = 1.0$ | $\beta = 100.0$ | $\beta = 1.0$ | $\beta = 0.1$ | – |
| halfcheetah-medium-replay | $\beta = 1.0$ | $\beta = 1.0$ | $\beta = 1.0$ | $\beta = 0.1$ | – |
| halfcheetah-medium-expert | $\beta = 0.1$ | $\beta = 0.1$ | $\beta = 0.1$ | $\beta = 0.1$ | – |
| halfcheetah-expert | $\beta = 0.1$ | $\beta = 100.0$ | $\beta = 0.1$ | $\beta = 0.1$ | – |
| | | | | | |
| button-press | $\beta = 10.0$ | $\beta = 10.0$ | $\beta = 1.0$ | $\beta = 100.0$ | – |
| door-open | $\beta = 100.0$ | $\beta = 10.0$ | $\beta = 10.0$ | $\beta = 1.0$ | – |
| drawer-close | $\beta = 1.0$ | $\beta = 10.0$ | $\beta = 10.0$ | $\beta = 100.0$ | – |
| drawer-open | $\beta = 10.0$ | $\beta = 1.0$ | $\beta = 10.0$ | $\beta = 10.0$ | – |
| peg-insert-side | $\beta = 1.0$ | $\beta = 1.0$ | $\beta = 1.0$ | $\beta = 1.0$ | – |
| pick-place | $\beta = 1.0$ | $\beta = 10.0$ | $\beta = 1.0$ | $\beta = 1.0$ | – |
| push | $\beta = 1.0$ | $\beta = 1.0$ | $\beta = 1.0$ | $\beta = 1.0$ | – |
| reach | $\beta = 10.0$ | $\beta = 100.0$ | $\beta = 10.0$ | $\beta = 1.0$ | – |
| window-close | $\beta = 1.0$ | $\beta = 1.0$ | $\beta = 1.0$ | $\beta = 1.0$ | – |
| window-open | $\beta = 0.1$ | $\beta = 1.0$ | $\beta = 10.0$ | $\beta = 100.0$ | – |
| bin-picking | $\beta = 1.0$ | $\beta = 1.0$ | $\beta = 1.0$ | $\beta = 1.0$ | – |
| box-close | $\beta = 1.0$ | $\beta = 1.0$ | $\beta = 1.0$ | $\beta = 1.0$ | – |
| door-lock | $\beta = 10.0$ | $\beta = 1.0$ | $\beta = 1.0$ | $\beta = 0.1$ | – |
| door-unlock | $\beta = 100.0$ | $\beta = 10.0$ | $\beta = 10.0$ | $\beta = 10.0$ | – |
| hand-insert | $\beta = 10.0$ | $\beta = 1.0$ | $\beta = 1.0$ | $\beta = 1.0$ | – |
| | | | | | |
| point-button1 | $\beta = 1.0$ | – | – | – | – |
| point-button2 | $\beta = 1.0$ | – | – | – | – |
| point-circle1 | $\beta = 0.1$ | – | – | – | – |
| point-circle2 | $\beta = 0.1$ | – | – | – | – |
| point-goal1 | $\beta = 0.1$ | – | – | – | – |
| point-goal2 | $\beta = 1.0$ | – | – | – | – |
| point-push1 | $\beta = 10.0$ | – | – | – | – |
| point-push2 | $\beta = 1.0$ | – | – | – | – |
| car-button1 | $\beta = 0.1$ | – | – | – | – |
| car-button2 | $\beta = 1.0$ | – | – | – | – |
| car-circle1 | $\beta = 0.1$ | – | – | – | – |
| car-circle2 | $\beta = 100.0$ | – | – | – | – |
| car-goal1 | $\beta = 10.0$ | – | – | – | – |
| car-goal2 | $\beta = 1.0$ | – | – | – | – |
| car-push1 | $\beta = 100.0$ | – | – | – | – |
| car-push2 | $\beta = 0.1$ | – | – | – | – |

Table 9: Tuned hyperparameter values for ATAC

| Dataset | Original | Zero | Random | Negative | No-terminal |
|---|---|---|---|---|---|
| hopper-random | $\beta = 10.0$ | $\beta = 10.0$ | $\beta = 100.0$ | $\beta = 100.0$ | $\beta = 100.0$ |
| hopper-medium | $\beta = 100.0$ | $\beta = 10.0$ | $\beta = 100.0$ | $\beta = 1.0$ | $\beta = 100.0$ |
| hopper-medium-replay | $\beta = 10.0$ | $\beta = 10.0$ | $\beta = 10.0$ | $\beta = 1.0$ | $\beta = 1.0$ |
| hopper-medium-expert | $\beta = 1.0$ | $\beta = 10.0$ | $\beta = 100.0$ | $\beta = 1.0$ | $\beta = 1.0$ |
| hopper-expert | $\beta = 1.0$ | $\beta = 1.0$ | $\beta = 1.0$ | $\beta = 0.1$ | $\beta = 1.0$ |
| | | | | | |
| walker2d-random | $\beta = 10.0$ | $\beta = 10.0$ | $\beta = 10.0$ | $\beta = 1.0$ | $\beta = 1.0$ |
| walker2d-medium | $\beta = 10.0$ | $\beta = 10.0$ | $\beta = 10.0$ | $\beta = 0.1$ | $\beta = 10.0$ |
| walker2d-medium-replay | $\beta = 10.0$ | $\beta = 10.0$ | $\beta = 10.0$ | $\beta = 1.0$ | $\beta = 100.0$ |
| walker2d-medium-expert | $\beta = 1.0$ | $\beta = 1.0$ | $\beta = 10.0$ | $\beta = 0.1$ | $\beta = 10.0$ |
| walker2d-expert | $\beta = 1.0$ | $\beta = 1.0$ | $\beta = 10.0$ | $\beta = 0.1$ | $\beta = 1.0$ |
| | | | | | |
| halfcheetah-random | $\beta = 100.0$ | $\beta = 100.0$ | $\beta = 0.1$ | $\beta = 1.0$ | – |
| halfcheetah-medium | $\beta = 1.0$ | $\beta = 10.0$ | $\beta = 1.0$ | $\beta = 0.1$ | – |
| halfcheetah-medium-replay | $\beta = 10.0$ | $\beta = 1.0$ | $\beta = 10.0$ | $\beta = 0.1$ | – |
| halfcheetah-medium-expert | $\beta = 0.1$ | $\beta = 1.0$ | $\beta = 0.1$ | $\beta = 0.1$ | – |
| halfcheetah-expert | $\beta = 0.1$ | $\beta = 1.0$ | $\beta = 10.0$ | $\beta = 0.1$ | – |
| | | | | | |
| button-press | $\beta = 0.1$ | $\beta = 10.0$ | $\beta = 1.0$ | $\beta = 100.0$ | – |
| door-open | $\beta = 10.0$ | $\beta = 1.0$ | $\beta = 10.0$ | $\beta = 0.1$ | – |
| drawer-close | $\beta = 1.0$ | $\beta = 1.0$ | $\beta = 10.0$ | $\beta = 10.0$ | – |
| drawer-open | $\beta = 100.0$ | $\beta = 10.0$ | $\beta = 10.0$ | $\beta = 10.0$ | – |
| peg-insert-side | $\beta = 1.0$ | $\beta = 1.0$ | $\beta = 1.0$ | $\beta = 10.0$ | – |
| pick-place | $\beta = 10.0$ | $\beta = 1.0$ | $\beta = 10.0$ | $\beta = 1.0$ | – |
| push | $\beta = 100.0$ | $\beta = 10.0$ | $\beta = 1.0$ | $\beta = 1.0$ | – |
| reach | $\beta = 100.0$ | $\beta = 100.0$ | $\beta = 10.0$ | $\beta = 10.0$ | – |
| window-close | $\beta = 0.1$ | $\beta = 0.1$ | $\beta = 1.0$ | $\beta = 0.1$ | – |
| window-open | $\beta = 10.0$ | $\beta = 10.0$ | $\beta = 10.0$ | $\beta = 0.1$ | – |
| bin-picking | $\beta = 1.0$ | $\beta = 1.0$ | $\beta = 1.0$ | $\beta = 1.0$ | – |
| box-close | $\beta = 1.0$ | $\beta = 1.0$ | $\beta = 1.0$ | $\beta = 1.0$ | – |
| door-lock | $\beta = 1.0$ | $\beta = 1.0$ | $\beta = 10.0$ | $\beta = 10.0$ | – |
| door-unlock | $\beta = 100.0$ | $\beta = 10.0$ | $\beta = 10.0$ | $\beta = 10.0$ | – |
| hand-insert | $\beta = 10.0$ | $\beta = 100.0$ | $\beta = 1.0$ | $\beta = 1.0$ | – |
| | | | | | |
| point-button1 | $\beta = 1.0$ | – | – | – | – |
| point-button2 | $\beta = 10.0$ | – | – | – | – |
| point-circle1 | $\beta = 1.0$ | – | – | – | – |
| point-circle2 | $\beta = 1.0$ | – | – | – | – |
| point-goal1 | $\beta = 0.1$ | – | – | – | – |
| point-goal2 | $\beta = 1.0$ | – | – | – | – |
| point-push1 | $\beta = 0.1$ | – | – | – | – |
| point-push2 | $\beta = 10.0$ | – | – | – | – |
| car-button1 | $\beta = 0.1$ | – | – | – | – |
| car-button2 | $\beta = 1.0$ | – | – | – | – |
| car-circle1 | $\beta = 0.1$ | – | – | – | – |
| car-circle2 | $\beta = 100.0$ | – | – | – | – |
| car-goal1 | $\beta = 1.0$ | – | – | – | – |
| car-goal2 | $\beta = 1.0$ | – | – | – | – |
| car-push1 | $\beta = 100.0$ | – | – | – | – |
| car-push2 | $\beta = 1.0$ | – | – | – | – |

Table 10: Tuned hyperparameter values for PSPI

| Dataset | Original | Zero | Random | Negative | No-terminal |
|---|---|---|---|---|---|
| hopper-random | $\beta = 0.3, \tau = 0.7$ | $\beta = 3.0, \tau = 0.9$ | $\beta = 0.3, \tau = 0.7$ | $\beta = 0.3, \tau = 0.9$ | $\beta = 3.0, \tau = 0.7$ |
| hopper-medium | $\beta = 0.3, \tau = 0.7$ | $\beta = 3.0, \tau = 0.7$ | $\beta = 0.3, \tau = 0.7$ | $\beta = 0.3, \tau = 0.7$ | $\beta = 0.3, \tau = 0.7$ |
| hopper-medium-replay | $\beta = 3.0, \tau = 0.7$ | $\beta = 3.0, \tau = 0.7$ | $\beta = 3.0, \tau = 0.7$ | $\beta = 0.3, \tau = 0.9$ | $\beta = 3.0, \tau = 0.9$ |
| hopper-medium-expert | $\beta = 0.3, \tau = 0.7$ | $\beta = 0.3, \tau = 0.7$ | $\beta = 0.3, \tau = 0.7$ | $\beta = 3.0, \tau = 0.7$ | $\beta = 0.3, \tau = 0.7$ |
| hopper-expert | $\beta = 0.3, \tau = 0.7$ | $\beta = 3.0, \tau = 0.7$ | $\beta = 3.0, \tau = 0.7$ | $\beta = 0.3, \tau = 0.9$ | $\beta = 0.3, \tau = 0.7$ |
| | | | | | |
| walker2d-random | $\beta = 3.0, \tau = 0.9$ | $\beta = 0.3, \tau = 0.7$ | $\beta = 0.3, \tau = 0.7$ | $\beta = 0.3, \tau = 0.7$ | $\beta = 0.3, \tau = 0.7$ |
| walker2d-medium | $\beta = 0.3, \tau = 0.7$ | $\beta = 0.3, \tau = 0.7$ | $\beta = 3.0, \tau = 0.7$ | $\beta = 0.3, \tau = 0.7$ | $\beta = 0.3, \tau = 0.7$ |
| walker2d-medium-replay | $\beta = 3.0, \tau = 0.7$ | $\beta = 3.0, \tau = 0.7$ | $\beta = 3.0, \tau = 0.7$ | $\beta = 0.3, \tau = 0.7$ | $\beta = 3.0, \tau = 0.7$ |
| walker2d-medium-expert | $\beta = 3.0, \tau = 0.7$ | $\beta = 0.3, \tau = 0.7$ | $\beta = 0.3, \tau = 0.7$ | $\beta = 0.3, \tau = 0.7$ | $\beta = 0.3, \tau = 0.7$ |
| walker2d-expert | $\beta = 3.0, \tau = 0.7$ | $\beta = 0.3, \tau = 0.7$ | $\beta = 0.3, \tau = 0.9$ | $\beta = 0.3, \tau = 0.7$ | $\beta = 3.0, \tau = 0.7$ |
| | | | | | |
| halfcheetah-random | $\beta = 3.0, \tau = 0.9$ | $\beta = 0.3, \tau = 0.7$ | $\beta = 3.0, \tau = 0.9$ | $\beta = 0.3, \tau = 0.7$ | – |
| halfcheetah-medium | $\beta = 3.0, \tau = 0.9$ | $\beta = 0.3, \tau = 0.9$ | $\beta = 0.3, \tau = 0.7$ | $\beta = 0.3, \tau = 0.7$ | – |
| halfcheetah-medium-replay | $\beta = 3.0, \tau = 0.9$ | $\beta = 0.3, \tau = 0.7$ | $\beta = 0.3, \tau = 0.7$ | $\beta = 3.0, \tau = 0.9$ | – |
| halfcheetah-medium-expert | $\beta = 3.0, \tau = 0.7$ | $\beta = 0.3, \tau = 0.9$ | $\beta = 0.3, \tau = 0.7$ | $\beta = 3.0, \tau = 0.7$ | – |
| halfcheetah-expert | $\beta = 3.0, \tau = 0.7$ | $\beta = 3.0, \tau = 0.7$ | $\beta = 3.0, \tau = 0.7$ | $\beta = 0.3, \tau = 0.7$ | – |
| | | | | | |
| button-press | $\beta = 0.3, \tau = 0.7$ | $\beta = 0.3, \tau = 0.7$ | $\beta = 0.3, \tau = 0.7$ | $\beta = 0.3, \tau = 0.7$ | – |
| door-open | $\beta = 3.0, \tau = 0.7$ | $\beta = 3.0, \tau = 0.7$ | $\beta = 0.3, \tau = 0.7$ | $\beta = 0.3, \tau = 0.7$ | – |
| drawer-close | $\beta = 0.3, \tau = 0.7$ | $\beta = 0.3, \tau = 0.9$ | $\beta = 0.3, \tau = 0.7$ | $\beta = 0.3, \tau = 0.7$ | – |
| drawer-open | $\beta = 0.3, \tau = 0.7$ | $\beta = 0.3, \tau = 0.7$ | $\beta = 0.3, \tau = 0.7$ | $\beta = 3.0, \tau = 0.7$ | – |
| peg-insert-side | $\beta = 0.3, \tau = 0.7$ | $\beta = 3.0, \tau = 0.7$ | $\beta = 0.3, \tau = 0.7$ | $\beta = 0.3, \tau = 0.7$ | – |
| pick-place | $\beta = 0.3, \tau = 0.9$ | $\beta = 3.0, \tau = 0.7$ | $\beta = 0.3, \tau = 0.7$ | $\beta = 0.3, \tau = 0.7$ | – |
| push | $\beta = 0.3, \tau = 0.9$ | $\beta = 3.0, \tau = 0.7$ | $\beta = 0.3, \tau = 0.9$ | $\beta = 0.3, \tau = 0.9$ | – |
| reach | $\beta = 0.3, \tau = 0.7$ | $\beta = 0.3, \tau = 0.7$ | $\beta = 0.3, \tau = 0.7$ | $\beta = 0.3, \tau = 0.7$ | – |
| window-close | $\beta = 0.3, \tau = 0.9$ | $\beta = 0.3, \tau = 0.7$ | $\beta = 0.3, \tau = 0.7$ | $\beta = 0.3, \tau = 0.7$ | – |
| window-open | $\beta = 0.3, \tau = 0.9$ | $\beta = 3.0, \tau = 0.9$ | $\beta = 0.3, \tau = 0.7$ | $\beta = 0.3, \tau = 0.9$ | – |
| bin-picking | $\beta = 3.0, \tau = 0.7$ | $\beta = 0.3, \tau = 0.7$ | $\beta = 0.3, \tau = 0.7$ | $\beta = 0.3, \tau = 0.7$ | – |
| box-close | $\beta = 0.3, \tau = 0.9$ | $\beta = 0.3, \tau = 0.7$ | $\beta = 0.3, \tau = 0.7$ | $\beta = 0.3, \tau = 0.9$ | – |
| door-lock | $\beta = 0.3, \tau = 0.7$ | $\beta = 0.3, \tau = 0.7$ | $\beta = 0.3, \tau = 0.7$ | $\beta = 0.3, \tau = 0.7$ | – |
| door-unlock | $\beta = 0.3, \tau = 0.7$ | $\beta = 3.0, \tau = 0.7$ | $\beta = 0.3, \tau = 0.9$ | $\beta = 0.3, \tau = 0.7$ | – |
| hand-insert | $\beta = 0.3, \tau = 0.9$ | $\beta = 3.0, \tau = 0.7$ | $\beta = 0.3, \tau = 0.7$ | $\beta = 0.3, \tau = 0.7$ | – |

Table 11: Tuned hyperparameter values for IQL

| Dataset | Original | Zero | Random | Negative | No-terminal |
|---|---|---|---|---|---|
| hopper-random | $\alpha = 0.1$ | $\alpha = 0.1$ | $\alpha = 100.0$ | $\alpha = 0.1$ | $\alpha = 1.0$ |
| hopper-medium | $\alpha = 1.0$ | $\alpha = 10.0$ | $\alpha = 10.0$ | $\alpha = 10.0$ | $\alpha = 10.0$ |
| hopper-medium-replay | $\alpha = 10.0$ | $\alpha = 10.0$ | $\alpha = 100.0$ | $\alpha = 10.0$ | $\alpha = 10.0$ |
| hopper-medium-expert | $\alpha = 10.0$ | $\alpha = 100.0$ | $\alpha = 100.0$ | $\alpha = 10.0$ | $\alpha = 10.0$ |
| hopper-expert | $\alpha = 10.0$ | $\alpha = 1.0$ | $\alpha = 0.1$ | $\alpha = 10.0$ | $\alpha = 100.0$ |
| | | | | | |
| walker2d-random | $\alpha = 0.1$ | $\alpha = 100.0$ | $\alpha = 100.0$ | $\alpha = 100.0$ | $\alpha = 10.0$ |
| walker2d-medium | $\alpha = 10.0$ | $\alpha = 10.0$ | $\alpha = 1.0$ | $\alpha = 10.0$ | $\alpha = 10.0$ |
| walker2d-medium-replay | $\alpha = 10.0$ | $\alpha = 10.0$ | $\alpha = 1.0$ | $\alpha = 100.0$ | $\alpha = 10.0$ |
| walker2d-medium-expert | $\alpha = 10.0$ | $\alpha = 0.1$ | $\alpha = 1.0$ | $\alpha = 10.0$ | $\alpha = 100.0$ |
| walker2d-expert | $\alpha = 10.0$ | $\alpha = 10.0$ | $\alpha = 100.0$ | $\alpha = 100.0$ | $\alpha = 10.0$ |
| | | | | | |
| halfcheetah-random | $\alpha = 0.1$ | $\alpha = 0.1$ | $\alpha = 1.0$ | $\alpha = 100.0$ | – |
| halfcheetah-medium | $\alpha = 0.1$ | $\alpha = 1.0$ | $\alpha = 1.0$ | $\alpha = 100.0$ | – |
| halfcheetah-medium-replay | $\alpha = 0.1$ | $\alpha = 100.0$ | $\alpha = 100.0$ | $\alpha = 100.0$ | – |
| halfcheetah-medium-expert | $\alpha = 100.0$ | $\alpha = 1.0$ | $\alpha = 1.0$ | $\alpha = 100.0$ | – |
| halfcheetah-expert | $\alpha = 100.0$ | $\alpha = 10.0$ | $\alpha = 100.0$ | $\alpha = 100.0$ | – |
| | | | | | |
| button-press | $\alpha = 100.0$ | $\alpha = 100.0$ | $\alpha = 10.0$ | $\alpha = 100.0$ | – |
| door-open | $\alpha = 1.0$ | $\alpha = 10.0$ | $\alpha = 100.0$ | $\alpha = 10.0$ | – |
| drawer-close | $\alpha = 100.0$ | $\alpha = 10.0$ | $\alpha = 10.0$ | $\alpha = 10.0$ | – |
| drawer-open | $\alpha = 100.0$ | $\alpha = 100.0$ | $\alpha = 100.0$ | $\alpha = 10.0$ | – |
| peg-insert-side | $\alpha = 1.0$ | $\alpha = 100.0$ | $\alpha = 100.0$ | $\alpha = 10.0$ | – |
| pick-place | $\alpha = 100.0$ | $\alpha = 100.0$ | $\alpha = 100.0$ | $\alpha = 100.0$ | – |
| push | $\alpha = 100.0$ | $\alpha = 100.0$ | $\alpha = 100.0$ | $\alpha = 100.0$ | – |
| reach | $\alpha = 100.0$ | $\alpha = 100.0$ | $\alpha = 100.0$ | $\alpha = 10.0$ | – |
| window-close | $\alpha = 1.0$ | $\alpha = 0.1$ | $\alpha = 10.0$ | $\alpha = 10.0$ | – |
| window-open | $\alpha = 100.0$ | $\alpha = 100.0$ | $\alpha = 10.0$ | $\alpha = 100.0$ | – |
| bin-picking | $\alpha = 100.0$ | $\alpha = 100.0$ | $\alpha = 100.0$ | $\alpha = 100.0$ | – |
| box-close | $\alpha = 1.0$ | $\alpha = 100.0$ | $\alpha = 100.0$ | $\alpha = 100.0$ | – |
| door-lock | $\alpha = 1.0$ | $\alpha = 10.0$ | $\alpha = 10.0$ | $\alpha = 100.0$ | – |
| door-unlock | $\alpha = 100.0$ | $\alpha = 10.0$ | $\alpha = 10.0$ | $\alpha = 10.0$ | – |
| hand-insert | $\alpha = 100.0$ | $\alpha = 100.0$ | $\alpha = 100.0$ | $\alpha = 100.0$ | – |

Table 12: Tuned hyperparameter values for CQL

| Dataset | Behavior | BC | Algo | Original | Zero | Random | Negative | No-terminal |
|---|---|---|---|---|---|---|---|---|
| hopper random | 1.2 | 3.6±0.2 | ATAC | 5.6±0.8 | 22.6±3.8 | 18.4±3.8 | 30.5±1.1 | 18.8±4.5 |
| | | | PSPI | 7.2±0.3 | 12.3±3.5 | 17.3±3.9 | 23.5±3.7 | 17.1±4.4 |
| | | | IQL | 7.7±0.2 | 5.0±0.7 | 1.0±0.0 | 0.9±0.1 | 1.0±0.0 |
| | | | CQL | 3.0±0.7 | 2.0±1.1 | 1.8±0.2 | 0.7±0.0 | 1.2±0.5 |
| | | | DT | 11.5±0.3 | 11.0±0.4 | 6.5±0.1 | 12.3±0.8 | – |
| hopper medium | 44.3 | 53.3±0.1 | ATAC | 84.8±1.5 | 94.0±0.6 | 92.3±0.3 | 91.5±0.5 | 88.1±2.0 |
| | | | PSPI | 94.5±1.7 | 88.4±1.3 | 92.7±0.5 | 89.4±0.6 | 87.2±3.0 |
| | | | IQL | 57.4±0.9 | 53.1±0.4 | 59.0±2.1 | 63.0±2.1 | 43.0±7.3 |
| | | | CQL | 85.0±3.7 | 55.0±11.7 | 36.9±9.8 | 89.0±1.5 | 46.3±11.5 |
| | | | DT | 62.4±0.6 | 63.5±0.8 | 63.3±0.6 | 62.6±0.8 | – |
| hopper medium replay | 15.0 | 26.7±0.6 | ATAC | 101.5±0.2 | 65.7±3.9 | 77.2±2.9 | 48.1±2.1 | 99.4±0.6 |
| | | | PSPI | 101.4±0.3 | 32.4±1.3 | 46.9±7.2 | 31.0±0.0 | 99.1±0.4 |
| | | | IQL | 61.0±1.9 | 16.5±2.5 | 1.5±0.3 | 3.5±2.2 | 6.1±3.1 |
| | | | CQL | 79.7±3.7 | 1.9±0.1 | 7.6±3.7 | 5.8±2.7 | 4.4±2.8 |
| | | | DT | 84.0±0.9 | 60.7±2.8 | 67.8±1.1 | 68.8±2.6 | – |
| hopper medium expert | 64.8 | 53.6±0.1 | ATAC | 104.2±1.5 | 106.4±1.1 | 92.2±8.0 | 90.4±0.4 | 98.9±4.1 |
| | | | PSPI | 108.9±0.6 | 93.3±5.0 | 58.8±9.4 | 95.3±0.3 | 108.3±0.6 |
| | | | IQL | 109.2±0.5 | 52.8±0.3 | 60.5±8.7 | 94.5±3.7 | 77.9±7.6 |
| | | | CQL | 88.8±3.6 | 49.7±8.1 | 60.7±9.4 | 88.6±6.1 | 47.4±11.0 |
| | | | DT | 89.9±2.7 | 89.8±2.6 | 89.7±2.2 | 93.8±2.3 | – |
| hopper expert | 108.5 | 110.6±0.0 | ATAC | 111.1±0.1 | 110.9±0.1 | 110.7±0.2 | 110.9±0.1 | 111.4±0.0 |
| | | | PSPI | 110.9±0.2 | 111.1±0.1 | 111.1±0.1 | 111.0±0.2 | 110.9±0.3 |
| | | | IQL | 110.3±0.2 | 66.7±12.8 | 73.5±11.2 | 48.5±11.8 | 24.3±8.9 |
| | | | CQL | 101.2±3.0 | 0.7±0.0 | 0.7±0.0 | 66.0±12.5 | 0.6±0.0 |
| | | | DT | 107.2±0.7 | 107.7±0.6 | 108.5±0.3 | 109.0±0.3 | – |
| walker2d random | 0.0 | 1.4±0.0 | ATAC | 6.4±0.7 | 5.8±0.5 | 3.2±0.9 | 5.6±0.6 | 4.9±1.1 |
| | | | PSPI | 6.3±0.4 | 2.1±0.8 | 1.6±0.5 | 1.0±0.5 | 5.5±0.0 |
| | | | IQL | 7.2±0.1 | 1.3±0.1 | 1.4±0.7 | 6.7±0.0 | 1.8±1.0 |
| | | | CQL | 4.0±1.8 | 0.9±0.9 | 4.3±1.4 | -0.4±0.0 | 0.2±0.4 |
| | | | DT | 6.5±0.2 | 5.3±0.5 | 1.7±0.0 | 6.4±0.2 | – |
| walker2d medium | 62.0 | 68.1±0.5 | ATAC | 86.8±1.6 | 76.6±0.3 | 76.8±0.7 | 72.7±0.2 | 86.4±0.6 |
| | | | PSPI | 87.7±0.7 | 71.2±0.5 | 72.2±0.6 | 77.5±0.5 | 85.2±0.5 |
| | | | IQL | 78.4±1.1 | 62.8±3.0 | 72.0±2.2 | 53.3±3.3 | 39.3±0.9 |
| | | | CQL | 81.4±0.3 | 2.0±0.9 | 1.2±0.7 | 14.5±7.4 | 2.1±0.8 |
| | | | DT | 76.6±0.3 | 75.7±0.5 | 75.8±0.2 | 75.5±0.3 | – |
| walker2d medium replay | 14.8 | 19.3±0.4 | ATAC | 85.7±1.5 | 67.2±2.6 | 66.5±1.6 | 42.3±6.7 | 80.2±1.0 |
| | | | PSPI | 84.9±3.2 | 69.9±0.7 | 68.1±2.2 | 56.2±4.5 | 81.3±0.8 |
| | | | IQL | 57.7±3.6 | 15.5±2.2 | 3.2±1.2 | 0.2±0.7 | 0.7±0.1 |
| | | | CQL | 74.1±1.2 | 0.5±0.3 | -0.2±0.1 | 0.2±0.3 | 0.4±0.4 |
| | | | DT | 56.3±1.3 | 53.9±1.0 | 62.0±1.1 | 53.5±1.5 | – |
| walker2d medium expert | 82.7 | 103.0±1.0 | ATAC | 105.9±5.4 | 96.8±4.4 | 93.6±4.8 | 72.7±0.3 | 108.9±4.5 |
| | | | PSPI | 110.0±0.1 | 80.6±6.3 | 109.2±0.1 | 75.5±0.7 | 111.7±1.3 |
| | | | IQL | 111.0±0.2 | 106.2±0.4 | 71.5±7.4 | 70.7±2.9 | 28.0±0.7 |
| | | | CQL | 109.3±0.3 | 0.8±0.5 | 0.1±0.2 | 28.0±9.1 | -0.2±0.0 |
| | | | DT | 108.5±0.1 | 108.4±0.1 | 108.5±0.2 | 108.4±0.2 | – |
| walker2d expert | 107.1 | 108.3±0.0 | ATAC | 110.2±0.1 | 108.0±0.1 | 107.1±0.5 | 107.8±0.0 | 110.3±0.0 |
| | | | PSPI | 109.3±0.0 | 107.9±0.1 | 102.1±2.8 | 107.8±0.2 | 109.5±0.0 |
| | | | IQL | 110.9±0.1 | 108.4±0.0 | 103.0±3.5 | 108.3±0.1 | 111.0±0.1 |
| | | | CQL | 108.3±0.2 | 108.2±0.2 | 108.0±0.1 | 108.1±0.2 | 108.3±0.3 |
| | | | DT | 109.0±0.1 | 108.9±0.1 | 109.0±0.1 | 109.0±0.1 | – |
| halfcheetah random | -0.1 | 1.9±0.0 | ATAC | 2.2±0.0 | 2.2±0.0 | 2.2±0.0 | 2.2±0.0 | – |
| | | | PSPI | 2.6±0.4 | 2.3±0.0 | 2.2±0.0 | 2.3±0.0 | – |
| | | | IQL | 18.6±0.3 | 1.8±0.1 | 2.0±0.1 | 2.2±0.0 | – |
| | | | CQL | 30.7±0.6 | 1.0±0.8 | -1.4±0.3 | -5.4±0.3 | – |
| | | | DT | 2.3±0.0 | 2.3±0.0 | 2.3±0.0 | 2.3±0.0 | – |
| halfcheetah medium | 40.7 | 42.2±0.0 | ATAC | 51.2±0.1 | 43.1±0.1 | 43.0±0.1 | 41.4±0.1 | – |
| | | | PSPI | 49.4±0.1 | 43.0±0.1 | 42.9±0.1 | 42.8±0.1 | – |
| | | | IQL | 50.0±0.1 | 39.3±2.8 | 42.3±0.1 | 34.7±0.3 | – |
| | | | CQL | 57.2±1.5 | 43.2±0.1 | 43.5±0.1 | 42.7±0.1 | – |
| | | | DT | 43.0±0.0 | 43.0±0.0 | 43.0±0.0 | 43.0±0.0 | – |
| halfcheetah medium replay | 27.2 | 35.9±0.1 | ATAC | 47.4±0.1 | 33.7±2.2 | 31.1±2.5 | 40.3±0.2 | – |
| | | | PSPI | 48.5±0.2 | 35.6±0.6 | 36.1±0.7 | 35.5±0.5 | – |
| | | | IQL | 43.7±0.2 | 36.1±0.5 | 35.3±0.3 | 18.1±0.4 | – |
| | | | CQL | 48.2±5.4 | 34.2±0.7 | 33.1±0.9 | 32.4±1.0 | – |
| | | | DT | 38.6±0.1 | 37.2±0.2 | 37.1±0.2 | 37.0±0.2 | – |
| halfcheetah medium expert | 64.4 | 67.9±0.4 | ATAC | 91.5±0.6 | 46.4±2.0 | 44.9±1.2 | 41.0±0.1 | – |
| | | | PSPI | 86.7±1.9 | 52.6±1.7 | 49.8±1.9 | 42.2±0.1 | – |
| | | | IQL | 94.8±0.1 | 56.5±3.2 | 70.8±0.8 | 37.9±0.2 | – |
| | | | CQL | 75.2±1.0 | 81.2±2.1 | 69.6±2.8 | 42.4±0.2 | – |
| | | | DT | 78.8±0.8 | 81.1±0.7 | 79.6±1.4 | 78.3±0.9 | – |
| halfcheetah expert | 88.1 | 92.9±0.0 | ATAC | 94.0±0.4 | 92.4±0.4 | 92.4±1.2 | 93.1±0.1 | – |
| | | | PSPI | 94.6±0.1 | 92.9±0.3 | 88.0±1.3 | 93.5±0.3 | – |
| | | | IQL | 96.8±0.1 | 92.9±0.1 | 92.9±0.0 | 89.3±0.1 | – |
| | | | CQL | 92.3±0.9 | 93.9±0.4 | 93.7±0.6 | 94.0±0.2 | – |
| | | | DT | 92.6±0.1 | 92.8±0.1 | 92.6±0.1 | 92.8±0.1 | – |

Table 13: D4RL normalized scores. The mean and standard error of normalized scores are computed across 10 random seeds. For each random seed, we evaluate the final policy over 50 episodes.

| Dataset | BC | Algo | Original | Zero | Random | Negative |
|---|---|---|---|---|---|---|
| button-press | 0.98±0.00 | ATAC | 1.00±0.00 | 0.90±0.09 | 1.00±0.00 | 0.90±0.09 |
| | | PSPI | 0.86±0.10 | 0.90±0.09 | 0.84±0.10 | 0.90±0.09 |
| | | IQL | 0.98±0.01 | 0.97±0.01 | 0.94±0.02 | 0.88±0.02 |
| | | CQL | 0.88±0.07 | 0.83±0.10 | 0.84±0.09 | 0.94±0.03 |
| | | DT | 1.00±0.00 | 1.00±0.00 | 1.00±0.00 | 1.00±0.00 |
| door-open | 0.81±0.02 | ATAC | 1.00±0.00 | 1.00±0.00 | 1.00±0.00 | 0.95±0.04 |
| | | PSPI | 1.00±0.00 | 0.95±0.03 | 0.96±0.02 | 0.81±0.09 |
| | | IQL | 0.94±0.02 | 0.86±0.03 | 0.92±0.03 | 0.92±0.02 |
| | | CQL | 0.82±0.09 | 0.07±0.07 | 0.12±0.08 | 0.00±0.00 |
| | | DT | 1.00±0.00 | 1.00±0.00 | 1.00±0.00 | 1.00±0.00 |
| drawer-close | 0.99±0.00 | ATAC | 0.89±0.09 | 0.97±0.03 | 1.00±0.00 | 1.00±0.00 |
| | | PSPI | 0.77±0.11 | 0.96±0.04 | 1.00±0.00 | 1.00±0.00 |
| | | IQL | 0.98±0.01 | 0.99±0.01 | 0.99±0.00 | 0.97±0.01 |
| | | CQL | 0.91±0.04 | 0.99±0.01 | 0.96±0.02 | 0.97±0.02 |
| | | DT | 0.99±0.01 | 1.00±0.00 | 1.00±0.00 | 1.00±0.00 |
| drawer-open | 0.88±0.01 | ATAC | 0.72±0.07 | 0.35±0.09 | 0.41±0.08 | 0.69±0.09 |
| | | PSPI | 0.61±0.10 | 0.42±0.09 | 0.57±0.07 | 0.74±0.06 |
| | | IQL | 0.93±0.02 | 0.84±0.02 | 0.73±0.04 | 0.52±0.03 |
| | | CQL | 0.78±0.04 | 0.66±0.07 | 0.65±0.07 | 0.84±0.03 |
| | | DT | 1.00±0.00 | 1.00±0.00 | 1.00±0.00 | 1.00±0.00 |
| peg-insert-side | 0.58±0.01 | ATAC | 0.82±0.04 | 0.88±0.06 | 0.83±0.05 | 0.92±0.03 |
| | | PSPI | 0.75±0.09 | 0.73±0.07 | 0.92±0.03 | 0.91±0.03 |
| | | IQL | 0.60±0.03 | 0.61±0.03 | 0.34±0.02 | 0.11±0.01 |
| | | CQL | 0.08±0.02 | 0.00±0.00 | 0.02±0.01 | 0.00±0.00 |
| | | DT | 0.07±0.01 | 0.00±0.00 | 0.01±0.00 | 0.00±0.00 |
| pick-place | 0.15±0.01 | ATAC | 0.38±0.06 | 0.35±0.05 | 0.35±0.05 | 0.24±0.05 |
| | | PSPI | 0.47±0.04 | 0.16±0.05 | 0.31±0.04 | 0.14±0.04 |
| | | IQL | 0.11±0.02 | 0.16±0.02 | 0.17±0.01 | 0.10±0.01 |
| | | CQL | 0.01±0.00 | 0.06±0.02 | 0.05±0.02 | 0.07±0.03 |
| | | DT | 0.06±0.01 | 0.06±0.01 | 0.06±0.01 | 0.06±0.01 |
| push | 0.19±0.01 | ATAC | 0.48±0.09 | 0.48±0.08 | 0.34±0.04 | 0.27±0.06 |
| | | PSPI | 0.47±0.05 | 0.42±0.04 | 0.35±0.08 | 0.37±0.08 |
| | | IQL | 0.10±0.02 | 0.18±0.02 | 0.14±0.01 | 0.13±0.01 |
| | | CQL | 0.00±0.00 | 0.01±0.00 | 0.01±0.00 | 0.03±0.02 |
| | | DT | 0.04±0.01 | 0.06±0.02 | 0.04±0.01 | 0.07±0.01 |
| reach | 0.59±0.01 | ATAC | 0.86±0.03 | 0.81±0.05 | 0.83±0.03 | 0.81±0.03 |
| | | PSPI | 0.92±0.02 | 0.77±0.04 | 0.84±0.03 | 0.84±0.04 |
| | | IQL | 0.51±0.02 | 0.59±0.04 | 0.57±0.02 | 0.58±0.02 |
| | | CQL | 0.17±0.03 | 0.29±0.04 | 0.25±0.05 | 0.23±0.03 |
| | | DT | 0.67±0.03 | 0.79±0.02 | 0.81±0.02 | 0.83±0.02 |
| window-close | 1.00±0.00 | ATAC | 0.98±0.01 | 1.00±0.00 | 0.90±0.09 | 0.86±0.10 |
| | | PSPI | 0.90±0.08 | 0.99±0.01 | 0.89±0.07 | 0.92±0.06 |
| | | IQL | 0.99±0.01 | 1.00±0.00 | 0.99±0.01 | 0.99±0.00 |
| | | CQL | 1.00±0.00 | 1.00±0.00 | 0.59±0.13 | 0.55±0.13 |
| | | DT | 1.00±0.00 | 0.98±0.01 | 0.99±0.00 | 1.00±0.00 |
| window-open | 0.98±0.00 | ATAC | 0.51±0.13 | 0.74±0.10 | 1.00±0.00 | 1.00±0.00 |
| | | PSPI | 1.00±0.00 | 0.98±0.01 | 1.00±0.00 | 0.70±0.12 |
| | | IQL | 0.97±0.01 | 0.98±0.01 | 0.98±0.00 | 0.96±0.02 |
| | | CQL | 0.83±0.03 | 0.82±0.05 | 0.92±0.02 | 0.83±0.07 |
| | | DT | 0.91±0.01 | 0.91±0.01 | 0.89±0.02 | 0.90±0.01 |
| bin-picking | 0.72±0.01 | ATAC | 0.41±0.12 | 0.49±0.09 | 0.51±0.12 | 0.37±0.12 |
| | | PSPI | 0.29±0.10 | 0.41±0.08 | 0.33±0.09 | 0.30±0.11 |
| | | IQL | 0.39±0.02 | 0.59±0.07 | 0.35±0.04 | 0.31±0.04 |
| | | CQL | 0.00±0.00 | 0.02±0.02 | 0.00±0.00 | 0.00±0.00 |
| | | DT | 0.00±0.00 | 0.00±0.00 | 0.00±0.00 | 0.00±0.00 |
| box-close | 0.57±0.01 | ATAC | 0.20±0.09 | 0.50±0.13 | 0.32±0.11 | 0.53±0.12 |
| | | PSPI | 0.34±0.10 | 0.44±0.10 | 0.28±0.09 | 0.24±0.08 |
| | | IQL | 0.58±0.02 | 0.54±0.02 | 0.40±0.04 | 0.23±0.01 |
| | | CQL | 0.04±0.01 | 0.07±0.02 | 0.01±0.00 | 0.05±0.01 |
| | | DT | 0.02±0.00 | 0.03±0.01 | 0.04±0.02 | 0.05±0.03 |
| door-lock | 0.96±0.00 | ATAC | 1.00±0.00 | 0.98±0.01 | 0.70±0.14 | 0.62±0.12 |
| | | PSPI | 0.78±0.12 | 0.80±0.12 | 0.99±0.01 | 0.82±0.10 |
| | | IQL | 0.99±0.00 | 0.96±0.01 | 0.91±0.01 | 0.69±0.04 |
| | | CQL | 0.75±0.04 | 0.61±0.10 | 0.61±0.11 | 0.60±0.08 |
| | | DT | 0.81±0.02 | 0.83±0.02 | 0.80±0.02 | 0.78±0.02 |
| door-unlock | 0.86±0.00 | ATAC | 0.97±0.01 | 0.93±0.05 | 0.98±0.01 | 0.94±0.03 |
| | | PSPI | 0.95±0.04 | 0.98±0.01 | 0.88±0.09 | 0.98±0.01 |
| | | IQL | 0.87±0.02 | 0.88±0.02 | 0.86±0.02 | 0.83±0.02 |
| | | CQL | 0.87±0.03 | 0.77±0.09 | 0.89±0.04 | 0.93±0.03 |
| | | DT | 0.97±0.01 | 0.97±0.01 | 0.98±0.01 | 0.98±0.01 |
| hand-insert | 0.39±0.01 | ATAC | 0.80±0.03 | 0.36±0.08 | 0.80±0.03 | 0.61±0.04 |
| | | PSPI | 0.81±0.02 | 0.62±0.08 | 0.75±0.04 | 0.81±0.03 |
| | | IQL | 0.35±0.03 | 0.34±0.02 | 0.18±0.03 | 0.14±0.02 |
| | | CQL | 0.13±0.04 | 0.22±0.06 | 0.34±0.04 | 0.32±0.03 |
| | | DT | 0.24±0.03 | 0.21±0.02 | 0.28±0.02 | 0.26±0.02 |

Table 14: Meta-World success rate. The mean and standard error of success rate are computed across 10 random seeds. For each random seed, we evaluate the final policy over 50 episodes.

# D Technical Details of Section 3

Here we provide the details of our theory on the robustness of offline RL in Section 3.

**Theorem 1.** *(Informal) Under Assumption 1 and certain regularity assumptions, if an offline RL algorithm* $Algo$ *is set to be sufficiently pessimistic* and *the data distribution* $\mu$ *has a positive bias, for any data reward* $\tilde{r} \in \tilde{\mathcal{R}}$, *the policy* $\hat{\pi}$ *learned by* $Algo$ *from the dataset* $\tilde{\mathcal{D}}$ *has performance guarantee* $V_r^{\pi^*}(d_0) - V_r^{\hat{\pi}}(d_0) \leq O(\iota)$ *as well as safety guarantee* $(1 - \gamma) \sum_{t=0}^{\infty} \gamma^t \mathrm{Prob}\left(\exists \tau \in [0, t], s_\tau \notin supp(\mu) | \hat{\pi}\right) \leq O(\iota)$ *for a small* $\iota$ *that decreases to zero as the degree of pessimism and dataset size increase.*

Below we first provide in Appendix D.1 some background for stating the formal version Theorem 1 in Appendix D.2. Then we present the formal claims on survival instinct and conditions on positive data bias in Appendix D.3 and Appendix D.4, respectively. Finally, we prove Theorem 1 in Appendix D.5. Our discussion here will be focusing the discounted infinite-horizon step. We discuss the finite horizon variant in Appendix E.

## D.1 Definitions

First, we introduce some definitions so that we can state the formal version of Theorem 1. We define constrained MDPs.

**Definition 3** (Constrained MDP). Let $f, g : \mathcal{S} \times \mathcal{A} \rightarrow [-1, 1]$. A CMDP problem $\mathcal{C}(\mathcal{S}, \mathcal{A}, f, g, P, \gamma)$ is defined as

$$\max_{\pi} \quad V_f^\pi(d_0), \quad \text{s.t.} \quad V_g^\pi(d_0) \leq 0.$$

Let $\pi^\dagger$ denote its optimal policy. For $\delta \geq 0$, we define the set of $\delta$-*approximately optimal policies*,

$$\Pi_{f,g}^\dagger(\delta) = \{\pi : V_f^{\pi^\dagger}(d_0) - V_f^\pi(d_0) \leq \delta, V_g^\pi(d_0) \leq \delta\}$$

We will drop the subscript $f, g$ from $\Pi_{f,g}^\dagger$ when it is clear from the context. In addition, we define a *sensitivity function* to approximately optimal policies,

$$\kappa(\delta) := \max_{\pi \in \Pi^\dagger(\delta)} V_f^\pi(d_0) - V_f^{\pi^\dagger}(d_0)$$

The sensitivity function measures how much the approximately optimal policies (which can slightly violate the constraint of the CMDP) would perform better than the optimal policy of the CMDP (which satisfies the constraint). By definition, $\kappa(\delta) \in [0, \frac{2}{1-\gamma}]$ and its value can be smaller than $\frac{2}{1-\gamma}$.

By duality, a CMDP problem is related to a saddle-point problem induced by its constraint. We review the definition of saddle points and this fact below.

**Definition 4.** Given a bifunction $\mathcal{L}(x, y) : \mathcal{X} \times \mathcal{Y} \rightarrow \mathbb{R}$, we say $(x^\dagger, y^\dagger) \in \mathcal{X} \times \mathcal{Y}$ is a *saddle point* of $\mathcal{L}$, if for all $x \in \mathcal{X}$ and $y \in \mathcal{Y}$ it holds that $\mathcal{L}(x, y^\dagger) \leq \mathcal{L}(x^\dagger, y^\dagger) \leq \mathcal{L}(x^\dagger, y)$.

**Lemma 1** (Duality). *[39] Let* $\Pi$ *denote the space of policies. Consider a CMDP* $\mathcal{C} := \mathcal{C}(\mathcal{S}, \mathcal{A}, f, g, P, \gamma)$. *Define the Lagrangian* $\mathcal{L}(\pi, \lambda) := V_f^\pi(d_0) - \lambda V_g^\pi(d_0)$ *for* $\pi \in \Pi$ *and* $\lambda \geq 0$. *Let* $\pi^\dagger$ *and* $\lambda^\dagger$ *denote the optimal policy and Lagrange multiplier, i.e.,*

$$\pi^\dagger \in \arg\max_{\pi} \min_{\lambda \geq 0} \mathcal{L}(\pi, \lambda) \quad and \quad \lambda^\dagger \in \arg\min_{\lambda \geq 0} \max_{\pi} \mathcal{L}(\pi, \lambda)$$

*Then together they form a saddle point* $(\pi^\dagger, \lambda^\dagger)$ *to the Lagrangian* $\mathcal{L}$.

We also introduce a definition to characterize the degree of pessimism of offline RL algorithms. We call it an admissibility condition defined below, which measures the size of the data-consistent reward functions that the offline RL algorithm can have a small regret.

**Definition 5.** [Admissibility] For $R \geq 0$, we say an offline RL algorithm $Algo$ is $R$-*admissible* with respect to $\tilde{\mathcal{R}}$, if for any $\tilde{r} \in \tilde{\mathcal{R}}$, given $\tilde{\mathcal{D}}$, $Algo$ learns a policy $\hat{\pi}$ satisfying the following with high probability: for any policy $\pi \in \Pi$,

$$\max_{\bar{r} \in \mathcal{R}_R(\tilde{r})} V_{\bar{r}}^\pi(d_0) - V_{\bar{r}}^{\hat{\pi}}(d_0) \leq \mathcal{E}(\pi, \mu),$$

where we define a data-consistent reward class

$$\mathcal{R}_R(\tilde{r}) := \{\bar{r} : \bar{r}(s,a) = \tilde{r}(s,a), \forall (s,a) \in \mathrm{supp}(\mu) \text{ and } \bar{r}(s,a) \in [-R,1], \forall (s,a) \in \mathcal{S} \times \mathcal{A}\},$$

$\mathcal{E}(\pi,\mu)$ is some regret upper bound such that $\mathcal{E}(\pi,\mu) = o(1)$ if $\sup_{s \in \mathcal{S}, a \in \mathcal{A}} \frac{d^\pi(s,a)}{\mu(s,a)} < \infty$, and $o(1)$ denotes a term that vanishes as the dataset size becomes infinite.

In Appendix F, we provide proofs of the degrees of admissibility for various existing algorithms, including model-free algorithms ATAC [2], VI-LCB [31] PPI/PQI [13], PSPI [7], as well as model-algorithms, ARMOR [34, 40], MOPO [32], MOReL [33], and CPPO [14]. We show that their degree of admissibility increases as the algorithm becomes more pessimistic (by setting the hyperparameters).

Finally, we recall the definition of positive data bias in Section 3 for completeness, where the implicit CMDP $\mathcal{C}_\mu(\tilde{r})$ is defined in (1).

**Definition 2** (Positive Data Bias). A distribution $\mu$ is $\frac{1}{\epsilon}$-*positively biased* w.r.t. a reward class $\tilde{\mathcal{R}}$ if

$$\max_{\tilde{r} \in \tilde{\mathcal{R}}} \max_{\pi \in \Pi^\dagger_{\tilde{r}, c_\mu}(\delta)} V_r^{\pi^*}(d_0) - V_r^\pi(d_0) \le \epsilon + O(\delta) \tag{2}$$

for all $\delta \ge 0$, where $\Pi^\dagger_{\tilde{r}, c_\mu}(\delta)$ denotes the set of $\delta$-approximately optimal policy of $\mathcal{C}_\mu(\tilde{r})$.

Note that the CMDP $\mathcal{C}_\mu(\tilde{r})$ is feasible because of the assumption on concentrability (Assumption 1). That is, we know $\pi^*$ is feasible policy, as $(1-\gamma)V_{c_\mu}^{\pi^*}(d_0) = \mathbb{E}_{d^{\pi^*}}[\mathbb{1}[(s,a) \notin \mathrm{supp}(\mu)]] = 0$ is implied by $\sup_{s,a} \frac{d^{\pi^*}(s,a)}{\mu(s,a)} < \infty$.

## D.2 Formal Statement of the Main Theorem

Now we state the formal version of Theorem 1.

**Assumption 2** (Data Assumption).

1. For all $\tilde{r} \in \tilde{\mathcal{R}}$, $\tilde{r}(s,a) \in [-1,1]$ for all $s \in \mathcal{S}$ and $a \in \mathcal{A}$.

2. The data distribution $\mu$ is $\frac{1}{\epsilon}$-positively biased with respect to $\tilde{\mathcal{R}}$.

3. There is a policy $\pi$ satisfying $V_{c_\mu} \le 0$.

4. For any $\tilde{r} \in \tilde{\mathcal{R}}$, the CMDP $\mathcal{C}_\mu(\tilde{r}) := \mathcal{C}(\mathcal{S}, \mathcal{A}, \tilde{r}, c_\mu, P, \gamma)$ in (1) has a sensitivity function $\kappa(\delta) \le K$ for some $K < \infty$ and $\lim_{\delta \to 0^+} \kappa(\delta) = 0$. In addition, its solution $\pi^\dagger$ satisfies $\sup_{s,a} \frac{d^{\pi^\dagger}(s,a)}{\mu(s,a)} < \infty$.

**Theorem 2** (Main Result). *Under the data assumption in Assumption 2, consider an offline RL algorithm Algo that is sufficiently pessimistic to be $(\frac{K}{\delta} + 2)$-admissible with respect to $\tilde{\mathcal{R}}$. Then for any $\tilde{r} \in \tilde{\mathcal{R}}$, with high probability, the policy $\hat{\pi}$ learned by Algo from the dataset $\tilde{\mathcal{D}} := \{(s,a,\tilde{r},s')|(s,a) \sim \mu, \tilde{r} = \tilde{r}(s,a), s' \sim P(\cdot|s,a)\}$ has both performance guarantee*

$$V_r^{\pi^*}(d_0) - V_r^{\hat{\pi}}(d_0) \le \epsilon + O(\iota)$$

*and safety guarantee*

$$(1-\gamma)\sum_{t=0}^\infty \gamma^t \mathrm{Prob}\left(\exists \tau \in [0,t] s_\tau \notin supp(\mu)|\hat{\pi}\right) \le \iota$$

*where $\iota := \delta + \kappa(\delta) + o(1)$ and $o(1)$ denotes a term that vanishes as the dataset size becomes infinite.*

In Assumption 2, the first assumption supposes the data reward (which may be the wrong reward) is bounded; the second assumption is that the data distribution is positively biased in view of the reward class $\tilde{\mathcal{R}}$. Finally, we make a regularity assumption on the convergence of $\kappa(\delta)$ to zero and $\sup_{s,a} \frac{d^{\pi^\dagger}(s,a)}{\mu(s,a)} < \infty$ to rule out some unrealistic $\tilde{\mathcal{R}}$ and $P$ combinations. We note that the

condition on the bounded density ratio is always true for tabular problem, and a degenerate case can happen when the state-action space is, e.g., continuous. On the other hand, a degenerate case that $\kappa$ is discontinuous at $\delta = 0$ can happen due to the infinite problem horizon in the worst case; roughly speaking, this happens when it is impossible to determine the constraint violation using a finite-horizon estimator. We provide the following lemma to quantify the rate of $\kappa(\delta)$, which is proved in Appendix D.6. In Appendix E, we show this regularity condition is always true in the finite horizon setting.

**Lemma 2.** *Suppose there is $\lambda^\dagger \leq L$ with some $L < \infty$ such that $\lambda^\dagger \in \arg\min_{\lambda \geq 0} \max_\pi V_{\tilde{r}}^\pi(d_0) - \lambda V_{c_\mu}^\pi(d_0)$. Then $\kappa(\delta) \leq L\delta$.*

Under the data assumption Assumption 2, Theorem 2 shows that if the offline RL algorithm $Algo$ is set to be sufficiently pessimistic (relative to the upper bound of the sensitivity function $\kappa$, which we know is at most $\frac{1}{1-\gamma}$ by the first assumption in Assumption 2), then $Algo$ have both a performance guarantee to the true reward $r$ and the a safety guarantee that the learned policy would have a small probability of leaving the data support.

We recall from Section 3 that Theorem 2 is the effects of the survival instinct of offline RL and the positive bias in the data distribution. As we will show, the admissibility condition in Theorem 2 would ensure that the survival instinct, which is what we stated informally as Proposition 1. In addition, we will provide concrete sufficient conditions for the data distribution being positively bias, as we alluded in Section 3.3. We discuss these details below and then we prove Theorem 2 in Appendix D.5.

### D.3   Survival Instinct

We show the formal version of Proposition 1 which says that if the offline algorithm is admissible then it has a survival instinct. That is, it implicitly solves the CMDP in (1) even though the constraint is never explicitly modeled in the algorithm. In Appendix F, we prove the admissibility for various existing algorithms, including model-free algorithms ATAC [2], VI-LCB [31] PPI/PQI [13], PSPI [7], as well as model-algorithms, ARMOR [34, 40], MOPO [32], MOReL [33], and CPPO [14].

**Proposition 2.** *For $\tilde{r} \in \tilde{\mathcal{R}}$, assume the CMDP $\mathcal{C}_\mu(\tilde{r}) := \mathcal{C}(\mathcal{S}, \mathcal{A}, \tilde{r}, c_\mu, P, \gamma)$ has a sensitivity function such that $\lim_{\delta \to 0^+} \kappa(\delta) = 0$ and $\kappa(\delta) \leq K$ for some $K \leq \frac{1}{1-\gamma}$. Consider an offline RL algorithm $Algo$ that is $(\frac{K}{\delta} + 2)$-admissible with respect to $\tilde{\mathcal{R}}$. Then, with high probability, the policy $\hat{\pi}$ learned by $Algo$ with the dataset $\tilde{\mathcal{D}} := \{(s, a, \tilde{r}, s')|(s,a) \sim \mu, \tilde{r} = \tilde{r}(s,a), s' \sim P(\cdot|s,a)\}$ satisfies*

$$V_{\tilde{r}}^{\pi^\dagger}(d_0) - V_{\tilde{r}}^{\hat{\pi}}(d_0) \leq o(1)$$
$$V_{c_\mu}^{\hat{\pi}}(d_0) \leq \delta + \kappa(\delta) + o(1)$$

*where $o(1)$ denotes a term that vanishes as the dataset size becomes infinite.*

*Proof of Proposition 2.* To begin, we introduce some extra notations. Given $\mathcal{C}_\mu(\tilde{r})$, we define a relaxed CMDP problem: $\max_\pi V_{\tilde{r}}^\pi(d_0)$, s.t.$V_{c_\mu}^\pi(d_0) \leq \delta$ and its Lagrangian $\mathcal{L}_\delta(\pi, \lambda) := V_{\tilde{r}}^\pi(d_0) - \lambda(V_{c_\mu}^\pi(d_0) - \delta)$, where $\lambda \geq 0$ denotes the Lagrange multiplier. Let $\pi_\delta^\dagger$ and $\lambda_\delta^\dagger$ denote the optimal policy and Lagrange multiplier of this relaxed CMDP, and let $\pi^\dagger$ denote the optimal policy to the CMDP $\mathcal{C}_\mu(\tilde{r})$. Finally we define a shorthand $r^\dagger(s, a) := \tilde{r}(s, a) - (\lambda_\delta^\dagger + 1)c_\mu(s, a)$.

Now we bound the regret and the constraint violation of $\hat{\pi}$ in the CMDP $\mathcal{C}_\mu(\tilde{r})$. The following holds for any $\delta \geq 0$. For the regret, we can derive

$$\begin{aligned}
V_{\tilde{r}}^{\pi^\dagger}(d_0) - V_{\tilde{r}}^{\hat{\pi}}(d_0) &= V_{\tilde{r}}^{\pi^\dagger}(d_0) - (\lambda_\delta^\dagger + 1)V_{c_\mu}^{\pi^\dagger}(d_0) - V_{\tilde{r}}^{\hat{\pi}}(d_0) && (V_{c_\mu}^{\pi^\dagger}(d_0) = 0) \\
&\leq V_{\tilde{r}}^{\pi^\dagger}(d_0) - (\lambda_\delta^\dagger + 1)V_{c_\mu}^{\pi^\dagger}(d_0) - V_{\tilde{r}}^{\hat{\pi}}(d_0) + (\lambda_\delta^\dagger + 1)V_{c_\mu}^{\hat{\pi}}(d_0) && (V_{c_\mu}^{\hat{\pi}}(d_0) \geq 0) \\
&= V_{r^\dagger}^{\pi^\dagger}(d_0) - V_{r^\dagger}^{\hat{\pi}}(d_0).
\end{aligned}$$

Here we use the additivity of value function over reward, namely, for any reward functions $r_1, r_2$ and policy $\pi$, we have $V_{r_1}^\pi + V_{r_2}^\pi = V_{r_1+r_2}^\pi$. Similarly for the constraint violation, we can derive

$$
\begin{aligned}
V_{c_\mu}^{\hat\pi}(d_0) &= (V_{c_\mu}^{\hat\pi}(d_0) - \delta) + \delta \\
&= (\lambda_\delta^\dagger + 1)(V_{c_\mu}^{\hat\pi}(d_0) - \delta) - \lambda_\delta^\dagger(V_{c_\mu}^{\hat\pi}(d_0) - \delta) + \delta \\
&= -V_{\tilde r}^{\hat\pi}(d_0) + (\lambda_\delta^\dagger + 1)(V_{c_\mu}^{\hat\pi}(d_0) - \delta) + V_{\tilde r}^{\hat\pi}(d_0) - \lambda_\delta^\dagger(V_{c_\mu}^{\hat\pi}(d_0) - \delta) + \delta \\
&= \mathcal{L}_\delta(\hat\pi, \lambda_\delta^\dagger) - \mathcal{L}_\delta(\hat\pi, \lambda_\delta^\dagger + 1) + \delta \\
&\leq \mathcal{L}_\delta(\pi_\delta^\dagger, \lambda_\delta^\dagger + 1) - \mathcal{L}_\delta(\hat\pi, \lambda_\delta^\dagger + 1) + \delta \\
&= V_{r^\dagger}^{\pi_\delta^\dagger}(d_0) - V_{r^\dagger}^{\hat\pi}(d_0) + \delta \\
&\leq V_{r^\dagger}^{\pi^\dagger}(d_0) - V_{r^\dagger}^{\hat\pi}(d_0) + \delta + \kappa(\delta)
\end{aligned}
$$

where the first inequality is due to that $(\pi_\delta^\dagger, \lambda_\delta^\dagger)$ is a saddle point to $\mathcal{L}_\delta$ (see Definition 4), and the second inequality follows from

$$
\begin{aligned}
V_{r^\dagger}^{\pi_\delta^\dagger}(d_0) &= V_{\tilde r}^{\pi_\delta^\dagger}(d_0) - (\lambda_\delta^\dagger + 1)V_{c_\mu}^{\pi_\delta^\dagger}(d_0), \\
&\leq V_{\tilde r}^{\pi_\delta^\dagger}(d_0) - (\lambda_\delta^\dagger + 1)V_{c_\mu}^{\pi^\dagger}(d_0) && \text{(using } V_{c_\mu}^{\pi^\dagger} \leq V_{c_\mu}^{\pi_\delta^\dagger}) \\
&\leq V_{\tilde r}^{\pi^\dagger}(d_0) + \kappa(\delta) - (\lambda_\delta^\dagger + 1)V_{c_\mu}^{\pi^\dagger}(d_0) && \text{(Definition of } \kappa.) \\
&\leq V_{r^\dagger}^{\pi^\dagger}(d_0) + \kappa(\delta)
\end{aligned}
$$

Next we bound on $V_{r^\dagger}^{\pi^\dagger}(d_0) - V_{r^\dagger}^{\hat\pi}(d_0)$. By Lemma 5 and Assumption 2, we know $\lambda_\delta^\dagger \leq \frac{K}{\delta}$, so $r^\dagger(s,a) \in [-\frac{K}{\delta} - 2, 1]$. This and the definition of $c_\mu$ together imply that $r^\dagger \in \mathcal{R}_\mu(\tilde r, \frac{K}{\delta})$. Since the offline RL algorithm $Algo$ is $\frac{K}{\delta}$-admissible, we have $V_{r^\dagger}^{\pi^\dagger}(d_0) - V_{r^\dagger}^{\hat\pi}(d_0) \leq \mathcal{E}(\pi^\dagger, \mu)$. Finally, we use the fact that $\pi^\dagger$ by definition satisfies $\sup_{s,a} \frac{d^{\pi^\dagger}(s,a)}{\mu(s,a)} < \infty$, so $\mathcal{E}(\pi^\dagger, \mu) = o(1)$. Thus, we have proved

$$
V_{\tilde r}^{\pi^\dagger}(d_0) - V_{\tilde r}^{\hat\pi}(d_0) \leq o(1)
$$
$$
V_{c_\mu}^{\hat\pi}(d_0) \leq o(1) + \delta + \kappa(\delta)
$$

That is, $\hat\pi \in \Pi_\iota(\mathcal{C}_\mu(\tilde r))$ with $\iota = \delta + \kappa(\delta) + o(1)$.

$\square$

### D.4 Implicit Data Bias

Below we provide example conditions for data distributions to have a positive implicit bias.

**Proposition 3.** *Under Assumption 1, the followings are true.*

1. *(RL setup) Suppose for any $\tilde r$ in $\tilde{\mathcal{R}}$, there is $h : \mathcal{S} \to \mathbb{R}$ such that*

$$
|r(s,a) + \gamma \mathbb{E}_{s' \sim \mathcal{P}|s,a}[h(s')] - h(s) - \tilde r(s,a)| \leq \begin{cases} \epsilon_1, & s, a \in supp(\mu) \\ \epsilon_2, & otherwise \end{cases}
$$

   *for some $\epsilon_1, \epsilon_2 < \infty$. Then $\mu$ is $\frac{1-\gamma}{2\epsilon_1}$-positively biased to $\tilde{\mathcal{R}}$.*

2. *(IL setup) If $\mu = d^{\pi^*}$, then $\mu$ is $\infty$-positively biased with respect to $\tilde{\mathcal{R}} = \{\tilde r : \tilde r : \mathcal{S} \times \mathcal{A} \to [-1,1]\}$. If $\mu = d^{\pi^e}$ such that $V_r^{\pi^*}(d_0) - V_r^{\pi^e}(d_0) \leq \frac{\epsilon'}{1-\gamma}$ and $\pi^e$ is deterministic, then $\mu$ is $\frac{1-\gamma}{\epsilon'}$-positively biased to the previous $\tilde{\mathcal{R}}$.*

3. *(Length bias) Suppose $V_{c_\mu}^\pi(d_0) \leq \delta$ implies $V_r^*(d_0) - V_r^\pi(d_0) \leq \frac{\epsilon' + O(\delta)}{1-\gamma}$. Then $\mu$ is $O\left(\frac{1-\gamma}{\epsilon'}\right)$-positively biased with respect to $\tilde{\mathcal{R}} = \{\tilde r : \tilde r : \mathcal{S} \times \mathcal{A} \to [-1,1]\}$. Below we list some sufficient intervention conditions on data collection that leads to the length bias:*

(a) *For all $(s,a) \in supp(\mu)$, $R_{\max} - r(s,a) \leq \epsilon'$, where $R_{\max} = \sup_{s \in \mathcal{S}, a \in \mathcal{A}} r(s,a) \leq 1$.*

(b) *For all $(s,a) \in supp(\mu)$, $Q_r^*(s,a) - V_r^*(s) \leq \epsilon'$.*

The first example is the typical RL assumption, which says $\mu$ always has a positive bias to $\tilde{\mathcal{R}}$ if all rewards in $\tilde{\mathcal{R}}$ provide the same ranking of policies in the support [41]. The second assumes the data is collectd by a single behavior policy, which includes the typical IL assumption that the data is collected by an optimal policy. Finally, the third example describes a length bias. Note that $V_{c_\mu}^\pi(s,a)$ measures the probability of going out of data support after taking $a$ at $s$, which is disproportional to expected length the agent can survive. This condition assumes longer trajectories in the data to have smaller optimality gap.

This length bias condition is typically satisfied when intervention is taken in the data collection process (despite the data collection policy being suboptimal), as we saw in the motivating example in Fig. 1 and Section 3.1, and empirically in Section 4. We give two sufficient conditions on such interventions. The first one is that the data collection is stopped when the instantaneous rewards is far from the maximum rewards, and the second is that the data collection agent does not take an action that no policy (even the optimal one) can perform well in the future. The consequence of these types of interventions is that longer trajectories perform better generally. These conditions are satisfied in `hopper` and `walker2d` dataset (except for -medium-replay) in Section 4, as well as in the grid world example in Section 3.1.

The proof of Proposition 3 is based on the performance difference lemma, which can be proved by a telescoping sum decomposition.

**Lemma 3** (Performance Difference Lemma). *Consider a discounted infinite-horizon MDP $\mathcal{M} = (\mathcal{S}, \mathcal{A}, r, P, \gamma)$. For any policy $\pi$ and any $h : \mathcal{S} \to \mathbb{R}$, it holds that*

$$V_r^\pi(d_0) - h(d_0) = \frac{1}{1-\gamma} \mathbb{E}_{s,a \sim d^\pi}[r(s,a) + \gamma \mathbb{E}_{s' \sim P|s,a}[h(s')] - h(s)].$$

*In particular, this implies for any policies $\pi, \pi'$,*

$$V_r^\pi(d_0) - V_r^{\pi'}(d_0) = \frac{1}{1-\gamma} \mathbb{E}_{s,a \sim d^\pi}[Q_r^{\pi'}(s,a) - V_r^{\pi'}(s)].$$

### D.4.1 Proof of RL setup in Proposition 3

*Proof.* Given $\tilde{r} \in \tilde{\mathcal{R}}$, consider some $h : \mathcal{S} \times \mathcal{A} \to \mathbb{R}$ satisfying the assumption. Define

$$\hat{r}(s,a) = r(s,a) + \gamma \mathbb{E}_{s' \sim \mathcal{P}|s,a}[h(s)] - h(s).$$

Notice

$$
\begin{aligned}
&(1-\gamma)(V_r^{\pi^*}(d_0) - V_r^\pi(d_0)) \\
&= (1-\gamma)(V_r^{\pi^*}(d_0) - h(d_0) - V_r^\pi(d_0) + h(d_0)) \\
&= \mathbb{E}_{s,a \sim d^{\pi^*}}[r(s,a) + \gamma \mathbb{E}_{s' \sim P|s,a}[h(s')] - h(s)] - \mathbb{E}_{s,a \sim d^\pi}[r(s,a) + \gamma \mathbb{E}_{s' \sim P|s,a}[h(s')] - h(s)] \\
&= \mathbb{E}_{s,a \sim d^{\pi^*}}[\hat{r}(s,a)] - \mathbb{E}_{s,a \sim d^\pi}[\hat{r}(s,a)] \\
&= (1-\gamma)(V_{\hat{r}}^{\pi^*}(d_0) - V_{\hat{r}}^\pi(d_0))
\end{aligned}
$$

By Lemma 3, for any $\pi \in \Pi^\dagger(\delta)$, it holds

$$
\begin{aligned}
&(1-\gamma)(V_r^{\pi^*}(d_0) - V_r^\pi(d_0)) \\
&= (1-\gamma)(V_{\hat{r}}^{\pi^*}(d_0) - V_{\hat{r}}^\pi(d_0)) \\
&= \mathbb{E}_{s,a \sim d^{\pi^*}}[\hat{r}(s,a)] - \mathbb{E}_{s,a \sim d^\pi}[\hat{r}(s,a)] \\
&= \mathbb{E}_{s,a \sim d^{\pi^*}}[\tilde{r}(s,a)] - \mathbb{E}_{s,a \sim d^\pi}[\tilde{r}(s,a)] + \mathbb{E}_{s,a \sim d^{\pi^*}}[\hat{r}(s,a) - \tilde{r}(s,a)] - \mathbb{E}_{s,a \sim d^\pi}[\hat{r}(s,a) - \tilde{r}(s,a)] \\
&= (1-\gamma)(V_{\tilde{r}}^{\pi^*}(d_0) - V_{\tilde{r}}^\pi(d_0)) + \mathbb{E}_{s,a \sim d^{\pi^*}}[\hat{r}(s,a) - \tilde{r}(s,a)] - \mathbb{E}_{s,a \sim d^\pi}[\hat{r}(s,a) - \tilde{r}(s,a)]
\end{aligned}
$$

Since $\pi^*$ is assumed to be in the support of $\mu$ in Assumption 1, we have

$$\mathbb{E}_{s,a\sim d^{\pi^*}}[\hat{r}(s,a) - \tilde{r}(s,a)] - \mathbb{E}_{s,a\sim d^\pi}[\hat{r}(s,a) - \tilde{r}(s,a)]$$

$$= \mathbb{E}_{s,a\sim d^{\pi^*}}[\hat{r}(s,a) - \tilde{r}(s,a)|(s,a) \in \text{supp}(\mu)] - \mathbb{E}_{s,a\sim d^\pi}[\hat{r}(s,a) - \tilde{r}(s,a)|(s,a) \in \text{supp}(\mu)]$$

$$\quad - \mathbb{E}_{s,a\sim d^\pi}[\hat{r}(s,a) - \tilde{r}(s,a)|(s,a) \notin \text{supp}(\mu)]$$

$$\leq 2\epsilon_1 - \sup_{s,a}|\hat{r}(s,a) - \tilde{r}(s,a)| \times \mathbb{E}_{s,a\sim d^\pi}[\mathbb{1}[s,a \notin \text{supp}(\mu)]]$$

$$\leq 2\epsilon_1 + (1-\gamma)\epsilon_2 V_{c_\mu}^\pi(d_0)$$

Putting these two inequalities together, we have shown

$$(1-\gamma)(V_r^{\pi^*}(d_0) - V_r^\pi(d_0)) \leq (1-\gamma)(V_{\tilde{r}}^{\pi^*}(d_0) - V_{\tilde{r}}^\pi(d_0)) + 2\epsilon + \epsilon_2(1-\gamma)V_{c_\mu}^\pi(d_0)$$

Since $\pi \in \Pi^\dagger(\delta)$, it implies

$$(1-\gamma)(V_r^{\pi^*}(d_0) - V_r^\pi(d_0)) \leq (1-\gamma)\delta + 2\epsilon_1 + (1-\gamma)\epsilon_2\delta = 2\epsilon_1 + O(\delta)$$

Thus, the distribution is $\frac{1-\gamma}{2\epsilon_1}$-positively biased. $\qquad\square$

### D.4.2 Proof of IL setup in Proposition 3

*Proof.* Let $\mu = d^{\pi^*}$. We can write

$$(1-\gamma)(V_r^{\pi^*}(d_0) - V_r^\pi(d_0))$$

$$= \mathbb{E}_{d^\pi}[V_r^{\pi^*}(s) - Q_r^{\pi^*}(s,a)]$$

$$= \mathbb{E}_{d^\pi}[V_r^{\pi^*}(s) - Q_r^{\pi^*}(s,a)|(s,a) \in \text{supp}(\mu)] + \mathbb{E}_{d^\pi}[V_r^{\pi^*}(s) - Q_r^{\pi^*}(s,a)|(s,a) \notin \text{supp}(\mu)]$$

$$\leq \delta$$

So the distribution is $\infty$-positively biased. For the case with the suboptimal expert $\pi^e$, we can derive similarly

$$(1-\gamma)(V_r^{\pi^*}(d_0) - V_r^\pi(d_0))$$

$$= (1-\gamma)(V_r^{\pi^*}(d_0) - V_r^{\pi^e}(d_0)) + (1-\gamma)(V_r^{\pi^e}(d_0) - V_r^\pi(d_0))$$

$$\leq \epsilon' + \mathbb{E}_{d^\pi}[V_r^{\pi^e}(s) - Q_r^{\pi^e}(s,a)]$$

$$= \epsilon' + \mathbb{E}_{d^\pi}[V_r^{\pi^e}(s) - Q_r^{\pi^e}(s,a)|(s,a) \in \text{supp}(\mu)] + \mathbb{E}_{d^\pi}[V_r^{\pi^e}(s) - Q_r^{\pi^e}(s,a)|(s,a) \notin \text{supp}(\mu)]$$

$$\leq \epsilon' + \delta$$

Therefore, $\mu$ is $\frac{1-\gamma}{\epsilon'}$-positively biased. $\qquad\square$

### D.4.3 Proof of Length Bias in Proposition 3

**Item 3a**

*Proof.* Suppose $R_{\max} - r(s,a) \leq \epsilon'$ for $(s,a) \in \text{supp}(\mu)$.

$$(1-\gamma)(V_r^{\pi^*}(d_0) - V_r^\pi(d_0))$$

$$= \mathbb{E}_{d^{\pi^*}}[r(s,a)] - \mathbb{E}_{d^\pi}[r(s,a)]$$

$$= \mathbb{E}_{d^{\pi^*}}[r(s,a)] - \mathbb{E}_{d^\pi}[r(s,a)|(s,a) \in \text{supp}(\mu)] - \mathbb{E}_{d^\pi}[r(s,a)|(s,a) \notin \text{supp}(\mu)]$$

$$\leq R_{\max} - \mathbb{E}_{d^\pi}[(R_{\max} - \epsilon')|(s,a) \in \text{supp}(\mu)] - 0$$

$$= \mathbb{E}_{d^\pi}[\epsilon'|(s,a) \in \text{supp}(\mu)] + [R_{\max}|(s,a) \notin \text{supp}(\mu)]$$

$$\leq \epsilon' + (1-\gamma)R_{\max}\delta$$

$\qquad\square$

**Item 3b**

*Proof.* Suppose $s, a \in \text{supp}(\mu)$ only if $V_r^*(s) - Q_r^*(s,a) \leq \epsilon'$. Then

$$(1-\gamma)(V_r^{\pi^*}(d_0) - V_r^\pi(d_0))$$
$$= \mathbb{E}_{d^\pi}[V_r^{\pi^*}(s) - Q_r^{\pi^*}(s,a)]$$
$$= \mathbb{E}_{d^\pi}[V_r^{\pi^*}(s) - Q_r^{\pi^*}(s,a)|(s,a) \in \text{supp}(\mu)] + \mathbb{E}_{d^\pi}[V_r^{\pi^*}(s) - Q_r^{\pi^*}(s,a)|(s,a) \notin \text{supp}(\mu)]$$
$$\leq \epsilon' + \delta$$

$\square$

Note that more broadly, we can relax the above condition to that $Q_{c_\mu}^*(s,a) - V_{c_\mu}^*(s) < \alpha$ implies $V_r^*(s) - Q_r^*(s,a) \leq \beta + O(\alpha)$, for a potentially smaller $\beta$, where we recall that $V_{c_\mu}^*, Q_{c_\mu}^*$ are the optimal value functions to (minimizing) the cost $c_\mu$ and $V_r^*, Q_r^*$ are the optimal value functions to (maximizing) the reawrd $r$. We can use the lemma below to prove the degree of positive bias is $\frac{1-\gamma}{\beta}$ in this more general case. The above is a special case for $\alpha = 1$, since $c_\mu(s,a) = 1$ when $s, a$ is not in the support of $\mu$.

**Lemma 4.** *Define $\mathcal{A}_s^\alpha = \{a \in \mathcal{A} : Q_{c_\mu}^*(s,a) - V_{c_\mu}^*(s) < \alpha\}$. Suppose for all $s \in \mathcal{S}$, $a \in \mathcal{A}_s^\alpha$ satisfies $V_r^{\pi'}(s) - Q_r^{\pi'}(s,a) \leq \beta$. For $\pi$ such that $V_{c_\mu}^\pi(d_0) \leq \delta$, it holds for any $\pi'$*

$$V_r^{\pi'}(d_0) - V_r^\pi(d_0) \leq \frac{\beta + \delta/\alpha}{1 - \gamma}$$

*Proof.*

$$(1-\gamma)(V_r^{\pi'}(d_0) - V_r^\pi(d_0)) = \mathbb{E}_{d^\pi}[V_r^{\pi'}(s) - Q_r^{\pi'}(s,a)]$$
$$= \mathbb{E}_{d^\pi}[V_r^{\pi'}(s) - Q_r^{\pi'}(s,a)|a \in \mathcal{A}_s^\alpha] + \mathbb{E}_{d^\pi}[V_r^{\pi'}(s) - Q_r^{\pi'}(s,a)|a \notin \mathcal{A}_s^\alpha]$$
$$\leq \beta + \frac{1}{1-\gamma}\mathbb{E}_{d^\pi}[\mathbb{1}[a \notin \mathcal{A}_s^\alpha]]$$

On the other hand, we have

$$(1-\gamma)\delta \geq (1-\gamma)V_{c_\mu}^\pi(d_0) = (1-\gamma)(V_{c_\mu}^\pi(d_0) - V_{c_\mu}^*(d_0))$$
$$= \mathbb{E}_{d^\pi}[Q_{c_\mu}^*(s,a) - V_{c_\mu}^*(s)|a \in \mathcal{A}_s] + \mathbb{E}_{d^\pi}[Q_{c_\mu}^*(s,a) - V_{c_\mu}^*(s)|a \notin \mathcal{A}_s]$$
$$\geq \mathbb{E}_{d^\pi}[Q_{c_\mu}^*(s,a) - V_{c_\mu}^*(s)|a \notin \mathcal{A}_s]$$
$$\geq \alpha\mathbb{E}_{d^\pi}[\mathbb{1}[a \notin \mathcal{A}_s]]$$

Therefore,

$$(1-\gamma)(V_r^{\pi'}(d_0) - V_r^\pi(d_0)) \leq \beta + \frac{\delta}{\alpha}$$

$\square$

### D.5 Proof of Theorem 2

*Proof.* By Proposition 2, we have proved

$$V_{\tilde{r}}^{\pi^\dagger}(d_0) - V_{\tilde{r}}^{\hat{\pi}}(d_0) \leq o(1)$$
$$V_{c_\mu}^{\hat{\pi}}(d_0) \leq o(1) + \delta + \kappa(\delta)$$

That is, $\hat{\pi} \in \Pi_\iota(\mathcal{C}_\mu(\tilde{r}))$ with $\iota = \delta + \kappa(\delta) + o(1)$.

Then by $\frac{1}{\epsilon}$-positively assumption on $\mu$, we have the performance bound

$$V_r^{\pi^\dagger}(d_0) - V_r^{\hat{\pi}}(d_0) \leq O(\iota)$$

To bound of out of support probability, we derive

$$V_{c_\mu}^{\hat\pi}(d_0) = \sum_{t=0}^{\infty} \gamma^t \mathbb{E}_{d_t^{\hat\pi}}[\mathbb{1}[\mu(s,a)=0]]$$

$$= (1-\gamma)\sum_{t=0}^{\infty}\gamma^t\left(\sum_{\tau=0}^{t}\mathbb{E}_{d_\tau^{\hat\pi}}[\mathbb{1}[\mu(s,a)=0]]\right)$$

$$\geq (1-\gamma)\sum_{t=0}^{\infty}\gamma^t\mathrm{Prob}\left(\exists\tau\in[0,t]s_\tau\notin\mathrm{supp}(\mu)|\hat\pi\right)$$

where second equality follows [45][Lemma 2]. $\qquad\square$

## D.6   Technical Details of Constrained MDP

**Lemma 5.** *For $g : \mathcal{S}\times\mathcal{A}\to[0,\infty)$, let $\mathcal{C}(\mathcal{S},\mathcal{A},f,g,P,\gamma)$ be a CMDP with sensitivity function $\kappa(\delta)$. Assume $\mathcal{C}(\mathcal{S},\mathcal{A},f,g,P,\gamma)$ is feasible. Consider a relaxed CMDP, $\max_\pi V_f^\pi(d_0)$, s.t. $V_g^\pi(d_0)\leq\delta$ and let $\lambda_\delta^\dagger\geq 0$ denote its optimal Lagrange multiplier. Then $\lambda_\delta^\dagger\leq\frac{\kappa(\delta)}{\delta}$.*

*Proof.* Denote the optimal policy of the relaxed CMDP as $\pi_\delta^\dagger$ and the optimal policy of the original CMDP as $\pi^\dagger$. By construction, $\pi_\delta^\dagger\in\Pi^\dagger(\delta)$. Therefore, we have $V_f^{\pi_\delta^\dagger}(d_0) - V_f^{\pi^\dagger}(d_0) = \max_{\pi\in\Pi^\dagger(\delta)}V_f^\pi(d_0) - V_f^{\pi^\dagger}(d_0) = \kappa(\delta)$.

To bound $\lambda_\delta^\dagger$, we use that the fact that $(\pi_\delta^\dagger,\lambda_\delta^\dagger)$ is a saddle point to the Lagrangian of the relaxed problem $\mathcal{L}_\delta(\pi,\lambda) := V_f^\pi(d_0) - \lambda(V_g^\pi(d_0)-\delta)$. As a result, we can derive

$$V_f^{\pi^\dagger}(d_0) - \lambda_\delta^\dagger(V_g^{\pi^\dagger}(d_0)-\delta) = \mathcal{L}_\delta(\pi^\dagger,\lambda_\delta^\dagger)\leq\mathcal{L}_\delta(\pi_\delta^\dagger,\lambda_\delta^\dagger)\leq\mathcal{L}_\delta(\pi_\delta^\dagger,0) = V_f^{\pi_\delta^\dagger}(d_0).$$

Since $V_g^{\pi^\dagger}(d_0)=0$, the above inequality implies

$$\lambda_\delta^\dagger\leq\frac{V_f^{\pi_\delta^\dagger}(d_0)-V_f^{\pi^\dagger}(d_0)}{\delta} = \frac{\kappa(\delta)}{\delta}$$

$\qquad\square$

**Lemma 2.** *Suppose there is $\lambda^\dagger\leq L$ with some $L<\infty$ such that $\lambda^\dagger\in\arg\min_{\lambda\geq 0}\max_\pi V_{\tilde r}^\pi(d_0) - \lambda V_{c_\mu}^\pi(d_0)$. Then $\kappa(\delta)\leq L\delta$.*

*Proof.* First we note by[11] [47, Theorem 4.1] we know that $(\pi^\dagger,\lambda^\dagger)$ is a saddle-point to the CMDP $\mathcal{C}_\mu(\tilde r) = \mathcal{C}(\mathcal{S},\mathcal{A},\tilde r,c_\mu,P,\gamma)$ in (1). Let $f = \tilde r$ and $g = c_\mu$. Define the Lagrangian $\mathcal{L}_\delta(\pi,\delta) = V_f^\pi(d_0) - \lambda(V_g^\pi(d_0)-\delta)$ for the relaxed CMDP problem $\max_{\pi:V_g^\pi(d_0)\leq\delta}V_f^\pi(d_0)$. Consider a saddle-point $(\pi_\delta^\dagger,\lambda_\delta^\dagger)$ to this relaxed problem .

By duality, we can first derive

$$\max_{\pi\in\Pi^\dagger(\delta)}V_f^\pi(d_0) - V_f^{\pi^\dagger}(d_0) = \max_{\pi:V_g^\pi(d_0)\leq\delta}V_f^\pi(d_0) - V_f^{\pi^\dagger}(d_0) = \mathcal{L}_\delta(\pi_\delta^\dagger,\lambda_\delta^\dagger) - V_f^{\pi^\dagger}(d_0)$$

Then we upper bound $\mathcal{L}_\delta(\pi_\delta^\dagger,\lambda_\delta^\dagger)$:

$$\mathcal{L}_\delta(\pi_\delta^\dagger,\lambda_\delta^\dagger)\leq\mathcal{L}_\delta(\pi_\delta^\dagger,\lambda^\dagger)$$
$$= V_f^{\pi_\delta^\dagger}(d_0) - \lambda^\dagger(V_f^{\pi_\delta^\dagger}(d_0)-\delta)$$
$$= V_f^{\pi_\delta^\dagger}(d_0) - \lambda^\dagger V_f^{\pi_\delta^\dagger}(d_0) + \delta\lambda^\dagger$$
$$= \mathcal{L}(\pi_\delta^\dagger,\lambda^\dagger) + \delta\lambda^\dagger$$
$$\leq\mathcal{L}(\pi^\dagger,\lambda^\dagger) + \delta\lambda^\dagger$$
$$\leq\mathcal{L}(\pi^\dagger,0) + \delta\lambda^\dagger.$$

---

[11]We use this theorem because the CMDP here does not satisfy the Slater's condition.

Thus we have

$$\max_{\pi \in \Pi^\dagger(\delta)} V_f^\pi(d_0) - V_f^{\pi^\dagger}(d_0) \le \mathcal{L}(\pi^\dagger, 0) + \delta\lambda^\dagger - V_f^{\pi^\dagger}(d_0) = V_f^{\pi^\dagger}(d_0) + \delta\lambda^\dagger - V_f^{\pi^\dagger}(d_0) = \delta\lambda^\dagger$$

$\square$

# E    Finite-Horizon Version

We discuss how to interpret Theorem 2 in the finite-horizon setup.

**Assumption 2** (Data Assumption)**.**

1. For all $\tilde{r} \in \tilde{\mathcal{R}}$, $\tilde{r}(s, a) \in [-1, 1]$ for all $s \in \mathcal{S}$ and $a \in \mathcal{A}$.

2. The data distribution $\mu$ is $\frac{1}{\epsilon}$-positively biased with respect to $\tilde{\mathcal{R}}$.

3. There is a policy $\pi$ satisfying $V_{c_\mu} \le 0$.

4. For any $\tilde{r} \in \tilde{\mathcal{R}}$, the CMDP $\mathcal{C}_\mu(\tilde{r}) := \mathcal{C}(\mathcal{S}, \mathcal{A}, \tilde{r}, c_\mu, P, \gamma)$ in (1) has a sensitivity function $\kappa(\delta) \le K$ for some $K < \infty$ and $\lim_{\delta \to 0^+} \kappa(\delta) = 0$. In addition, its solution $\pi^\dagger$ satisfies $\sup_{s,a} \frac{d^{\pi^\dagger}(s,a)}{\mu(s,a)} < \infty$.

**Theorem 2** (Main Result)**.** *Under the data assumption in Assumption 2, consider an offline RL algorithm Algo that is sufficiently pessimistic to be $(\frac{K}{\delta} + 2)$-admissible with respect to $\tilde{\mathcal{R}}$. Then for any $\tilde{r} \in \tilde{\mathcal{R}}$, with high probability, the policy $\hat{\pi}$ learned by Algo from the dataset $\tilde{\mathcal{D}} := \{(s, a, \tilde{r}, s')|(s, a) \sim \mu, \tilde{r} = \tilde{r}(s, a), s' \sim P(\cdot|s, a)\}$ has both performance guarantee*

$$V_r^{\pi^*}(d_0) - V_r^{\hat{\pi}}(d_0) \le \epsilon + O(\iota)$$

*and safety guarantee*

$$(1 - \gamma) \sum_{t=0}^\infty \gamma^t \mathrm{Prob}\left(\exists \tau \in [0, t] s_\tau \notin supp(\mu)|\hat{\pi}\right) \le \iota$$

*where $\iota := \delta + \kappa(\delta) + o(1)$ and $o(1)$ denotes a term that vanishes as the dataset size becomes infinite.*

**Notation**    To translate the previous notation to the finite-horizon setting, we suppose the state $s$ contains time information and the state space is layered. That is, $\mathcal{S} = \bigcup_{0=1}^{H-1} \mathcal{S}_t$, where $H$ is the problem horizon, and $\mathcal{S}_t$ denotes the set of states at time $t$. For example, for a trajectory $s_0, s_2, \ldots, s_{H-1}$ starting from $t = 0$, we have $s_t \in \mathcal{S}_t$, for $t \in [0, H-1]$. Therefore, we can use the previous notation to model time-varying functions naturally (needed in the finite horizon), without explicitly listing the time dependency. In this section, with abuse of notation, we define the value $V_r^\pi$ of a policy $\pi$ to reward $r$ at $s \in \mathcal{S}_\tau$ as

$$V_r^\pi(s) := \mathbb{E}_{\pi, P}\left[\sum_{t=\tau}^{H-1} \tilde{r}(s_t, a_t) \mid s_\tau = s\right]$$

**MDP and CMDP Problems**    The task MDP becomes $\mathcal{M}_H = (\mathcal{S}, \mathcal{A}, r, P, H)$ and solving it means

$$\max_\pi \mathbb{E}_{\pi, P}\left[\sum_{t=0}^{H-1} r(s_t, a_t) \mid s_0 \sim d_0\right]$$

and the CMDP problem in (1) becomes

$$\max_\pi \mathbb{E}_{\pi, P}\left[\sum_{t=0}^{H-1} \tilde{r}(s_t, a_t) \mid s_0 \sim d_0\right] \quad \text{s.t.} \quad \mathbb{E}_{\pi, P}\left[\sum_{t=0}^{H-1} c_\mu(s_t, a_t) \mid s_0 \sim d_0\right] \le 0$$

### E.1 Main Theorem for Finite-horizon Problems

**Assumption 3** (Data Assumption (Finite Horizon))**.**

1. For all $\tilde{r} \in \tilde{\mathcal{R}}$, $\tilde{r}(s, a) \in [-1, 1]$ for all $s \in \mathcal{S}$ and $a \in \mathcal{A}$.

2. The data distribution $\mu$ is $\frac{1}{\epsilon}$-positively biased with respect to $\tilde{\mathcal{R}}$.

3. The solution to the CDMP $\pi^\dagger$ satisfies $\sup_{s,a} \frac{d^{\pi^\dagger}(s,a)}{\mu(s,a)} < \infty$, where we note that $d^{\pi^\dagger}$ is defined without discounts but as the average over the episode length.

**Theorem 3** (Main Result (Finite Horizon))**.** *Under the data assumption in Assumption 2, consider an offline RL algorithm $Algo$ that is sufficiently pessimistic to be $(\frac{K}{\delta} + 2)$-admissible with respect to $\tilde{\mathcal{R}}$. Then for any $\tilde{r} \in \tilde{\mathcal{R}}$, with high probability, the policy $\hat{\pi}$ learned by $Algo$ from the dataset $\tilde{\mathcal{D}} := \{(s, a, \tilde{r}, s') | (s, a) \sim \mu, \tilde{r} = \tilde{r}(s, a), s' \sim P(\cdot|s, a)\}$ has both performance guarantee*

$$V_r^{\pi^*}(d_0) - V_r^{\hat{\pi}}(d_0) \le \epsilon + O(\iota)$$

*and safety guarantee*

$$\text{Prob}\left(\exists \tau \in [0, H - 1], s_\tau \notin supp(\mu) | \hat{\pi}\right) \le \iota$$

*where $\iota := \delta + \kappa(\delta) + o(1)$ and $o(1)$ denotes a term that vanishes as the dataset size becomes infinite.*

The main difference between the infinite-horizon setup and the finite-horizon setup is that the finite-horizon setup always satisfies the regularity assumption (the third point) in Assumption 2. In addition, the safety guarantee is more explicit compared the discounted infinite-horizon counterpart. This is because $\text{Prob}\left(\exists \tau \in [0, H - 1], s_\tau \notin supp(\mu) | \hat{\pi}\right) \le V_{c_\mu}^{\hat{\pi}}(d_0)$ for the finite horizon, whereas we need an additional conversion in Appendix D.5.

We prove $\lim_{\delta \to 0+} \kappa(\delta) = 0$ is always true for the finite horizon version.

**Proposition 4.** *For the $H$-horizon constrained MDP of* (1)*, we have an optimal dual variable $\lambda^\dagger = 2H + 1$ and $\kappa(\delta) \le \min\{2H, (2H + 1)\delta\}$.*

*Proof.* Define the Lagrange reward $r^\dagger(s, a) = \tilde{r}(s, a) - \lambda^\dagger c_\mu(s, a)$. Let $\hat{\pi}^\dagger$ denote the optimal policy to the Lagrange reward. Notice that $V_{c_\mu}^\pi(d_0) \ge 1$ for any $\pi$ such that $V_{c_\mu}^\pi(d_0) > 0$. Therefore, with $\lambda^\dagger > 2H$,

$$\max_{\pi: V_{c_\mu}^\pi(d_0) > 0} V_{\tilde{r}}^\pi(d_0) - \lambda^\dagger V_{c_\mu}^\pi(d_0) \le \max_{\pi: V_{c_\mu}^\pi(d_0) > 0} V_{\tilde{r}}^\pi(d_0) - \lambda^\dagger$$

$$< -H$$

$$\le \max_{\pi: V_{c_\mu}^\pi(d_0) = 0} V_{\tilde{r}}^\pi(d_0) - \lambda^\dagger V_{c_\mu}^\pi(d_0)$$

Therefore, $\pi^\dagger$ satisfies $V_{c_\mu}^{\pi^\dagger}(d_0) = 0$. This implies

$$\max_\pi V_{\tilde{r}}^\pi(d_0) - \lambda^\dagger V_{c_\mu}^\pi(d_0) \ge \min_{\lambda \ge 0} \max_\pi V_{\tilde{r}}^\pi(d_0) - \lambda V_{c_\mu}^\pi(d_0) = V_{\tilde{r}}^{\pi^\dagger}(d_0)$$

On the other hand, it is always true that

$$\max_\pi V_{\tilde{r}}^\pi(d_0) - \lambda^\dagger V_{c_\mu}^\pi(d_0) \ge \min_{\lambda \ge 0} \max_\pi V_{\tilde{r}}^\pi(d_0) - \lambda V_{c_\mu}^\pi(d_0)$$

Therefore,

$$\lambda^\dagger \in \min_{\lambda \ge 0} \max_\pi V_{\tilde{r}}^\pi(d_0) - \lambda V_{c_\mu}^\pi(d_0)$$

Consider $\lambda^\dagger = 2H + 1$. By Lemma 2, we have $\kappa(\delta) \le (2H + 1)\delta$. In addition $\kappa(\delta) \le 2H$ by definition. $\qquad\square$

## E.2 Length Bias

For finite-horizon problems, we provide a simple sufficient condition for length bias, which assumes long trajectories obtained in data collections are near optimal.

**Proposition 5.** *Suppose the data are generated by rolling out policies starting from $t = 0$ and the initials state distribution $d_0$. Let $\xi = (s_0, a_0, s_1, \ldots, s_{T_\xi - 1}, a_{T_\xi - 1})$ denote a trajectory of length $T_\xi$. If with probability one (over randomness of $\xi$) that $T_\xi = H$ implies*

$$\sum_{t=0}^{H-1} r(s_t, a_t) \geq V^{\pi^*}(d_0) - H\epsilon'$$

*then the data distribution is $\frac{1}{H\epsilon'}$-positively biased.*

*Proof.* Consider some $\pi \in \Pi^\dagger(\delta)$. We can derive

$$V_r^{\pi^*}(d_0) - V_r^\pi(d_0) = V_r^{\pi^*}(d_0) - \mathbb{E}_{\pi,P}\left[\sum_{t=0}^{H-1} r(s_t, a_t)\right]$$

$$= \mathbb{E}_{\pi,P}\left[V_r^{\pi^*}(d_0) - \sum_{t=0}^{H-1} r(s_t, a_t) \mid \exists \tau \in [0, H-1], s_\tau \notin \mathrm{supp}(\mu)\right]$$

$$+ \mathbb{E}_{\pi,P}\left[V_r^{\pi^*}(d_0) - \sum_{t=0}^{H-1} r(s_t, a_t) \mid \forall \tau \in [0, H-1], s_\tau \in \mathrm{supp}(\mu)\right]$$

$$\leq H \times \mathrm{Prob}\left(\exists \tau \in [0, H-1], s_\tau \notin \mathrm{supp}(\mu)|\pi\right) + H\epsilon'$$

$$\leq H \times V_{c_\mu}^\pi(d_0) + H\epsilon'$$

$$\leq H\delta + H\epsilon'$$

$\square$

**Remark on the existence of full-length trajectories.** Note that we do not explicitly make an assumption on the likeliness of full-length trajectories in Proposition 5. This is because the probability of having a full-length trajectory in the data distribution is guaranteed to be strictly greater than zero given the concentrability assumption in Assumption 3. Notice that time information is part of the state in a finite horizon problem, so Assumption 3 implies that there is a non-zero probability of having full-horizon data trajectories (otherwise, the concentrability coefficient would be $\infty$). When Assumption 3 holds, statistical errors due to finite dataset size are included in the $\iota$ term Theorem 2.

## E.3 Goal-oriented Problems

Finally we make a remark on positive data bias in the goal-oriented finite-horizon setting. In the infinite-horizon setting that a goal is marked as an absorbing state, which means that the agent once entering will stay there forever. The exact instantaneous (wrong) reward obtained at this absorbing state is not relevant (it can be anything in $[-1, 1]$) since the admissibility condition has already accounted for the range of the associated Lagrange reward. Namely, the agent is set pessimistic such that it views partial transitions/trajectories obtaining a worse return than $-\frac{1}{1-\gamma}$, which is a lower bound of the return the absorbing goal state.

To make sure the finite-horizon setting has the same kind of positive data bias, one way is to virtually extend the problem's original horizon (say $H$) by (as least) one and let the goal state be the only state where the agent can stay until the last time step[12] $H$ (i.e., all the other trajectories continue maximally up to time step $H - 1$ and therefore have a length at most $H$). Then we apply the offline

---

[12]We use zero-based numbering here where 0 is the initial step and $H$ is the last step for a problem with horizon $H + 1$.

RL algorithm to this extended problem (e.g., of horizon $H + 1$). We would need to set the horizon in the previous theoretical results accordingly for this longer horizon.

The reason for this extension is to ensure that the effect of absorbing state in the infinite-horizon setting (which ensures trajectories going into the absorbing state is by definition the longest) can carry over to the finite horizon case. If we apply the agent directly to solve the $H$ step problem without such an extension, there may be other trajectories which can be as long as a goal-reaching one. As a result, there is no positive data bias.

An alternate way to create the positive data bias in the finite horizon setting is to truncate trajectories that do not reach goal at time step $H - 1$ to be no longer than $H - 1$. That is, they now time out at time step $H - 2$, whereas only goal-reaching trajectories can continue up to time step $H - 1$.

# F   Algorithm Specific Results

In this appendix, we provide a few examples of admissible offline RL algorithms. We choose algorithms to cover both model-free [2, 7, 31, 13] and model-based [34, 40, 14, 32, 33] approaches. The pessimism in these algorithms are constructed through different ways, including adversarial training [2, 7, 34, 40, 14], value bonus [31, 32] and action truncation [13, 33]. These algorithms are listed in Table 15.

|  | Adversarial Training | Value Bonus | Action Truncation |
|---|---|---|---|
| Model-free | ATAC [2] (Appendix F.1) PSPI [7] (Appendix F.2) | VI-LCB [31] (Appendix F.5) | PPI/PQI [13] (Appendix F.6) |
| Model-based | ARMOR [34, 40] (Appendix F.3) CPPO [14] (Appendix F.4) | MOPO [32] (Appendix F.7) | MOReL [33] (Appendix F.7) |

Table 15: We show that all of the offline RL algorithms above are admissible. Note that this is not a complete list of all admissible offline RL algorithms.

We recall the definition of admissibility below.

**Definition 5.** [Admissibility] For $R \geq 0$, we say an offline RL algorithm $Algo$ is $R$-*admissible* with respect to $\tilde{\mathcal{R}}$, if for any $\tilde{r} \in \tilde{\mathcal{R}}$, given $\tilde{\mathcal{D}}$, $Algo$ learns a policy $\hat{\pi}$ satisfying the following with high probability: for any policy $\pi \in \Pi$,

$$\max_{\bar{r} \in \mathcal{R}_R(\tilde{r})} V_{\bar{r}}^{\pi}(d_0) - V_{\bar{r}}^{\hat{\pi}}(d_0) \leq \mathcal{E}(\pi, \mu),$$

where we define a data-consistent reward class

$$\mathcal{R}_R(\tilde{r}) := \left\{ \bar{r} : \bar{r}(s, a) = \tilde{r}(s, a), \forall (s, a) \in \text{supp}(\mu) \text{ and } \bar{r}(s, a) \in [-R, 1], \forall (s, a) \in \mathcal{S} \times \mathcal{A} \right\},$$

$\mathcal{E}(\pi, \mu)$ is some regret upper bound such that $\mathcal{E}(\pi, \mu) = o(1)$ if $\sup_{s \in \mathcal{S}, a \in \mathcal{A}} \frac{d^{\pi}(s,a)}{\mu(s,a)} < \infty$, and $o(1)$ denotes a term that vanishes as the dataset size becomes infinite.

For the sake of clarity, we consider the tabular setting in our analysis. That is, we assume the state space $\mathcal{S}$ and the action space $\mathcal{A}$ are countable and finite. (We use $|\mathcal{S}|$ and $|\mathcal{A}|$ to denote their cardinalities). To establish admissibility, we sometimes make minor changes over the function classes in these algorithms, because some algorithms assume that the reward is known or has value bounded by $[0, 1]$. We highlight these changes in blue. We would like to clarify that the goal of our analysis is to prove that these algorithms are admissible rather than provide a tight bound for their performance. We recommend readers who are interested in performance bound to read the original papers, as their bound can be tighter than ours.

One remarkable technical details in our analysis is that there is no need of using an union bound over all rewards in $\mathcal{R}_R(\tilde{r})$ in the admissibility definition when proving these high probability statements. Instead we found that we can reuse the bound proved for a single reward directly. The main reason is that all rewards in the admissibility definition agree with each other on the support of the data distribution, and the statistical analysis of concentration only happens within this support. Outside of the support, a uniform bound based on the reward range can be used to bound the error. This will be made more clearly later in the derivations.

### F.1 ATAC

We first show that ATAC [2] is an admissible offline RL algorithm. We first introduce notations that we will use for analyzing ATAC, which will also be useful for studying PSPI in Appendix F.2.

#### F.1.1 Notations

For any $\mu, \nu \in \Delta(\mathcal{S} \times \mathcal{A})$, we denote $(\mu \setminus \nu)(s, a) := \max(\mu(s, a) - \nu(s, a), 0)$. For any $\mu \in \Delta(\mathcal{S} \times \mathcal{A})$ and any $f : \mathcal{S} \times \mathcal{A} \to \mathbb{R}$, we define $\langle \mu, f \rangle := \sum_{(s,a) \in \mathcal{S} \times \mathcal{A}} \mu(s, a) f(s, a)$. For an MDP with transition probability $P : \mathcal{S} \times \mathcal{A} \to \Delta(\mathcal{S})$, we define $(\mathcal{P}^\pi f)(s, a) := \gamma \mathbb{E}_{s' \sim P(\cdot|s,a)} f(s', \pi)$ for any $f : \mathcal{S} \times \mathcal{A} \to \mathbb{R}$ and $\pi : \mathcal{S} \times \Delta(\mathcal{A})$.

Given function class $\mathcal{F} \subseteq (\mathcal{S} \times \mathcal{A} \to \mathbb{R})$, policy $\pi : \mathcal{S} \to \Delta(\mathcal{A})$ and reward $\bar{r} : \mathcal{S} \times \mathcal{A} \to \mathbb{R}$, we introduce Bellman error transfer coefficient [7, 2] to measure the distribution shift between two probability measure $\mu$ and $\nu$.

**Definition 6** (Bellman error transfer coefficient). The Bellman error transfer coefficient between $\nu$ and $\mu$ under function class $\mathcal{F}$, policy $\pi$ and reward $\bar{r}$ is defined as,

$$\mathscr{C}(\nu; \mu, \mathcal{F}, \pi, \bar{r}) := \max_{f \in \mathcal{F}} \frac{\|f - \bar{r} - \mathcal{P}^\pi f\|_{2,\nu}^2}{\|f - \bar{r} - \mathcal{P}^\pi f\|_{2,\mu}^2}. \tag{4}$$

We note that the Bellman error transfer coefficient is a weaker notion than the density ratio, which is established in the lemma below.

**Lemma 6.** *For any distributions $\nu, \mu \in \Delta(\mathcal{S} \times \mathcal{A})$, function class $\mathcal{F} \subseteq (\mathcal{S} \times \mathcal{A} \to \mathbb{R})$, policy $\pi : \mathcal{S} \to \Delta(\mathcal{A})$ and reward $\bar{r} : \mathcal{S} \times \mathcal{A} \to \mathbb{R}$, we have*

$$\mathscr{C}(\nu; \mu, \mathcal{F}, \pi, \bar{r}) \leq \left( \sup_{(s,a) \in \mathcal{S} \times \mathcal{A}} \frac{\nu(s, a)}{\mu(s, a)} \right). \tag{5}$$

*Proof.*

$$\begin{aligned}
\|f - \bar{r} - \mathcal{P}^\pi f\|_{2,\nu}^2 &= \sum_{(s,a) \in \mathcal{S} \times \mathcal{A}} \nu(s, a) \big( f(s, a) - \bar{r}(s, a) - \mathcal{P}^\pi f(s, a) \big)^2 \\
&= \sum_{(s,a) \in \mathcal{S} \times \mathcal{A}} \mu(s, a) \frac{\nu(s, a)}{\mu(s, a)} \big( f(s, a) - \bar{r}(s, a) - \mathcal{P}^\pi f(s, a) \big)^2 \\
&\leq \sum_{(s,a) \in \mathcal{S} \times \mathcal{A}} \mu(s, a) \left( \sup_{(s,a) \in \mathcal{S} \times \mathcal{A}} \frac{\nu(s, a)}{\mu(s, a)} \right) \big( f(s, a) - \bar{r}(s, a) - \mathcal{P}^\pi f(s, a) \big)^2 \\
&= \left( \sup_{(s,a) \in \mathcal{S} \times \mathcal{A}} \frac{\nu(s, a)}{\mu(s, a)} \right) \sum_{(s,a) \in \mathcal{S} \times \mathcal{A}} \mu(s, a) \big( f(s, a) - \bar{r}(s, a) - \mathcal{P}^\pi f(s, a) \big)^2 \\
&= \left( \sup_{(s,a) \in \mathcal{S} \times \mathcal{A}} \frac{\nu(s, a)}{\mu(s, a)} \right) \|f - \bar{r} - \mathcal{P}^\pi f\|_{2,\mu}^2
\end{aligned}$$

Therefore,

$$\begin{aligned}
\mathscr{C}(\nu; \mu, \mathcal{F}, \pi, \bar{r}) &= \max_{f \in \mathcal{F}} \frac{\|f - \bar{r} - \mathcal{P}^\pi f\|_{2,\nu}^2}{\|f - \bar{r} - \mathcal{P}^\pi f\|_{2,\mu}^2} \\
&\leq \left( \sup_{(s,a) \in \mathcal{S} \times \mathcal{A}} \frac{\nu(s, a)}{\mu(s, a)} \right) \max_{f \in \mathcal{F}} \frac{\|f - \bar{r} - \mathcal{P}^\pi f\|_{2,\mu}^2}{\|f - \bar{r} - \mathcal{P}^\pi f\|_{2,\mu}^2} \\
&= \sup_{(s,a) \in \mathcal{S} \times \mathcal{A}} \frac{\nu(s, a)}{\mu(s, a)}
\end{aligned}$$

$\square$

### F.1.2 Analysis

ATAC considers a critic function class $\mathcal{F}$ and a policy class $\Pi$. Since we consider the tabular setting, we choose $\mathcal{F} : (\mathcal{S} \times \mathcal{A} \to [-V_{\max}, V_{\max}])^{13}$ with $\frac{1}{1-\gamma} \leq V_{\max} < \infty$ and $\Pi : (\mathcal{S} \to \Delta(\mathcal{A}))$. ATAC formulates offline RL as the following Stackelberge game,

$$\hat{\pi} \in \arg\max_{\pi \in \Pi} \mathcal{L}_{\tilde{\mathcal{D}}}(\pi, f^{\pi}) \tag{6}$$

$$\text{s.t.} \quad f^{\pi} \in \arg\min_{f \in \mathcal{F}} \mathcal{L}_{\tilde{\mathcal{D}}}(\pi, f) + \beta \mathcal{E}_{\tilde{\mathcal{D}}}(\pi, f),$$

with $\beta \geq 0$ being hyperparameter, and

$$\mathcal{L}_{\tilde{\mathcal{D}}}(\pi, f) \coloneqq \mathbb{E}_{\tilde{\mathcal{D}}}\big[f(s, \pi) - f(s, a)\big], \tag{7}$$

$$\mathcal{E}_{\tilde{\mathcal{D}}}(\pi) \coloneqq \mathbb{E}_{\tilde{\mathcal{D}}}\big[\big(f(s, a) - \tilde{r} - \gamma f(s', \pi)\big)^2\big] - \min_{f' \in \mathcal{F}} \mathbb{E}_{\tilde{\mathcal{D}}}\big[\big(f'(s, a) - \tilde{r} - \gamma f(s', \pi)\big)^2\big].$$

For any policy $\pi$, the critic $f^{\pi}$ provides a relative pessimistic (with respect to the behavior policy $\mu$) value estimate of the policy $\pi$. The hyperparameter $\beta$ balances pessimism, given by $\mathcal{L}_{\tilde{\mathcal{D}}}(\pi, f)$, and Bellman consistency, given by $\mathcal{E}_{\tilde{\mathcal{D}}}(\pi)$. The learned policy $\hat{\pi}$ maximizes the relative pessimistic value estimate given by the critic.

We state the performance guarantee of ATAC with respect to any comparator policy $\pi$ under the set of data-consistent reward functions $\{\bar{r} : \bar{r}(s, a) = \tilde{r}(s, a), \forall(s, a) \in \text{supp}(\mu) \text{ and } |\bar{r}(s, a)| \leq (1 - \gamma)V_{\max}, \forall(s, a) \in \mathcal{S} \times \mathcal{A}\}$ in the following proposition. We make an additional assumption on the data distribution for ATAC, which is needed in its original proof.

**Assumption 4.** For ATAC, we assume $\mu$ is the (mixture) average of state-action visitation distribution for $d_0$.

**Proposition 6.** *Fix some $\tilde{r} : \mathcal{S} \times \mathcal{A} \to [-1, 1]$. Consider $\mathcal{F} : (\mathcal{S} \times \mathcal{A}) \to [-V_{max}, V_{max}]$ with $\frac{1}{1-\gamma} \leq V_{max} < \infty$ and $\Pi : (\mathcal{S} \to \Delta(\mathcal{A}))$. Let $\hat{\pi}$ be the solution to (6) and let $\pi \in \Pi$ be any comparator policy. Let $\nu \in \Delta(s, a)$ be an arbitrary distribution. Under Assumption 4, for any $\delta \in (0, 1]$, choosing $\beta = \tilde{\Theta}\left(\frac{1}{V_{max}} \sqrt[3]{\frac{C|\tilde{\mathcal{D}}|^2}{\left(|\mathcal{S}||\mathcal{A}|\log(1/\delta)\right)^2}}\right)$, with probability $1 - \delta$, it holds that*

$$V_{\bar{r}}^{\pi}(d_0) - V_{\bar{r}}^{\hat{\pi}}(d_0) \leq \tilde{O}\left(\frac{V_{max}\big(\mathscr{C}(\nu; \mu, \mathcal{F}, \pi, \bar{r})|\mathcal{S}||\mathcal{A}|\log(1/\delta)\big)^{1/3}}{(1 - \gamma)|\tilde{\mathcal{D}}|^{1/3}}\right) + \frac{\langle d^{\pi} \setminus \nu, \bar{r} + \mathcal{P}^{\pi}f^{\pi} - f^{\pi}\rangle}{1 - \gamma} \tag{8}$$

*for all $\bar{r}$ such that $\bar{r}(s, a) = \tilde{r}(s, a), \forall(s, a) \in supp(\mu)$ and $|\bar{r}(s, a)| \leq (1-\gamma)V_{max}, \forall(s, a) \in \mathcal{S} \times \mathcal{A}$.*

*Proof.* We first show that $\mathcal{F} : (\mathcal{S} \times \mathcal{A} \to [-V_{\max}, V_{\max}])$ is both realizable and Bellman complete with repect to $\bar{r}$. For any $\bar{r}$ such that $|\bar{r}(s, a)| \leq (1-\gamma)V_{\max}$, we have $Q_{\bar{r}}^{\pi}(s, a) \in [-V_{\max}, V_{\max}]$. This means that $Q_{\bar{r}}^{\pi} \in \mathcal{F}$ for any $\pi \in \Pi$, $s \in \mathcal{S}$, $a \in \mathcal{A}$. Moreover, for any $f \in \mathcal{F}$, $|\bar{r}(s, a) + \mathcal{P}^{\pi}f(s, a)| \leq |\bar{r}(s, a)| + |\mathcal{P}^{\pi}f(s, a)| \leq (1 - \gamma)V_{\max} + \gamma V_{\max} = V_{\max}$. In other words, $(\bar{r}(s, a) + \mathcal{P}^{\pi}f(s, a)) \in \mathcal{F}$ for any $f \in \mathcal{F}$.

By construction of $\bar{r}$, we have $\bar{\mathcal{D}} \coloneqq \{(s, a, \bar{r}, s')|(s, a, s') \in \mathcal{D}, \bar{r} = \bar{r}(s, a)\} \equiv \tilde{\mathcal{D}}$. That is, solving the Stackelberg game (6) given by $\tilde{\mathcal{D}}$ is equivalent to solving the game given by $\bar{\mathcal{D}}$. The rest of the proof follows from Theorem C.12 in [55] by choosing the reward class to only contain the reward $\bar{r}$, i.e., $\mathcal{G} = \{\bar{r}\}$ (which implies $d_{\mathcal{G}} = 0$) and using $d_{\mathcal{F}, \Pi} = \tilde{O}(|\mathcal{S}||\mathcal{A}|\log(1/\delta))$.

Note that the above derivation does not need an additional union bound to cover all rewards in $\{\bar{r} : \bar{r}(s, a) = \tilde{r}(s, a), \forall(s, a) \in \text{supp}(\mu) \text{ and } |\bar{r}(s, a)| \leq (1 - \gamma)V_{\max}, \forall(s, a) \in \mathcal{S} \times \mathcal{A}\}$. This is because the concentration analysis is only taken on the support of the data distribution, where all rewards in this reward class agree, and a uniform bound based on $V_{\max}$ is used for out of support places, which again applies to all the rewards in this reward class.

$\square$

---

[13]The original theoretical statement of ATAC assumes $\mathcal{F}$ to contain only non-negative functions. However, the analysis can be extended to any $\mathcal{F}$ with bounded value, such as $[-V_{\max}, V_{\max}]$. We note that the practical implementation of ATAC using function approximators does not make assumption on $\mathcal{F}$ being non-negative.

We then show that ATAC [2] is admissible based on Proposition 6.

**Corollary 3** (ATAC is admissible). *For any $V_{max} \geq \frac{R}{1-\gamma}$, ATAC is R-admissible with respect to $\tilde{\mathcal{R}}$ for any $\tilde{\mathcal{R}} \subseteq (\mathcal{S} \times \mathcal{A} \to [-1, 1])$.*

*Proof.* Given any $\tilde{r} \in \tilde{\mathcal{R}}$, we want to show that, with high probability, $V_{\bar{r}}^{\pi}(d_0) - V_{\bar{r}}^{\hat{\pi}}(d_0) = o(1)$ for all $\bar{r} \in \mathcal{R}_R(\tilde{r})$ and all policy $\pi$ such that $\sup_{(s,a) \in \mathcal{S} \times \mathcal{A}} \frac{d^{\pi}(s,a)}{\mu(s,a)} < \infty$. For any such $\pi$, define $C_{\infty} := \sup_{(s,a) \in \mathcal{S} \times \mathcal{A}} \frac{d^{\pi}(s,a)}{\mu(s,a)}$, we have $\mathscr{C}(\nu; \mu, \mathcal{F}, \pi, \bar{r}) \leq C_{\infty}$ for any $\bar{r}$. By taking $\nu = d^{\pi}$ in Proposition 6, we have, with probability $1 - \delta$,

$$V_{\bar{r}}^{\pi}(d_0) - V_{\bar{r}}^{\hat{\pi}}(d_0) \leq \tilde{O}\left( \frac{V_{\max}\left(C_{\infty}|\mathcal{S}||\mathcal{A}|\log(1/\delta)\right)^{1/3}}{(1-\gamma)|\tilde{\mathcal{D}}|^{1/3}} \right) + \frac{\langle d^{\pi} \setminus d^{\pi}, \bar{r} + \mathcal{P}^{\pi}f^{\pi} - f^{\pi} \rangle}{1-\gamma}$$

$$= \tilde{O}\left( \frac{V_{\max}\left(C_{\infty}|\mathcal{S}||\mathcal{A}|\log(1/\delta)\right)^{1/3}}{(1-\gamma)|\tilde{\mathcal{D}}|^{1/3}} \right) = o(1)$$

for all $\bar{r}$ such that $\bar{r}(s,a) = \tilde{r}(s,a), \forall(s,a) \in \text{supp}(\mu)$ and $|\bar{r}(s,a)| \leq (1-\gamma)V_{\max}, \forall(s,a) \in \mathcal{S} \times \mathcal{A}$. Since $V_{\max} \geq \frac{R}{1-\gamma}$, we have that $V_{\bar{r}}^{\pi}(d_0) - V_{\bar{r}}^{\hat{\pi}}(d_0) = o(1)$ for all $\bar{r} \in \mathcal{R}_R(\tilde{r})$. $\qquad \square$

### F.2  PSPI

We show that PSPI [7] is also admissible. PSPI [7] is similar to ATAC. Given critic function class $\mathcal{F} : (\mathcal{S} \times \mathcal{A} \to [-V_{\max}, V_{\max}])$[14] with $\frac{1}{1-\gamma} \leq V_{\max} < \infty$ and policy class $\Pi : (\mathcal{S} \to \Delta(\mathcal{A}))$, PSPI solves the following Stackelburg game,

$$\hat{\pi} \in \arg\max_{\pi \in \Pi} (1-\gamma)f^{\pi}(d_0, \pi) \tag{9}$$

$$\text{s.t.} \quad f^{\pi} \in \arg\min_{f \in \mathcal{F}} (1-\gamma)f(d_0, \pi) + \beta\mathcal{E}_{\tilde{\mathcal{D}}}(\pi, f),$$

with $\beta \geq 0$ being hyperparameter, and

$$\mathcal{E}_{\tilde{\mathcal{D}}}(\pi) := \mathbb{E}_{\tilde{\mathcal{D}}}\left[ \left(f(s,a) - \tilde{r} - \gamma f(s', \pi)\right)^2 \right] - \min_{f' \in \mathcal{F}} \mathbb{E}_{\tilde{\mathcal{D}}}\left[ \left(f'(s,a) - \tilde{r} - \gamma f(s', \pi)\right)^2 \right].$$

In PSPI, the critic $f^{\pi}$ provides an absolute pessimistic value estimate of policy $\pi$. The hyperparameter $\beta$ trades off pessimism and Bellman-consistency. The learned policy $\hat{\pi}$ maximizes such a pessimistic value. We state the performance guarantee of the learned policy $\hat{\pi}$ with respect to any comparator policy $\pi$ under the set of data-consistent reward functions $\{\bar{r} : \bar{r}(s,a) = \tilde{r}(s,a), \forall(s,a) \in \text{supp}(\mu) \text{ and } |\bar{r}(s,a)| \leq (1-\gamma)V_{\max}, \forall(s,a) \in \mathcal{S} \times \mathcal{A}\}$ in the following proposition.

**Proposition 7** ([7, 55]). *Let $\hat{\pi}$ be the solution to (6) and let $\pi \in \Pi$ be any comparator policy. Let $\nu \in \Delta(s, a)$ be an arbitrary distribution. For any $\delta \in (0, 1]$, choosing $\beta = \tilde{\Theta}\left( \frac{1}{V_{max}} \sqrt[3]{\frac{C|\tilde{\mathcal{D}}|^2}{\left(|\mathcal{S}||\mathcal{A}|\log(1/\delta)\right)^2}} \right)$, with probability $1 - \delta$,*

$$V_{\bar{r}}^{\pi}(d_0) - V_{\bar{r}}^{\hat{\pi}}(d_0) \leq \tilde{O}\left( \frac{V_{max}\left(\mathscr{C}(\nu; \mu, \mathcal{F}, \pi, \bar{r})|\mathcal{S}||\mathcal{A}|\log(1/\delta)\right)^{1/3}}{(1-\gamma)|\tilde{\mathcal{D}}|^{1/3}} \right) + \frac{\langle d^{\pi} \setminus \nu, \bar{r} + \mathcal{P}^{\pi}f^{\pi} - f^{\pi} \rangle}{1-\gamma} \tag{10}$$

*for all $\bar{r}$ such that $\bar{r}(s,a) = \tilde{r}(s,a), \forall(s,a) \in \text{supp}(\mu)$ and $|\bar{r}(s,a)| \leq (1-\gamma)V_{max}, \forall(s,a) \in \mathcal{S} \times \mathcal{A}$.*

*Proof.* The proof is similar to that of Proposition 6. First, observe that $\mathcal{F} : (\mathcal{S} \times \mathcal{A} \to [-V_{\max}, V_{\max}])$ is both realizable and Bellman complete with respect to $\bar{r}$. The rest of the proof follows from Theorem D.1 in [55] (taking $\mathcal{G} = \{\bar{r}\}$) and Lemma 6. $\qquad \square$

---

[14]Similar to ATAC, we extend the critic function class to contain functions that can take negative values. The theoretical analysis can be generalized to this case. The practical implementation of PSPI can directly handle critics which take negative values.

The proposition above implies that PSPI [7] is admissible. We omit the proof of the corollary below as it is the same as the proof of Corollary 3.

**Corollary 4** (PSPI is admissible). *For any $V_{max} \geq \frac{R}{1-\gamma}$, PSPI is R-admissible with respect to $\tilde{\mathcal{R}}$ for any $\tilde{\mathcal{R}} \subseteq (\mathcal{S} \times \mathcal{A} \rightarrow [-1, 1])$.*

### F.3 ARMOR

We show that ARMOR [34, 40] is admissible. ARMOR is a model-based offline RL algorithm. We denote a model as $M = (\mathcal{S}, \mathcal{A}, P_M, r_M, \gamma)$, where $P_M : \mathcal{S} \times \mathcal{A} \rightarrow \Delta(\mathcal{S})$ is the model dynamics, and $r_M : \mathcal{S} \times \mathcal{A} \rightarrow [-R_{\max}, R_{\max}]^{15}$ is the reward function with $1 \leq R_{\max} < \infty$. Since we consider the tabular setting, we use a model class that contains all possible dynamics and all reward functions bounded within $[-R_{\max}, R_{\max}]$, i.e., $\mathcal{M}_{model} = \{M : P_M \in (\mathcal{S} \times \mathcal{A} \rightarrow \Delta(\mathcal{S})), r_M \in (\mathcal{S} \times \mathcal{A} \rightarrow [-R_{\max}, R_{\max}])\}$. For any reference policy $\pi_{\text{ref}} : \mathcal{S} \rightarrow \Delta(\mathcal{A})$, ARMOR solves the following two-player game,

$$\hat{\pi} = \arg\max_{\pi \in \Pi} \min_{M \in \mathcal{M}_{\tilde{\mathcal{D}}}^{\alpha}} V_M^{\pi}(d_0) - V_M^{\pi_{\text{ref}}}(d_0) \tag{11}$$

where $V_M^{\pi}(d_0) := \mathbb{E}_{\pi, P_M}[\sum_{t=0}^{\infty} \gamma^t r_M(s_t, a_t) | s_0 \sim d_0]$, and

$$\mathcal{M}_{\tilde{\mathcal{D}}}^{\alpha} := \{M \in \mathcal{M}_{model} : \mathcal{E}_{\tilde{\mathcal{D}}}(M) - \min_{M' \in \mathcal{M}_{model}} \mathcal{E}_{\tilde{\mathcal{D}}}(M') \leq \alpha\}, \tag{12}$$

with

$$\mathcal{E}_{\tilde{\mathcal{D}}}(M) := \sum_{(s,a,\tilde{r},s') \in \tilde{\mathcal{D}}} -\log P_M(s'|s,a) + (r_M(s,a) - \tilde{r})^2. \tag{13}$$

Intuitively, ARMOR constructs a relative pessimistic model (with respect to $\pi_{\text{ref}}$) $M$ which is also approximately consistent with data. The learned policy $\hat{\pi}$ maximizes the value estimate given by the pessimistic model. We define the generalized single policy concentrability to characterize the distribution shift from $\mu$ to the state-action visitation $d^{\pi}$ of any policy $\pi$.

**Definition 7** (Generalized Single-policy Concentrability [40]). We define the generalized single-policy concentration for policy $\pi$, model class $\mathcal{M}_{model}$, reward $\bar{r}$ and data distribution $\mu$ as

$$\mathscr{C}_{model}(\pi; \mathcal{M}_{model}, \bar{r}) := \sup_{M \in \mathcal{M}_{model}} \frac{\mathbb{E}_{d^{\pi}}[\mathcal{E}(M; \bar{r})]}{\mathbb{E}_{\mu}[\mathcal{E}(M; \bar{r})]}$$

with $\mathcal{E}(M; \bar{r}) = D_{TV}(P_M(\cdot|s,a), P(\cdot|s,a))^2 + (r_M(s,a) - \bar{r}(s,a))^2$.

The single-policy concentrability $\mathscr{C}_{model}(\pi)$ can be considered as a model-based version of the Bellman error transfer coefficient in Definition 6. Following the same steps as the proof of Lemma 6, it can be shown that $\mathscr{C}_{model}(\pi; \mathcal{M}_{model}, \bar{r}) \leq \sup_{(s,a) \in \mathcal{S} \times \mathcal{A}} \frac{d^{\pi}(s,a)}{\mu(s,a)}$. We provide the high-probability performance guarantee for the learned policy $\hat{\pi}$. This implies that ARMOR is an admissible algorithm with a sufficiently large $R_{\max}$.

**Proposition 8.** *For any $\delta \in (0, 1]$, there exists an absolute constant $c$ such that when choosing $\alpha = c|\mathcal{S}|^2|\mathcal{A}| \log(1/\delta)$, for any comparator policy $\pi \in \Pi$, with probability $1 - \delta$, the policy $\hat{\pi}$ learned by ARMOR (11) satisfies*

$$V_{\bar{r}}^{\pi}(d_0) - V_{\bar{r}}^{\hat{\pi}}(d_0) \leq O\left(\sqrt{\mathscr{C}_{model}(\pi; \mathcal{M}_{model}, \bar{r})} \frac{R_{max}}{(1-\gamma)^2} \sqrt{\frac{|\mathcal{S}|^2|\mathcal{A}| \log(1/\delta)}{|\tilde{\mathcal{D}}|}}\right), \tag{14}$$

*for all $\bar{r}$ such that $\bar{r}(s,a) = \tilde{r}(s,a), \forall (s,a) \in supp(\mu)$ and $|\bar{r}(s,a)| \leq R_{max}, \forall (s,a) \in \mathcal{S} \times \mathcal{A}$.*

*Proof.* By construction, we have $\bar{M} = (\mathcal{S}, \mathcal{A}, P, \bar{r}, \gamma) \in \mathcal{M}_{model}$. Since $\bar{r}(s,a) = \tilde{r}(s,a)$ for any $(s,a) \in supp(\mu)$, we have $\mathcal{M}_{\tilde{\mathcal{D}}}^{\alpha} \equiv \mathcal{M}_{\tilde{\mathcal{D}}}^{\alpha}$. The rest of the proof follows mostly from Theorem 2 in [40] by taking $\pi_{\text{ref}} = \mu$. By a similar argument as in the proof of Proposition 6, we do not need an additional union bound to cover all rewards in $\{\bar{r} : \bar{r}(s,a) = \tilde{r}(s,a), \forall (s,a) \in supp(\mu)$ and $|\bar{r}(s,a)| \leq R_{\max}, \forall (s,a) \in \mathcal{S} \times \mathcal{A}\}$. $\square$

---

[15]The theoretical statement of ARMOR assumes the value of $r_M$ is bounded by $[0, 1]$. The original analysis can be extended as long as the value of $r_M$ is bounded by a finite value $R_{\max} < \infty$. The practical implementation of ARMOR does not assume the reward only takes value in $[0, 1]$.

**Corollary 5** (ARMOR is admissible). *For any $R_{max} \geq R$, ARMOR is $R$-admissible with respect to $\tilde{\mathcal{R}}$ for any $\tilde{\mathcal{R}} \subseteq (\mathcal{S} \times \mathcal{A} \rightarrow [-1, 1])$.*

*Proof.* Given any $\tilde{r} \in \tilde{\mathcal{R}}$, we want to show that, with high probability, $V_{\bar{r}}^{\pi}(d_0) - V_{\bar{r}}^{\hat{\pi}}(d_0) = o(1)$ for all $\bar{r} \in \mathcal{R}_R(\tilde{r})$ and all policy $\pi$ such that $\sup_{(s,a) \in \mathcal{S} \times \mathcal{A}} \frac{d^{\pi}(s,a)}{\mu(s,a)} < \infty$. For any such $\pi$, $\mathscr{C}_{model}(\pi; \mathcal{M}_{model}, \bar{r}) \leq \sup_{(s,a) \in \mathcal{S} \times \mathcal{A}} \frac{d^{\pi}(s,a)}{\mu(s,a)} := C_{\infty} < \infty$. By Proposition 8, with probability $1 - \delta$, we have

$$V_{\bar{r}}^{\pi}(d_0) - V_{\bar{r}}^{\hat{\pi}}(d_0) \leq O\left(\sqrt{C_{\infty}} \frac{R_{\max}}{(1-\gamma)^2} \sqrt{\frac{|\mathcal{S}|^2 |\mathcal{A}| \log(1/\delta)}{|\tilde{\mathcal{D}}|}}\right) = o(1). \tag{15}$$

for all $\bar{r}$ such that $\bar{r}(s, a) = \tilde{r}(s, a), \forall(s, a) \in \text{supp}(\mu)$ and $|\bar{r}(s, a)| \leq R, \forall(s, a) \in \mathcal{S} \times \mathcal{A}$. Therefore, $V_{\bar{r}}^{\pi}(d_0) - V_{\bar{r}}^{\hat{\pi}}(d_0) = o(1)$ for all $\bar{r} \in \mathcal{R}_R(\tilde{r})$ with high probability. $\square$

### F.4 CPPO

CPPO[14] is another admissible model-based offline algorithm. Again, we denote a model as $M = (\mathcal{S}, \mathcal{A}, P_M, r_M, \gamma)$, where $P_M : \mathcal{S} \times \mathcal{A} \rightarrow \Delta(\mathcal{S})$ is the model dynamics, and $r_M : \mathcal{S} \times \mathcal{A} \rightarrow [-R_{\max}, R_{\max}]^{16}$ is the reward function with $1 \leq R_{\max} < \infty$. CPPO solves the two-player game,

$$\hat{\pi} = \arg\max_{\pi \in \Pi} \min_{M \in \mathcal{M}_{\tilde{\mathcal{D}}}^{\alpha}} V_M^{\pi}(d_0) \tag{16}$$

with $V_M^{\pi}(d_0) := \mathbb{E}_{\pi, P_M}[\sum_{t=0}^{\infty} \gamma^t r_M(s_t, a_t)|s_0 \sim d_0]$ and $\mathcal{M}_{\tilde{\mathcal{D}}}^{\alpha}$ defined in (12). CPPO constructs a pessimistic model $M$ which is also approximately consistent with data. The learned policy $\hat{\pi}$ maximizes the value estimate given by the pessimistic model. We provide the high-probability performance guarantee for the learned policy $\hat{\pi}$. This implies that CPPO is an admissible algorithm with sufficiently large $R_{\max}$.

**Proposition 9.** *For any $\delta \in (0, 1]$, there exists an absolute constant $c$ such that when choosing $\alpha = c|\mathcal{S}|^2 |\mathcal{A}| \log(1/\delta)$, for any comparator policy $\pi \in \Pi$, with probability $1 - \delta$, the policy $\hat{\pi}$ learned by CPPO (16) satisfies*

$$V_{\bar{r}}^{\pi}(d_0) - V_{\bar{r}}^{\hat{\pi}}(d_0) \leq O\left(\sqrt{\mathscr{C}_{model}(\pi; \mathcal{M}_{model}, \bar{r})} \frac{R_{max}}{(1-\gamma)^2} \sqrt{\frac{|\mathcal{S}|^2 |\mathcal{A}| \log(1/\delta)}{|\tilde{\mathcal{D}}|}}\right), \tag{17}$$

*for all $\bar{r}$ such that $\bar{r}(s, a) = \tilde{r}(s, a), \forall(s, a) \in supp(\mu)$ and $|\bar{r}(s, a)| \leq R_{max}, \forall(s, a) \in \mathcal{S} \times \mathcal{A}$.*

*Proof.* By construction, we have $\bar{M} = (\mathcal{S}, \mathcal{A}, P, \bar{r}, \gamma) \in \mathcal{M}_{model}$. Since $\bar{r}(s, a) = \tilde{r}(s, a)$ for any $(s, a) \in \text{supp}(\mu)$, we have $\mathcal{M}_{\tilde{\mathcal{D}}}^{\alpha} \equiv \mathcal{M}_{\tilde{\mathcal{D}}}^{\alpha}$. The rest of the proof follows similarly from the proof of Theorem 2 in [40]. By Lemma 5 from [40], with high probability, $\bar{M} \in \mathcal{M}_{\tilde{\mathcal{D}}}^{\alpha}$. Therefore,

$$V_{\bar{r}}^{\pi}(d_0) - V_{\bar{r}}^{\hat{\pi}}(d_0) = V_{\bar{M}}^{\pi}(d_0) - V_{\bar{M}}^{\hat{\pi}}(d_0) \leq V_{\bar{M}}^{\pi}(d_0) - \min_{M \in \mathcal{M}_{\tilde{\mathcal{D}}}^{\alpha}} \left(V_M^{\hat{\pi}}(d_0)\right)$$

$$\leq V_{\bar{M}}^{\pi}(d_0) - \min_{M \in \mathcal{M}_{\tilde{\mathcal{D}}}^{\alpha}} \left(V_M^{\pi}(d_0)\right)$$

$$\leq \max_{M \in \mathcal{M}_{\tilde{\mathcal{D}}}^{\alpha}} |V_{\bar{M}}^{\pi}(d_0) - V_M^{\pi}(d_0)|$$

where the second inequality follows from the optimality of $\hat{\pi}$. By (20) from [40], for any $M \in \mathcal{M}_{\tilde{\mathcal{D}}}^{\alpha}$, $|V_{\bar{M}}^{\pi}(d_0) - V_M^{\pi}(d_0)| \leq O\left(\sqrt{\mathscr{C}_{model}(\pi; \mathcal{M}_{model}, \bar{r})} \frac{R_{\max}}{(1-\gamma)^2} \sqrt{\frac{|\mathcal{S}|^2 |\mathcal{A}| \log(1/\delta)}{|\tilde{\mathcal{D}}|}}\right)$. Following a similar argument to the proof of Proposition 6, there is no need to take the union bound with respect to $\bar{r}$, which concludes the proof. $\square$

We omit the proof for the following corollary as it is the same of the proof of Corollary 5.

**Corollary 6** (CPPO is admissible). *For any $R_{max} \geq R$, CPPO is $R$-admissible with respect to $\tilde{\mathcal{R}}$ for any $\tilde{\mathcal{R}} \subseteq (\mathcal{S} \times \mathcal{A} \rightarrow [-1, 1])$.*

---

**Algorithm 1** VI-LCB [31]

1: **Input:** Batch dataset $\tilde{\mathcal{D}}$, discount factor $\gamma$, confidence level $\delta$, and value range $V_{\max}$.
2: Set $J := \frac{\log N}{1-\gamma}$.
3: Randomly split $\tilde{\mathcal{D}}$ into $J+1$ sets $\tilde{\mathcal{D}}_t = \{(s_i, a_i, r_i, s_i')\}_{i=1}^m$ for $j \in \{0, 1, \cdots, J\}$ with $m = \frac{N}{J+1}$.

4: Set $m_0(s, a) := \sum_{i=1}^m \mathbb{1}\{(s_i, a_i) = (s, a)\}$ based on dataset $\tilde{\mathcal{D}}_0$.
5: For all $a \in \mathcal{A}$ and $s \in \mathcal{S}$, initialize $Q_0(s, a) = -V_{\max}$, $V_0(s) = -V_{\max}$ and set $\pi_0(s) = \arg\max_a m_0(s, a)$.
6: **for** $j = 1, \cdots, J$ **do**
7:     Initialize $r_t(s, a) = 0$ and set $P_{s,a}^j$ to be a random probability vector.
8:     Set $m_t(s, a) := \sum_{i=1}^m \mathbb{1}\{(s_i, a_i) = (s, a)\}$ based on dataset $\tilde{\mathcal{D}}_t$.
9:     Compute penalty $b_t(s, a)$ for $L = 2000 \log(2(J+1)|\mathcal{S}||\mathcal{A}|/\delta)$

$$b_t(s, a) := 2V_{\max}\sqrt{\frac{L}{\max(m_t(s,a), 1)}}.$$

10:     **for** $(s, a) \in (\mathcal{S}, \mathcal{A})$ **do**
11:         **if** $m_t(s, a) \geq 1$ **then**
12:             Set $P_{s,a}^j$ to be empirical transitions and $r_t(s, a)$ be empirical average of rewards.
13:         **end if**
14:         Set $Q_j(s, a) \leftarrow r_t(s, a) - b_t(s, a) + \gamma P_{s,a}^j V_{j-1}$.
15:     **end for**
16:     Compute $V_t^{mid}(s) \leftarrow \max_a Q_t(s, a)$ and $\pi_t^{mid}(s) \in \arg\max_a Q_t(s, a)$.
17:     **for** $s \in \mathcal{S}$ **do**
18:         **if** $V_t^{mid}(s) \leq V_{j-1}(s)$ **then**
19:             $V_t(s) \leftarrow V_{j-1}(s)$ and $\pi_t(s) \leftarrow \pi_{j-1}(s)$.
20:         **else**
21:             $V_t(s) \leftarrow V_t^{mid}(s)$ and $\pi_t(s) \leftarrow \pi_t^{mid}(s)$.
22:         **end if**
23:     **end for**
24: **end for**
25: **Return** $\hat{\pi} = \pi_J$

---

## F.5   VI-LCB

We show VI-LCB [31] is admissible. VI-LCB is a pessimistic version of value iteration. It adds negative bonuses to the Bellman backup step in order to underestimate the value when there are missing data. By being pessimistic in value estimation, it can overcome the issue of $\mu$ being non-exploratory. We recap the VI-LCB algorithm in Algorithm 1. We highlight the changes we make in blue. These changes are due to that originally the authors in [31] assume the rewards are non-negative, but the rewards in the admissibility definition Definition 5 can take negative values. Therefore, we make these changes accordingly. We state and prove the guarantee below.[17]

**Proposition 10.** *For $V_{\max} \geq \frac{R}{1-\gamma}$, VI-LCB is R-admissible with respect to any $\tilde{\mathcal{R}} \subseteq (\mathcal{S} \times \mathcal{A} \rightarrow [-1, 1])$.*

*Proof.* First, we introduce a factor of 2 in their proof of Lemma 1 [31] to accomodate that the rewards here can be negative. This is reflected in the updated bonus definition in Algorithm 1. This updated Lemma 1 of [31] now captures the good event that the bonus can upper bound the error of the empirical estimate of Bellman backup. It shows this good event is true with high probability.

We remark that while originally Lemma 1 of [31] is proved for a single reward function, it actually holds for simultaneously for all the reward functions in $\mathcal{R}_R(\tilde{r})$, without the need of introducing

---

[16]The original algorithm assumes that the reward function is known. Here we consider the generalization of the algorithm with unknown reward.

[17]We referenced to lemmas and equations based on the version arXiv:2103.12021.

---

**Algorithm 2** Pessimistic Policy Iteration (PPI) [13]

---

1: **Input:** Batch dataset $\tilde{\mathcal{D}}$, function class $\mathcal{F}$, probability estimator $\hat{\mu}$, hyperparameter $b$.
2: **for** $i \in \{0, \ldots, I-1\}$ **do**
3:     **for** $j \in \{0, \ldots, J\}$ **do**
4:         $f_{i,j+1} \leftarrow \arg\min_{f \in \mathcal{F}} \mathcal{L}_{\tilde{D}}(f, f_{i,j}, \hat{\pi}_i)$
5:     **end for**
6:     $\hat{\pi}_{i+1} \leftarrow \arg\max_\pi \mathbb{E}_{\tilde{\mathcal{D}}}[\mathbb{E}_\pi[\zeta \circ f_{i,J+1}]]$
7: **end for**
8: **Return** $\hat{\pi} = \hat{\pi}_I$

---

**Algorithm 3** Pessimistic Q Iteration (PQI) [13]

---

1: **Input:** Batch dataset $\tilde{\mathcal{D}}$, function class $\mathcal{F}$, probability estimator $\hat{\mu}$, hyperparameter $b$.
2: **for** $i \in \{0, \ldots, I-1\}$ **do**
3:     $f_{i+1} \leftarrow \arg\min_{f \in \mathcal{F}} \mathcal{L}_{\tilde{D}}(f, f_i)$
4:     $\hat{\pi}_{i+1}(s) \leftarrow \arg\max_a \zeta \circ f_{i+1}(s, a)$
5: **end for**
6: **Return** $\hat{\pi} = \hat{\pi}_I$

---

additionally a union bound. The reason is that in the proof of Lemma 1 in [31], the concentration is used only for the estimating the Bellman on the data support. This bound would apply to all rewards in $\mathcal{R}_R(\tilde{r})$ since they agree exactly on the data. For state-action pairs out of the support, the proof takes a uniform bound based on the size of $V_{\max}$, which also applies to all rewards in $\mathcal{R}_R(\tilde{r})$.

Therefore, we can apply the updated Lemma 1 of [31] to prove the desired high probability bound needed in the admissibility condition. Under this good event, we can use the upper bound in (54b) in [31] to bound the regret. Consider some $\pi$ such that $C := \sup_{s,a} \frac{d^{\pi(s,a)}}{\mu(s,a)} < \infty$. Suppose $\tilde{D}$ has $N$ transitions. Take some $V_{\max} \geq \frac{R}{1-\gamma}$. Then running VI-LCB with $J$ iterations ensures[18] for any $\bar{r} \in \mathcal{R}_R(\tilde{r})$,

$$V_{\bar{r}}^\pi(d_0) - V_{\bar{r}}^{\hat{\pi}}(d_0) \leq \gamma^J 2V_{\max} + \frac{64V_{\max}}{1-\gamma}\sqrt{\frac{L|\mathcal{S}||\mathcal{A}|C(J+1)}{N}}$$

where we mark the changes due to using rewards which can takes negative values in blue. Setting $J = \frac{\log N}{1-\gamma}$ as in [31], with probability $1 - \delta$, it holds that

$$V_{\bar{r}}^\pi(d_0) - V_{\bar{r}}^{\hat{\pi}}(d_0) \leq \tilde{O}\left(\gamma^J V_{\max} + \frac{V_{\max}}{1-\gamma}\sqrt{\frac{L|\mathcal{S}||\mathcal{A}|CJ}{N}}\right)$$

$$= \tilde{O}\left(V_{\max}\sqrt{\frac{|\mathcal{S}||\mathcal{A}|C\log\frac{1}{\delta}}{N(1-\gamma)^3}}\right)$$

Thus, VI-LCB is $R$-admissible. □

### F.6 PPI and PQI

We show PPI and PQI proposed in [13] are admissible. We present their algorithms in Algorithm 2 and Algorithm 3, with $(\zeta \circ f)(s,a) := \zeta(s,a)f(s,a)$ for any $\zeta, f : \mathcal{S} \times \mathcal{A} \to \mathbb{R}$. These algorithms use a probability estimator of the data distribution (denoted as $\hat{\mu}$), which in the tabular case is the empirical distribution. Based on $\hat{\mu}$, they define a filter function

$$\zeta(s, a; \hat{\mu}, b) = \mathbb{1}[\hat{\mu}(s, a) \geq b]$$

This filter function classifies whether a state-action pair is in the support, and it is used to modify the Bellman operators in dynamics programming as shown in line 6 of Algorithm 2 and line 4 of

---

[18]We add back an extra dependency on $|\mathcal{A}|$ as their original proof assumes $\pi$ is deterinistic, which is not necessarily the case here.

Algorithm 3. As a result, the Bellman backup and the policy optimization only consider in-support actions, which mitigates the issue of learning with non-exploratory $\mu$. The loss functions (i.e., $\mathcal{L}_{\widehat{\mathcal{D}}}$) in Algorithm 2 and Algorithm 3 denote the squared Bellman error (with a target network) as used in fitted Q iteration [78]. We omit the details here.

In summary, PPI and PQI follow the typical policy iteration and value iteration schemes, except that the backup and the argmax are only taken within the observed actions (or actions with sufficiently evidence to be in the support when using function approximators). This idea is similar to the spirit of the later IQL algorithm [8], except IQL doesn't not construct the filter function explicitly but relies instead on expectile maximization.

**Proposition 11.** *Suppose $\mathcal{F} = (\mathcal{S} \times \mathcal{A} \to [-V_{\max}, V_{\max}])$[19]. For $V_{\max} \geq \frac{R}{1-\gamma}$, PPI/PQI is R-admissible with respect any $\tilde{\mathcal{R}} \subseteq (\mathcal{S} \times \mathcal{A} \to [-1, 1])$.*

*Proof.* We prove for PPI; the proof for PQI follows similarly. Consider some $\pi$ such that $C :=$ $\sup_{s,a} \frac{d^\pi(s,a)}{\mu(s,a)} < \infty$. Suppose $\tilde{D}$ has $N$ transitions. Take some $V_{\max} \geq \frac{R}{1-\gamma}$ and some $\bar{r}$ from $\mathcal{R}_R(\tilde{r})$. We use the Corollary 1 in [13]: With $I$ large enough, it holds with probability $1 - \delta$,

$$V_{\bar{r}}^\pi(d_0) - V_{\bar{r}}^{\hat{\pi}}(d_0) \leq \tilde{O}\left( \frac{V_{\max}}{(1-\gamma)^3} \left( \sqrt{\frac{|\mathcal{S}||\mathcal{A}|\ln(1/\delta)}{N}} + \frac{\sup_{s,a,t} d_t^\pi(s,a)}{b} \epsilon_\mu + (1-\gamma)\mathbb{E}_{\pi^*,P}[\mathbb{1}[\mu(s,a) \leq 2b]] \right) \right)$$

where $\epsilon_\mu = D_{TV}(\hat{\mu}, \mu) = O(\frac{1}{\sqrt{N}})$. Further,

$$(1-\gamma)\mathbb{E}_{\pi^*,P}[\mathbb{1}[\mu(s,a) \leq 2b]] = \sum_{s,a} d^{\pi^*}(s,a)\mathbb{1}[\mu(s,a) \leq 2b]$$

$$= \sum_{s,a} \mu(s,a)\frac{d^{\pi^*}(s,a)}{\mu(s,a)}\mathbb{1}[\mu(s,a) \leq 2b]$$

$$\leq C\mathbb{E}_\mu[\mathbb{1}[\mu(s,a) \leq 2b]].$$

We can then upper bound the performance difference above as

$$V_{\bar{r}}^\pi(d_0) - V_{\bar{r}}^{\hat{\pi}}(d_0) \leq \tilde{O}\left( \frac{V_{\max}}{(1-\gamma)^3} \left( \sqrt{\frac{|\mathcal{S}||\mathcal{A}|\ln(1/\delta)}{N}} + \frac{\sup_{s,a,t} d_t^\pi(s,a)}{b}\frac{1}{\sqrt{N}} + C\mathbb{E}_\mu[\mathbb{1}[\mu(s,a) \leq 2b]] \right) \right)$$

We bound the latter two terms by tuning $b$. Notice $\mathbb{E}_\mu[\mathbb{1}[\mu(s,a) \leq 2b]] \leq |\mathcal{S}||\mathcal{A}|2b$ at most. This implies

$$\inf_b \frac{\sup_{s,a,t} d_t^\pi(s,a)}{b}\frac{1}{\sqrt{N}} + C\mathbb{E}_\mu[\mathbb{1}[\mu(s,a) \leq 2b]]$$

$$\leq \inf_b \frac{\sup_{s,a,t} d_t^\pi(s,a)}{b}\frac{1}{\sqrt{N}} + C|\mathcal{S}||\mathcal{A}|2b$$

$$= O\left( \frac{\sqrt{C|\mathcal{S}||\mathcal{A}|}}{N^{1/4}} \right)$$

Thus,

$$V_{\bar{r}}^\pi(d_0) - V_{\bar{r}}^{\hat{\pi}}(d_0) \leq \tilde{O}\left( \frac{V_{\max}}{(1-\gamma)^3} \left( \sqrt{\frac{|\mathcal{S}||\mathcal{A}|\ln(1/\delta)}{N}} + \frac{\sqrt{C|\mathcal{S}||\mathcal{A}|}}{N^{1/4}} \right) \right)$$

We can apply this similar argument used in the previous proofs to show that this bound simultaneously applies to all $\bar{r} \in \mathcal{R}_R(\tilde{r})$ since the high-probability concentration analysis is only taken on the data distribution where all rewards in $\mathcal{R}_R(\tilde{r})$ are equal. Thus, when $V_{\max} \geq \frac{R}{1-\gamma}$, PPQ (and similarly PQI) is R-admissibile. $\qquad\square$

---

[19]The original algorithms of PPI and PQI assumes $\mathcal{F} = (\mathcal{S} \times \mathcal{A} \to [0, V_{\max}])$. We note that our extension to $[-V_{\max}, V_{\max}]$ do not change the performance bound as it absorbed by the $\tilde{O}$ notation.

## F.7 MOReL and MOPO

MOReL [33] and MOPO [32] are two popular model-based offline RL algorithms. The two algorithms operate in a similar manner, except that MOReL [33] uses the truncation approach (similar to PPI and PQI [13]) which classifies state-actions into known and unknown sets, whereas MOPO [32] uses negative bonuses (similar to VI-LCB [31]). In this section, we show both algorithms are admissible.

We denote a model as $M = (\mathcal{S}, \mathcal{A}, P_M, r_M, \gamma)$, where $P_M : \mathcal{S} \times \mathcal{A} \to \Delta(\mathcal{S})$ is the model dynamics, and $r_M : \mathcal{S} \times \mathcal{A} \to [-1,1]^{20}$ with $1 \leq R_{\max} < \infty$ being the reward function. The model class is denoted as $\mathcal{M}_{model}$.

### F.7.1 MOReL

We analyze a variant of MOReL which also learns the reward function, whereas MOReL in the originally paper [33] assumes the reward function is given. We suppose model reward $r_M$ and model dynamics $P_M$ are learned by

$$r_M^\star, P_M^\star \in \operatorname*{arg\,min}_{M \in \mathcal{M}_{model}} \mathcal{E}_{\tilde{\mathcal{D}}}(M)$$

where

$$\mathcal{E}_{\tilde{\mathcal{D}}}(M) := \sum_{(s,a,\tilde{r},s') \in \tilde{\mathcal{D}}} -\log P_M(s'|s,a) + (r_M(s,a) - \tilde{r})^2.$$

We define the set of known state-actions $\mathcal{K}_\xi$ as

$$\mathcal{K}_\xi = \Big\{ (s,a) : \sup_{M_1, M_2 \in \mathcal{M}_{\tilde{\mathcal{D}}}^\alpha} \big\{ |r_{M_1}(s,a) - r_{M_2}(s,a)| + \gamma V_{\max} D_{TV}(P_{M_1}(\cdot|s,a), P_{M_2}(\cdot|s,a)) \big\} \leq \xi \Big\},$$
$$\tag{18}$$

with

$$\mathcal{M}_{\tilde{\mathcal{D}}}^\alpha := \{ M \in \mathcal{M}_{model} : \mathcal{E}_{\tilde{\mathcal{D}}}(M) - \min_{M' \in \mathcal{M}_{model}} \mathcal{E}_{\tilde{\mathcal{D}}}(M') \leq \alpha \},$$

We note that $\alpha$ can be chosen as $\Theta(|\mathcal{S}|^2 |\mathcal{A}| \log 1/\delta)$ so that $\mathcal{M}_\alpha$ contains model $(\mathcal{S}, \mathcal{A}, P, \tilde{r}, \gamma)$ with probability $1 - \delta$ for any $\delta \in (0, 1]$ (see [40] for details).

**Remark** Note that we change the definition for the set of known state-actions. In [33], the set is given by

$$\mathcal{K}_\xi^0 = \{ (s,a) : |r_M(s,a) - \tilde{r}(s,a)| + \gamma V_{\max} D_{TV}(P_M(\cdot|s,a), P(\cdot|s,a)) \leq \xi \}. \tag{19}$$

However, it is not impossible to use such a set in practice since the reward function $\tilde{r}$ and true model $P(\cdot|s,a)$ are not given to the learner. Our known state-action set $\mathcal{K}_\xi$, in comparison, is easier to construct. It simply measures the maximum disagreement between two data-consistent models. With a considerate choice of $\alpha$ (see [40]), $\mathcal{K}_\xi$ is always a subset of $\mathcal{K}_\xi^0$. Our definition also matches better with the practical implementation in [33] which uses the disagreement between any two models in an ensemble.

Then MOReL learns policy $\hat{\pi}$ by solving the MDP $(\mathcal{S}, \mathcal{A}, \hat{P}, \hat{r}, \gamma)$, where

$$\hat{r}(s,a) = \begin{cases} r_M^\star(s,a), & s,a \in \mathcal{K}_\xi \\ -X, & \text{otherwise} \end{cases}$$

$$\hat{P}(s'|s,a) = \begin{cases} P_M^\star(s'|s,a), & s,a \in \mathcal{K}_\xi \\ \mathbb{1}(s' = s^\dagger), & \text{otherwise} \end{cases}$$

Note MOReL introduces an absorbing state $s^\dagger$.

Below we prove a performance guarantee of MOReL. The proof here is different from that in the original paper, because here we need to consider reward learning and the original proof does not provide an exact rate that converges to zero as the data size increases. (In the original paper there is a non-zero bias term that depends on the comparator policy.)

---

[20]Both MOPO and MOReL assume that the reward function is known. Here we study the variants of the two algorithms which also learn the reward function.

**Proposition 12** (MOReL Performance Guarantee). *Given data $\tilde{\mathcal{D}} = \{(s, a, \tilde{r}, s')\}$ of size $N$ of an unknown MDP $(\mathcal{S}, \mathcal{A}, \tilde{r}, P, \gamma)$. Suppose $\tilde{r}, r_M^\star \in [-1, 1]$ and $(\tilde{r}, \hat{P}) \in \mathcal{M}_{\tilde{\mathcal{D}}}^\alpha$. Choose $\xi = \min(1, O(V_{max}(|\mathcal{S}|^2|\mathcal{A}|\log(1/\delta)/N)^{1/4}))$. Let $X \geq R$. For $\pi$ such that $\sup_{s,a} \frac{d^\pi(s,a)}{\mu(s,a)} = C < \infty$, with probability $1 - \delta$, it holds that*

$$(1 - \gamma)\left(V_{\tilde{r}}^\pi(d_0) - V_{\tilde{r}}^{\hat{\pi}}(d_0)\right) \leq O\left(V_{\max}\left(\frac{|\mathcal{S}|^2|\mathcal{A}|\log(\frac{1}{\delta})}{N}\right)^{1/4}\right), \quad \text{with } V_{\max} := \frac{R}{1 - \gamma}$$

*for all $\bar{r} \in \mathcal{R}_R(\tilde{r}) := \{\bar{r} : \bar{r}(s,a) = \tilde{r}(s,a), \forall(s,a) \in supp(\mu) \text{ and } \bar{r}(s,a) \in [-R, 1], \forall(s,a) \in \mathcal{S} \times \mathcal{A}\}$.*

*Proof.* We extend $\tilde{r}$ as $\tilde{r}(s^\dagger, a) = -X$ for all $a \in \mathcal{A}$. Let $\hat{M} = (\hat{r}, \hat{P})$.

First, by the optimality of $\hat{\pi}$, we write

$$V_{\bar{r}}^\pi(d_0) - V_{\bar{r}}^{\hat{\pi}}(d_0)$$
$$= V_{\bar{r}}^\pi(d_0) - V_{\hat{M}}^\pi(d_0) + V_{\hat{M}}^\pi(d_0) - V_{\hat{M}}^{\hat{\pi}}(d_0) + V_{\hat{M}}^{\hat{\pi}}(d_0) - V_{\bar{r}}^{\hat{\pi}}(d_0)$$
$$\leq V_{\bar{r}}^\pi(d_0) - V_{\hat{M}}^\pi(d_0) + V_{\hat{M}}^{\hat{\pi}}(d_0) - V_{\bar{r}}^{\hat{\pi}}(d_0)$$

where $V_{\hat{M}}^\pi$ denotes the value of policy $\pi$ with respect to the reward $\hat{r}$ and dynamics $\hat{P}$. The last inequality follows from the optimality of $\hat{\pi}$ on $\hat{M}$.

Then we notice the fact that $\mathcal{K}_\xi \subseteq \text{supp}(\mu)$ with a small $\xi$. The proof of the lemma below is given in Appendix F.7.3.

**Lemma 7.** *If $\xi \leq 1$, $\mathcal{K}_\xi \subseteq supp(\mu)$.*

Let $\hat{d}^\pi$ denote the average state-action visitation of $\pi$ with respect to $\hat{P}$.

$$(1 - \gamma)\left(V_{\hat{M}}^{\hat{\pi}}(d_0) - V_{\bar{r}}^{\hat{\pi}}(d_0)\right)$$
$$= \mathbb{E}_{s,a\sim\hat{d}^{\hat{\pi}}}[\hat{r}(s,a) + \gamma\mathbb{E}_{s'\sim\hat{P}|s,a}[V_{\bar{r}}^{\hat{\pi}}(s')] - \bar{r}(s,a) - \gamma\mathbb{E}_{s'\sim P|s,a}[V_{\bar{r}}^{\hat{\pi}}(s')]]$$
$$\leq \mathbb{E}_{s,a\sim\hat{d}^{\hat{\pi}}}[\hat{r}(s,a) + \gamma\mathbb{E}_{s'\sim\hat{P}|s,a}[V_{\bar{r}}^{\hat{\pi}}(s')] - \bar{r}(s,a) - \gamma\mathbb{E}_{s'\sim P|s,a}[V_{\bar{r}}^{\hat{\pi}}(s')]|(s,a) \notin \mathcal{K}_\xi]$$
$$\quad + \mathbb{E}_{s,a\sim\hat{d}^{\hat{\pi}}}[|\hat{r}(s,a) - \bar{r}(s,a)| + \gamma V_{\max}D_{TV}(\hat{P}(\cdot|s,a), P(\cdot|s,a))|(s,a) \in \mathcal{K}_\xi]$$
$$\leq \xi + \mathbb{E}_{s,a\sim\hat{d}^{\hat{\pi}}}[\hat{r}(s,a) + \gamma\mathbb{E}_{s'\sim\hat{P}|s,a}[V_{\bar{r}}^{\hat{\pi}}(s')] - \bar{r}(s,a) - \gamma\mathbb{E}_{s'\sim P|s,a}[V_{\bar{r}}^{\hat{\pi}}(s')]|(s,a) \notin \mathcal{K}_\xi]$$
$$\leq \xi + \mathbb{E}_{s,a\sim\hat{d}^{\hat{\pi}}}[-V_{\max} - \bar{r}(s,a) + \gamma V_{\max}|(s,a) \notin \mathcal{K}_\xi]$$
$$\leq \xi$$

where in the second inequality we use $\mathcal{K}_\xi \subseteq \text{supp}(\mu)$ (which implies $\tilde{r} = \bar{r}$ in $\mathcal{K}_\xi$).

Similarly we can show

$$(1 - \gamma)\left(V_{\hat{M}}^\pi(d_0) - V_{\bar{r}}^\pi(d_0)\right)$$
$$\geq -\xi + \mathbb{E}_{s,a\sim d^\pi}[\hat{r}(s,a) + \gamma\mathbb{E}_{s'\sim\hat{P}|s,a}[V_{\hat{M}}^\pi(s')] - \bar{r}(s,a) - \gamma\mathbb{E}_{s'\sim P|s,a}[V_{\hat{M}}^\pi(s')]|(s,a) \notin \mathcal{K}_\xi]$$
$$\geq -\xi - 2V_{\max}\mathbb{E}_{s,a\sim d^\pi}[\mathbb{1}[(s,a) \notin \mathcal{K}_\xi]]$$

Combining the three inequalities above gives

$$(1 - \gamma)\left(V_{\bar{r}}^\pi(d_0) - V_{\bar{r}}^{\hat{\pi}}(d_0)\right) \leq 2V_{\max}\mathbb{E}_{s,a\sim d^\pi}[\mathbb{1}[(s,a) \notin \mathcal{K}_\xi]] + 2\xi$$

By the concentratability assumption, we can further upper bound

$$\mathbb{E}_{s,a\sim d^\pi}[\mathbb{1}[(s,a) \notin \mathcal{K}_\xi]] \leq C\mathbb{E}_{s,a\sim\mu}[\mathbb{1}[(s,a) \notin \mathcal{K}_\xi]]$$

Therefore we have

$$(1 - \gamma)\left(V_{\bar{r}}^\pi(d_0) - V_{\bar{r}}^{\hat{\pi}}(d_0)\right) \leq 2V_{\max}\mathbb{E}_{s,a\sim\mu}[\mathbb{1}[(s,a) \notin \mathcal{K}_\xi]] + 2\xi$$
$$\leq 2V_{\max}\frac{\mathbb{E}_{s,a\sim\mu}[E_{\sup}(s,a)]}{\xi} + 2\xi$$
$$\leq 4V_{\max}\frac{\sup_{r_M, P_M}\mathbb{E}_{s,a\sim\mu}[E(s,a; r_M, P_M)]}{\xi} + 2\xi$$

where the last step is Markov inequality,

$$E_{\sup}(s,a) = \sup_{M_1, M_2 \in \mathcal{M}_{\tilde{\mathcal{D}}}^\alpha} \left\{ |r_{M_1}(s,a) - r_{M_2}(s,a)| + \gamma V_{\max} D_{TV}(P_{M_1}(\cdot|s,a), P_{M_2}(\cdot|s,a)) \right\},$$

and

$$E(s,a; r_M, P_M) = |r_M(s,a) - \tilde{r}(s,a)| + \gamma V_{\max} D_{TV}(P_M(\cdot|s,a), P(\cdot|s,a)).$$

Now we upper bound the expectation of $E$ over $\mu$. With probability greater than $1 - \delta$,

$$
\begin{aligned}
\mathbb{E}_\mu[E(s,a; r_M, P_M)] &\leq \sqrt{\mathbb{E}_\mu[(|r_M(s,a) - \tilde{r}(s,a)| + \gamma V_{\max} D_{TV}(P_M(\cdot|s,a), P(\cdot|s,a)))^2]} \\
&\leq \sqrt{2}\sqrt{\mathbb{E}_\mu[(r_M(s,a) - \tilde{r}(s,a))^2 + (\gamma V_{\max})^2 D_{TV}(P_M(\cdot|s,a), P(\cdot|s,a))^2]} \\
&\leq (1 + \gamma V_{\max})\sqrt{2}\sqrt{\mathbb{E}_\mu[(r_M(s,a) - \tilde{r}(s,a))^2 + D_{TV}(P_M(\cdot|s,a), P(\cdot|s,a))^2]} \\
&\leq V_{\max}\sqrt{2}\sqrt{\mathbb{E}_\mu[(r_M(s,a) - \tilde{r}(s,a))^2 + D_{TV}(P_M(\cdot|s,a), P(\cdot|s,a))^2]} \\
&\leq V_{\max}\sqrt{2}\sqrt{\frac{|\mathcal{S}|^2|\mathcal{A}|\log(\frac{1}{\delta})}{N}}
\end{aligned}
$$

where the last step is based on Lemma 8.

**Lemma 8.** *[34] With probability $1 - \delta$, for any MDP model $M$,*

$$\mathbb{E}_\mu[|r_M(s,a) - r(s,a)|^2 + D_{TV}(P_M(\cdot|s,a), P(\cdot|s,a))^2] \leq O\left(\frac{\mathcal{E}_\mathcal{D}(M) - \min_{M'} \mathcal{E}_\mathcal{D}(M') + |\mathcal{S}|^2|\mathcal{A}|\log(\frac{1}{\delta})}{N}\right).$$

Thus, we have

$$
\begin{aligned}
(1 - \gamma)\left(V_{\tilde{r}}^\pi(d_0) - V_{\tilde{r}}^{\hat{\pi}}(d_0)\right) &\leq O\left(\frac{V_{\max}^2 \sqrt{|\mathcal{S}|^2|\mathcal{A}|\log(\frac{1}{\delta})}}{\xi\sqrt{N}}\right) + 2\xi \\
&\leq O\left(V_{\max}\left(\frac{|\mathcal{S}|^2|\mathcal{A}|\log(\frac{1}{\delta})}{N}\right)^{1/4}\right).
\end{aligned}
$$

$\square$

By Proposition 12, we can apply the previous analysis technique to MOReL to show it is admissible without addition union bounds, which leads to the corollary below.

**Corollary 7.** *For $X \geq R$, MOReL is $R$-admissible with respect any $\tilde{\mathcal{R}} \subseteq (\mathcal{S} \times \mathcal{A} \to [-1, 1])$.*

### F.7.2 MOPO

MOPO [32] is very similar to MOReL except that it uses negative bonuses (like VI-LCB) instead of truncation. In the original paper of MOPO, the authors assume the reward is given. Here we consider a variant that also learns the reward. Specifically, given $P_M^\star$ and $r_M^\star$ above it solves the MDP $(\mathcal{S}, \mathcal{A}, P_M^\star, \hat{r}, \gamma)$, where

$$\hat{r}(s,a) = r_M^\star(s,a) - b(s,a)$$

While the original MOPO paper does not give a specific design of $b(s,a)$ that is provably correct, in principle making MOPO provably correct is possible. Essentially, we need to choose a large enough bonus to cover the error of the model-based Bellman operator (defined by the estimated reward and dynamics). This would lead to a bonus of order $V_{\max}$ (see the analysis of VI-LCB in Appendix F.5). Then we can proceed with an analysis that combines that of MOReL and VI-LCB to prove MOPO's performance guarantee. This can then be used to show (like the previous proofs) that MOPO is admissible. We omit the proof here.

**Corollary 8.** *Suppose $b(s,a) = \Theta(V_{\max})$ for $(s,a) \notin supp(\mu)$. For $V_{\max} \geq \frac{R}{1-\gamma}$, MOPO is $R$-admissible with respect any $\tilde{\mathcal{R}} \subseteq (\mathcal{S} \times \mathcal{A} \to [-1, 1])$.*

### F.7.3 Proof of Technical Lemma

*Proof of Lemma 7.* Consider any $(\bar{s}, \bar{a}) \notin \text{supp}(\mu)$. Given any $M = (\mathcal{S}, \mathcal{A}, P_M, r_M, \gamma) \in \mathcal{M}_{\mathcal{D}}^{\alpha}$, we construct $M' = (\mathcal{S}, \mathcal{A}, P_M, r_{M'}, \gamma)$ with $r_{M'} = r_M$ for all $(s, a) \in (\mathcal{S} \times \mathcal{A}) \setminus \{(\bar{s}, \bar{a})\}$, and $r_{M'}(\bar{s}, \bar{a}) = -\text{sign}(r_M(\bar{s}, \bar{a}))$. By construction, since $M$ and $M'$ agrees on all state-actions except $(\bar{s}, \bar{a})$, which is not in data support, we have $M' \in \mathcal{M}_{\mathcal{D}}^{\alpha}$. However, we have that

$$\sup_{M_1, M_2 \in \mathcal{M}_{\mathcal{D}}^{\alpha}} \left\{ |r_{M_1}(\bar{s}, \bar{a}) - r_{M_2}(\bar{s}, \bar{a})| + \gamma V_{\max} D_{TV}(P_{M_1}(\cdot|\bar{s}, \bar{a}), P_{M_2}(\cdot|\bar{s}, \bar{a})) \right\}$$
$$\geq |r_M(\bar{s}, \bar{a}) - r_{M'}(\bar{s}, \bar{a})| + \gamma V_{\max} D_{TV}(P_M(\cdot|\bar{s}, \bar{a}), P_M(\cdot|\bar{s}, \bar{a}))$$
$$= |r_M(\bar{s}, \bar{a}) - r_{M'}(\bar{s}, \bar{a})| \geq 1 \geq \xi.$$

This means that $(\bar{s}, \bar{a}) \notin \mathcal{K}_{\xi}$. Therefore $\mathcal{K}_{\xi} \subseteq \text{supp}(\mu)$. $\qquad\square$

## G  Grid World

In this section, we describe the full details of the grid world study that was discussed in Section 3.1.

**Environment.**  We consider goal-directed navigation in the environment shown in Fig. 10. We use a fixed grid world layout shown in the figure. The agent's state is given by a tuple $(x, y, d)$ where $(x, y) \in [5]^2$ represents the 2-d coordinate and $d \in \{N, W, E, S\}$ encodes the North, West, East, and South direction respectively that the agent is facing. The agent's action space is $\mathcal{A} = \{f, l, r\}$ where $f$ denotes the action of moving to grid square in front of the agent, $l$ denotes a left turn of 90 degrees, and $r$ denotes a right turn of 90 degrees. The agent can go through the lava (orange wavy square) but cannot go through the wall (grey square). The goal (key) is an absorbing state. The agent gets a reward of +1 when first visiting the goal followed by a reward of 0 for then staying in the goal. The agent gets a reward of -1 for reaching lava and a reward of -0.01 for all other actions to incentivize the agent to reach the goal along the shortest safe path. We use a horizon of $H = 20$. The agent deterministically starts in the top-right corner shown in Fig. 10.

**Dataset.**  We collect a dataset $\mathcal{D}$ of 500 episodes by taking 100 identical optimal episodes along with 400 identical episodes that all touch the topmost lava field. We introduce a length bias by terminating an episode if the agent touches the lava or fails to reach the goal in $H - 1$ steps (i.e., the agent doesn't survive). In addition to the original setting with the observed reward, we consider three additional settings where the rewards in the dataset are replaced by: (i) a constant zero reward, (ii) their negative, and (iii) a random number sampled uniformly in $[0, 1]$.

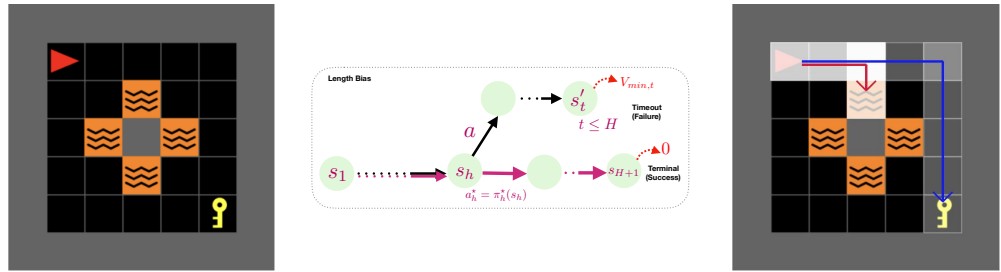

Figure 10: **Left**: A goal-directed gridworld navigation task where the agent (red triangle) has to reach the goal (key) and avoid lava (orange waves). **Center**: Shows the length bias that helps offline RL succeed even when rewards are entirely replaced with random or adversarial values. **Right**: Results on the grid world task. Behavior cloning fails to solve the task (red path) while using an offline RL method leads to success (blue path) *even with wrong rewards*. The opacity of the grid square overlay indicates how frequent that state is in the dataset (more opaque means more frequent).

**Algorithm 4** PEVI($\mathcal{S}, \mathcal{A}, H, \mathcal{D}, \beta, V_{\min}, V_{\max}$) [12]. We are given access to a set $\mathcal{D}$ of episodes ($\tau$). Additionally, we are given a hyperparameter $\beta \in \mathcal{R}_{\geq 0}$ that controls pessimism. The hyperparameters $V_{\min}, V_{\max} \in \mathcal{R}^H$ indicate bounds on policy value with $V_{\min,h}$ and $V_{\max,h}$ denoting the minimum and maximum policy values respectively for time step $h \in [H]$. Unlike the original PEVI algorithm of [12], we treat $V_{\min}, V_{\max}$ as hyperparameters.

---

   Set $V_{H+1} = 0$
   **for** $h = H, \cdots, 1$ **do**
     **for** $s \in \mathcal{S}$ **do**
       **for** $a \in \mathcal{A}$ **do**
         $n_h(s,a) = \sum_{\tau \in \mathcal{D}} \mathbb{1}\{s_h^\tau = s \wedge a_h^\tau = a\}$
         **If** $n_h(s,a) > 0$
           $\tilde{Q}_h(s,a) = \frac{1}{n_h(s,a)} \sum_{\tau \in \mathcal{D}} \mathbb{1}\{s_h^\tau = s \wedge a_h^\tau = a\}\{r_h^\tau + V_{h+1}(s_{h+1}^\tau)\}$
           $\bar{Q}_h(s,a) = \tilde{Q}_h(s,a) - \frac{\beta}{\sqrt{n_h(s,a)}}$
           $\widehat{Q}_h(s,a) = \max\{V_{\min,h}, \min\{\bar{Q}_h(s,a), V_{\max,h}\}\}$
         **Else**
           $\widehat{Q}_h(s,a) = V_{\min,h}$
       **end for**
       $\pi_h(s) = \arg\max_{a \in \mathcal{A}} \widehat{Q}_h(s,a)$
       $V_h(s) = \widehat{Q}_h(s, \pi_h(s))$
     **end for**
   **end for**
   **Return** $\pi = (\pi_{1:H})$

---

**Methods.** We evaluate two methods: behavior cloning which simply takes the action with the highest empirical count in a given state and a random action if the state was unseen, and PEVI [12] an offline RL method with guarantees for tabular MDP. We provide a pseudocode of PEVI in Algorithm 4. Unlike the original PEVI in [12], we provide minimum and maximum bounds for the value function for each time step as an additional hyperparameter. These bounds should be dictated by our CMDP framework and should penalize out-of-distribution actions. In principle, we need to set these bounds such that the algorithm is admissible. Please see Appendix D and Appendix E for details.

Intuitively, PEVI performs dynamic programming similar to a standard value iteration with two key changes. Firstly, it uses pessimistic value initialization where the learned value $V_h(s)$ of a state $s$ at time step $h$ is set to the lowest possible return $V_{min,h}$ if the state is unseen at time step $h$. Secondly, when performing value iteration it adds a pessimism penalty of $-\beta/\sqrt{n(s,a)}$ to the dataset reward where $n(s,a)$ is the empirical state-action count in $\mathcal{D}$. This pessimism penalty ensures that the learned value function lower bounds the true value function (hence a pessimistic estimate). In our experiments, we vary $\beta$ and set $V_{\min,h} = \tilde{V}_{\min,h} - \beta(H - h + 1) - 1$ where $\tilde{V}_{\min,h}$ is the minimum possible return for the given reward type that we are working with. The value of $\tilde{V}_{\min,h}$ is $-(H - h + 1)$ for the original reward, $-1$ for the negative reward, and $0$ for the zero reward and random reward cases). We define $V_{\max,h} = \tilde{V}_{\max,h}$ where $\tilde{V}_{\max,h}$ is the maximum possible return for the given reward type. The value of $\tilde{V}_{\max,h}$ is 1 of the original reward, $(H - h + 1)$ for the random reward and the negative reward, and 0 for the zero reward case.

**Results.** We show numerical results in Table 16. We visualize results in Fig. 10. The behavior cloning simply imitates the most common trajectory in the dataset which results in going to the lava (shown in red in the figure). In contrast, the PEVI is able to reach the goal in every reward situation.

**Explanation.** The reason why PEVI works can be understood simply by pessimistic initialization and length bias. The length bias in this setting is visualized in the center of Fig. 10. PEVI assigns the final state $s_t'$ of a failed trajectory will get a pessimistic value of $V_t(s_t') = V_{\min,t}$. This value isn't updated as we never take any action in $s_t'$ due to timeout. In our case, $s_t'$ will be the state where the agent first visits the lava. In contrast, the final state of a successful trajectory $s_{H+1}$ gets assigned a value of 0. If $V_{\min}$ is sufficiently negative, then PEVI will only consider policies that stay all $H$ steps within the data support. Due to length bias, this is not true for the non-surviving trajectories that

| Reward Type | Behavior Cloning | PEVI |
|---|---|---|
| Original Reward | $-1.19 \pm 0.00$ | $0.92 \pm 0.00$ |
| Zero Reward | $-1.19 \pm 0.00$ | $0.92 \pm 0.00$ |
| Random Reward | $-1.19 \pm 0.00$ | $0.92 \pm 0.00$ |
| Negative Reward | $-1.19 \pm 0.00$ | $0.92 \pm 0.00$ |

Table 16: **Gridworld results:** Mean return on 1000 test episodes. PEVI achieves the optimal return of $V^\star = 0.92$.

reach the lava. Further, the only trajectories in our case that complete end up reaching and staying in the goal. Therefore, under the assumption that $V_{\min,h}$ are all sufficiently small, PEVI will learn a policy reaches the goal irrespective of the correctness of the reward in the data. The sufficiency of the negativity of $V_{\min,h}$ is implied by our CMDP framework. Lastly, note that as the world is deterministic, we didn't need to use the pessimism arising from the $\frac{-\beta}{\sqrt{n(s,a)}}$ penalty term.

