# OpenReview forum: "Survival Instinct in Offline Reinforcement Learning"
_NeurIPS.cc/2023/Conference — NeurIPS 2023 spotlight_

### Official Review · Reviewer_kbuj · 2023-06-28

**Soundness:** 4 excellent
**Presentation:** 4 excellent
**Contribution:** 3 good
**Rating:** 7
**Confidence:** 3

**Summary:**

The authors make an interesting and important observation about offline Reinforcement Learning. They note that an agent can learn from offline data even when the reward signal is not the same one as that used to train the online agent. Moreover it can be vastly different, or even with no reward signal at all, within certain types of task and with certain datasets. In particular, when a dataset has limited and biased data coverage, the aim of an offline agent to stay within the data distribution means that it has an equivalent of a survival instinct. This point is noted first empirically and then proven formally as well as being shown in an extensive set off environments and settings.

The conditions necessary of the training data for an agent to learn a near-optimal policy are identified. Finally, a perspective on offline RL is given which means that the reward signal can be essentially ignored, so long as the appropriately biased dataset is provided.

**Strengths:**

The originality and quality of the paper are both very strong. The combination of formal mathematical proofs of the insensitivity to the rewards and the experiments which show this to be true are married well together.  The paper as a whole is very well written, and the appendix is in particular very thorough and shows a good understanding of the need for transparency and clarity in experimental settings. The amount of formality within the bulk of the paper is also appropriate, giving enough to understand the results, but not to swamp them.

The nature of the findings are themselves original and are likely to have impact in the field of offline RL.

**Weaknesses:**

I see few weaknesses overall, although there are a few typos throughout:

In most of the figures, the number of seeds is provided, though in figure 1 it is not. This should be added.
Line 120: "worst case a necessary"
Line 143: "In following"
Line 178: "s, a" may be clearer as (s,a).
Line 214: "This is contrast"
Line 227: "Positive data bias assumption"
Footnote on page 5 has an additional bracket at the end.
Line 232: "benefits"->"benefit"
Line 245: "states have a safe action"
Line 247: "is also mild"
Footnote on page 6: "we include a ablation"
Line 290: "with wrong can"
Line 295 needs to be rewritten.
Line 310: "with provably pessimism"
Line 315: "A implicit"
Line 318: "due the"
Line 332: "the end of episode"
Line 334: "there is no timeouts"
Line 915 in the appendix: "the followings are true"
Line 1025: "first we the"
Line 1089: "and let the goal state to be"





**Questions:**

I think that a discussion under which circumstances an adversarial reward signal could be designed to attack an offline agent may be interesting. While it is clear that with the appropriately biased data, a random reward signal has little effect, there may be circumstances where a strong enough adversarial reward signal and an inappropriately biased dataset may not be sufficient to lead to near optimal performance.

Is it possible to make claims in the multi-agent offline setting similar to those in the single agent setting?

**Limitations:**

Limitations of the work are given, but none of the potential societal impact of this work has been discussed. Given that Offline RL may well play a large role in ML systems going forward, this should be mentioned.

---

> ### Author Rebuttal · Authors · 2023-08-10
>
> We are happy that the reviewer overall feels positive about our paper. We want to thank the reviewer for reviewing our paper and pointing out typos in detail. We will fix these typos in the final version.
>
> **Adversarial Reward**
>
> Regarding the question on adversarial reward, we agree with the reviewer that when a dataset does not have positive bias, offline RL may suffer from bad data reward. Indeed, in our experiments on the halfcheetah datasets, we did observe that the wrong data reward can make offline RL algorithms produce bad policies (see Fig. 3). From Definition 2, *theoretically*, an admissible (see Definition 4) offline RL learning a positively biased data can guard against any adversarial reward within the reward class $\tilde{\mathcal{R}}$ . However, we currently don’t have way to *empirically* verify whether a deep offline RL implementation is admissible, or quantify the amount of positive bias of a data distribution, like the size of reward class $\tilde{\mathcal{R}}$. Currently, our suggestion is to actually run the offline RL algorithm with wrong data rewards and see whether the learned behavior differs from that given by learning with the true reward. One interesting future direction is to investigate whether/how offline RL algorithms can be robust to adversarial reward that changes during training (Definition 2 only concerns a fixed reward).
>
> **Multi-agent Offline RL**
>
> As the reviewer also pointed out, exploring whether multi-agent offline RL algorithms have a similar survival instinct is another interesting future direction. We will discuss these future research directions in the limitations in the final version of the manuscript.
>
> **Societal Impact**
>
> Lastly, we agree with the reviewer on the growing societal impact of offline RL going forward. We think the discovery here might shed light on a potential positive impact of offline RL. By survival instinct, offline RL may be able to train sequential-decision policies that have good behaviors without needing to collect negative data. This is different from the common belief that RL can learn to behave well only if it has seen both positive (success) and negative (failure) data. This ability to learn from one-sided data is important especially for applications where collecting negative data is unethical, e.g., learning not to produce hateful speech by first collecting hateful speech, learning not to harm patients in medical applications by first harming some, etc.
>
> On the flip side, also by survival instinct, offline RL can be prone to existing societal (e.g., gender, racial) biases in data, and, moreover, such a bias cannot be easily corrected by relabeling data with a different reward. As a result, when using an offline RL algorithm, more strategic thinking on data collection might be needed. We encourage researchers and practitioners to collect datasets using methodology such as those proposed in [1] and provide details such as how data was collected and cleaned so that users can assess whether it is appropriate to train offline RL algorithms with these datasets.
>
> [1]  Gebru, Timnit, et al. “Datasheets for datasets.” Communications of the ACM 2021.

---

> > ### Comment · Reviewer_kbuj · 2023-08-12
> > **Reply to authors**
> >
> > I am happy with these comments and will keep my initial rating for this paper.

---

### Official Review · Reviewer_k1TU · 2023-07-02

**Soundness:** 4 excellent
**Presentation:** 3 good
**Contribution:** 3 good
**Rating:** 6
**Confidence:** 3

**Summary:**

This work displays a new observation that offline reinforcement learning (RL) algorithms can develop efficient policies, even when trained with incorrect reward labels. The authors attribute this resilience to the pessimistic nature of offline RL algorithms and the inherent biases in data collection processes. The findings introduce a ‘survival instinct’ in the systems that inform how we understand offline RL benchmarks and create future ones. The study, therefore, recommends a new strategy for offline RL that promotes learning of desired behavior through flawed rewards but intentional bias in data coverage.

**Strengths:**

1. A good warm-up experiment to help readers understand the motivation.
2. Sound theoretical analysis to show the property of ‘survival instinct’ in current offline RL algorithms.
3. Comprehensive experiments on current D4RL and meta-world benchmarks with different offline RL algorithms.

**Weaknesses:**

1. There exists a gap between the intuition (longer trajectories in the data have a smaller optimality gap) and the definition of the Positive Data Bias. On the other hand, Positive Data Bias does not have a quantitative value for each dataset so that the reader can clearly see the difference between the different datasets.
2. Authors claim that offline RL is intrinsically safe. However, missing relevant experiments to validate their claims.

**Questions:**

3. Regarding the intuition about the positive data bias, I am pretty interested in the experiments in the Antmaze dataset from D4RL, where it is a multi-goal task and there does not exist positive data bias claimed by the authors.
4. A detailed analysis of the influence of the three types of “wrong” reward labels can make this work more complete and robust.
5. More experiments about safe RL are highly appreciated [1].
6. In line 247 about the safe RL. "This condition can be easily satisfied by filtering out unsafe states in post processing." From the previous work [2], one biggest reasons why we prefer offline RL over BC is that offline RL can learn that those states are dangerous. What does the author think about this?

[1] Liu Z, Guo Z, Lin H, et al. Datasets and Benchmarks for Offline Safe Reinforcement Learning[J]. arXiv preprint arXiv:2306.09303, 2023.

[2] Kumar A, Hong J, Singh A, et al. When should we prefer offline reinforcement learning over behavioral cloning?[J]. arXiv preprint arXiv:2204.05618, 2022.


**Limitations:**

see the above

---

> ### Author Rebuttal · Authors · 2023-08-10
>
> **Quantifying Positive Bias**
>
> We would like to clarify that positive bias is a general concept of when approximate optimality to the CMDP implies approximate optimality w.r.t the true reward, while *"longer trajectories in the data have a smaller optimality gap"* refers to a special form of positive data bias called *length bias*. In Prop. 3 (Appendix C.4), we provide a few examples of positive data bias, including length bias.
>
> We argue that the performance gap between running an offline RL algorithm with true reward and other data rewards can be used to measure positive data bias (like our experiments). An interesting future direction is to quantify the positive bias in a dataset without having to run offline RL algorithms. We will discuss this in the limitations.
>
> **Inherent Safety**
>
> Our results on episode lengths for D4RL datasets in Fig. 1 (b) and Appendix B.4 are evident that offline RL is inherently safe. For hopper and walker, the cost can be defined by whether the agent falls, and an episode stops when the agent incurs a non-zero cost, i.e., an agent is the safest if it does not fall. In Fig. 7, we show that offline RL algorithms, e.g., ATAC and PSPI, can achieve long episode lengths even when the behavior policy is unsafe. We conduct new experiments on offline safety gymnasium [1] below, as the reviewer suggested.
>
> **Antmaze**
>
> As discussed in Section 4 (line 260), we acknowledge not all datasets have positive bias. With that said, AntMaze datasets technically have a positive bias, albeit a weak one. Although AntMaze datasets are collected by multi-goal policies, the terminal flags are *re-labeled based on a single test-time goal*. In other words, trajectories reaching the test-time goal are marked as terminal (which mathematically has an infinite length) while others have a bounded length of at most 1001. We refer the reviewer to Appendices D.2 and D.3 where we formally discuss how this creates a positive data bias.
>
> However, such a length bias is hard to be picked up by a deep offline RL implementation, because failed trajectories, which can be very long, would have similar *discounted* returns as the successful ones. In fact, it has been proved that, for AntMaze medium and large datasets, good performance can only be achieved when using *an algorithm-specific transformed reward*: IQL requires one to be subtracted from the rewards and CQL requires the reward to be shifted and scaled to [-5, 5] [2].  During our earlier experiments, we found the results on the AntMaze inconclusive, as it is hard to separate optimization difficulty from the effect of different data rewards.
>
> **Analysis on Wrong Rewards**
>
> Empirically, we observe that, for ATAC and PSPI, the learned policies from zero and random rewards are similar, as the random reward is in expectation a constant. In more diverse datasets, such as the medium-replay datasets, we often find policies learned with the negative reward worse than those learned using zero and random reward.
>
> On the theoretical side, Definition 2 allows a dataset to have positive bias w.r.t. one of the wrong rewards, but not with the others. In Prop. 3 (Appendix C), we provide several factors (reward shaping reward, expert demonstrations, and length bias) that can lead to positive data bias.
>
> **Extra Experiments on Offline Safety RL**
>
> We conduct experiments on offline safety gymnasium [1] and include the results in the attached PDF. We make a few remarks which are important to interpreting the results.
> 1. Our notion of inherent safety is different from [1]. Ours (Corollary 2) focus on offline RL’s ability to produce safe policies given a dataset that *only* contains safe states, while [1] focuses on learning the concept of safety from a dataset which contains *both* safe and unsafe behaviors. Our notion of inherent safety can be more favorable when it is inadmissible/unethical to collect unsafe data.
> 2. For this reason, the datasets in [1] contain unsafe states. We use a naïve filtering strategy, which removes all transitions with non-zero costs. This ensures our data only covers safe states, but it also means that we use less data than the comparators, algorithms in [1]. Better results may be possible with a more sophisticated filtering strategy.
> 3. Many algorithms from [1] use a cost target, which sets the maximum total cost allowed by safety. [1] uses three different cost targets, {20, 40, 80}, and reports the normalized total reward and cost for each algorithm averaged over the three targets. By contrast, standard offline RL algorithms do not use such a cost target. We report results of offline RL algorithms according to an “effective cost target” of 34.29, which is similar to how BC_All results are presented in [1].
>
> We observe that *offline RL with a naïve data filtering strategy achieves comparable performance as the per-task best performing offline safe RL algorithm*. Our agents in general incur low cost except for in the circle tasks. We hope these new results can give the reviewer more confidence on the safety property of offline RL.
>
> **Filtering Unsafe States**
>
> Filtering often naturally happens during data collection, e.g., a robot is likely to be stopped when it’s about to crash. In this case, we have a dataset with only safe states, and some of the trajectories are incomplete due to intervention. By Corollary 2, offline RL can learn a safe policy on such a dataset, *provided that a safe in-support policy exists*. This is because due to survival instinct, offline RL algorithms would 1) assign a low value to such an unsafe state and 2) propagate this low value to states and actions encountered earlier. BC, however, may follow an unsafe trajectory and enter an unsafe state. This is demonstrated in the D4RL episode length results in Appendix B.4.
>
> [1] Liu et al. Datasets and Benchmarks for Offline Safe Reinforcement Learning. 2023
>
> [2] Tarasov et al. CORL: Research-oriented deep offline reinforcement learning library. 2022

---

> > ### Comment · Reviewer_k1TU · 2023-08-21
> > **Official Reply from Reviewer k1TU**
> >
> > I appreciate the authors' clarifications, and most of my concerns have been addressed. At the same time, I am impressed by the experiments in [1] which show that offline RL methods with naïve data filtering achieve comparable performance to state-of-the-art offline safe RL algorithms. I believe this work and the d4rl Mujoco dataset will attract the attention of the offline RL community. The previous work [2] has mentioned the phenomenon of learning with incorrect reward signals, but the strength of this work lies in its exhaustive experimentation and analysis of this phenomenon. I will maintain my rating, and I suggest citing the work of Shin D et al. [2] in this paper.
> >
> > [2] Shin D, Dragan A D, Brown D S. Benchmarks and algorithms for offline preference-based reward learning[J]. arXiv preprint arXiv:2301.01392, 2023.

---

> > > ### Author Response · Authors · 2023-08-21
> > >
> > > We also found this paper [2] very recently. We will cite it in the final version.

---

### Official Review · Reviewer_Sf8f · 2023-07-02

**Soundness:** 4 excellent
**Presentation:** 4 excellent
**Contribution:** 3 good
**Rating:** 6
**Confidence:** 4

**Summary:**

This  paper reports a very interesting new phenomenon that would be interesting to the offline RL community. It demonstrates that even when trajectories have the wrong reward labels, offline RL can learn good policies. The paper's experiments attempt to dissect the reasons for why this surprising phenomenon emerges, and argues that special to offline RL algorithms, pessimism endows the agent with survival instinct to stay within the data support, so that safe policies can be learned within these constraints.


**Strengths:**

The phenomenon that has been identified is surprising; the experiments done to dissect this phenomenon are clear and solid. Overall a solid paper.

**Weaknesses:**

I have only one concern, and welcome further clarification by the authors. The authors claim that their algorithm demonstrates this robustness phenomenon arises due to the survival instinct of offline RL algorithm. But it seems from the paper that the circumstance of this only arises in the situation that the data collected for offline RL has long timescale trajectories. Is this the case? For instance, it does not arise in other circumstance investigated, like when there are multiple sources of data, or other circumstances that deviate from this central feature.

If this is the case, then the authors should state this clearly. This is still very much an interesting phenomenon, but is slightly more limited than the first impression one gets when reading the abstract and introduction that this is a potentially quite a universal phenomenon in offline RL.

If long trajectories is not the only circumstance in which this phenomenon, the authors can make the paper even more clearer by listing all the circumstnaces in which this phenomenon arise, which would be very helpful for the reader.

Regardless, as long as the authors match their conclusions to their demonstrations, this paper demonstrates indeed a very interesting phenomenon arising in offline RL that is worthy of publication.


**Questions:**

See questions present in the above sections.
And one more question, about the suggestion at the final part of the abstract (and conclusion):
-  if the implicit bias for long trajectories in offline data is the sole feature that is responsible for this wrong labels phenomenon, then is it still reasonable to suggest that "whereby an agent is “nudged” to learn a desirable behavior with 20 imperfect reward but purposely biased data coverage"?

In other words, does the "survival instinct" as discussed in the paper really have the sensitivity to make such a (next step) hypothesis reasonable?

**Limitations:**

The authors have a limitations section, and are fair. They also nicely clarify that of course, offline RL does not always learn from wrong rewards.

---

> ### Author Rebuttal · Authors · 2023-08-10
>
> **When positive bias arises & clarification on "long timescale trajectories"**
>
>
> We thank reviewer Sf8f for providing feedback to our manuscript. The robustness to reward is attributable to an interplay between survival instinct in ofﬂine RL algorithms (due to their pessimistic nature) and positive data bias. There can be *multiple* possible factors that cause positive data bias. Length bias is one such factor which requires the length of the trajectory to correlate with higher returns (see line 220-221). Note that length bias is not the same as the dataset having more long trajectories. Further, there can be other factors of positive data bias (Prop. 3 in Appendix C), e.g., potential-based reward shaping or near-optimal behavioral policies (see Appendix B.5 for experiments) which are not related to problem horizon or data trajectory length.
>
> **Positive bias is not a universal phenomenon but it can be common**
>
> We want to highlight that we do *not* intend to mean all existing offline RL datasets have positive data bias. In fact, we explicitly highlighted this in Section 4 (line 258) and showed examples like halfcheetah datasets, where there is no positive data bias. Nonetheless, we also point out that positive data bias often arises (due to factors listed in Prop. 3) in common data collection practices, such as intervention of unsafe or bad rollouts, or using expert demonstrations.
>
>
> **On nudging the agent to learn desirable behavior**
>
> Finally, to answer the reviewer’s question about how to purposely increase positive bias in data. Let us consider length bias as an example. During data collection, we can early stop a data collection episode if the agent is not performing well. This creates a positive correlation between data trajectory length and performance, and thereby increases the length bias in the dataset. Recall that survival instinct gives offline RL agents the incentive to produce trajectories that stay within support in a long term. Therefore, offline RL algorithms, due to survival instinct, can learn better with such a dataset. We are happy to answer any follow-up questions.

---

> > ### Comment · Reviewer_Sf8f · 2023-08-15
> >
> > I appreciate the clarifications. This is a good paper; I keep my score at 6.

---

### Official Review · Reviewer_NS2u · 2023-07-04

**Soundness:** 3 good
**Presentation:** 4 excellent
**Contribution:** 4 excellent
**Rating:** 8
**Confidence:** 3

**Summary:**

In this paper, the authors show that offline RL algorithms have an implicit survival instinct that often allows them to learn good policies with incorrect rewards. The authors argue that this is due to the data being positively biased and the pessimism that constrains offline RL algorithms to the data. This argument is supported by theory saying that when these conditions are met, the learned policy is close to the optimal policy and meets certain safety guarantees. They show several ways in which data bias can be introduced that benefit offline RL algorithms, such as length bias. Lastly, this phenomenon is shown empirically across several offline RL algorithms and benchmarks.

**Strengths:**

* Overall, I really enjoyed this paper. While pessimism is known to play an important role in offline RL, I think that this paper does a good job of illuminating how important pessimism is in these offline benchmarks. I think this will be an important paper for the offline RL community, especially when considering these benchmark tasks.

* The paper is very well written.

* The paper is very comprehensive with a strong Appendix section detailing experiment details and additional results.

* The paper has a lot of theory explaining the survival instinct phenomenon and showing how algorithms in the literature have this survival instinct.

**Weaknesses:**

* I still do not fully understand the argument around length bias (see Questions), and I am not convinced that this is a main cause of the phenomenon shown in Figure 3. In particular, how do we know specifically the length of the trajectories is the important property? I would assume that the medium and medium-expert trajectories datasets just have more support over actions that prevent the agent from falling down. The length of the trajectory just seems like a byproduct of these good actions. More discussion or evidence here would be appreciated.

* The main paper makes it unclear how data was generated for the grid world. While the appendix does have some more information about that, a little more should be explained here to make the example more understandable.
     * The offline agent is used on the “wrong reward” but what is the wrong reward? It seems like in the appendix you consider three wrong rewards, but which one is used? It seems like different rewards here could change the outcome drastically.

* (Line 219) The authors claim that a dataset generated by the optimal policy is infinitely positively biased regardless of the reward. This seems like a bold claim to make for all MDPs and all reward functions without a proof.

* What happens with expert D4RL datasets? It would be interesting to see what performance is on this, especially since the claim is that datasets from the optimal policy are infinitely positively biased regardless of the reward.

* Potential typo on Line 1415: The agent starts deterministically in the upper left it seems, not right.

**Questions:**

* Line 217 you say that any policy is infinitely positively biased for any rewards resulting from potential-based reward shaping. Isn’t positive bias a property of the offline dataset and not the reward? This didn’t make sense to me.

* Do you have any hypotheses for why some algorithms have different levels of survival instincts? Is it because of the amount of pessimism in the algorithms?

* I still have several questions about length bias:
    * I understand that often in benchmarks longer trajectories correspond to higher returns, but it is not clear to me why this always helps. For instance, suppose we have a dataset with trajectories where half the time the robot fell and half the time it did not. If we label all rewards with -1, wouldn’t the agent be incentivized to fall as fast as possible to incur less negative rewards.
    * Perhaps I am misunderstanding the assumptions in Proposition 5, but it seems like there is no assumption on how likely long trajectories are to appear in the dataset, only on the returns of long trajectories. I would think there has to be some assumption on the former.

**Limitations:**

Although I do not think this work necessarily needs to address these limitations, it would be beneficial to have some discussion about them in my opinion:

* While the paper does cover many relevant offline RL benchmarks, it does not explore all types of datasets. In particular, it does not appear that the authors explored cases where there are only sub trajectories of good behavior that need to be stitched together.

* The assumption of positively biased data in Theorem 1 seems strong. Since the definition of positively biased data is about the value of the delta-optimal policy set of the CMDP, there seems to be no intuition for when Theorem 1 will hold when looking at properties of the offline dataset only. In other words, it does not appear that there are any methods that practitioners could use to estimate the role of survival instincts for any arbitrary dataset.

* While there are theoretical results for offline model-based RL works, the paper’s experiments focus on model-free offline RL algorithms only.

---

> ### Author Rebuttal · Authors · 2023-08-10
>
> **Cause of Phenomenon in Fig 3**
>
> We would like to first clarify that length bias means the length of an in-support trajectory *positively correlates* with its return (line 220), not that the dataset has many long trajectories. In fact, a dataset would have no length bias if it has lots of long bad trajectories, e.g., halfcheetah datasets.
>
> We argue that length bias is one of the main causes of Fig. 3’s phenomenon. First, we visualize the presence of length bias in most datasets by showing the positive correlation in Fig. 4. Second, when trained on datasets without length bias, e.g., medium-replay and halfcheetah datasets, we observe that offline RL algorithms are more susceptible to wrong rewards.
>
> In fact, the medium-replay datasets and halfcheetah datasets are more diverse and cover more full-length trajectories than some other datasets with length bias. For example, hopper-medium-replay has 44 full-length trajectories and yet hopper-medium has only 1. This suggests that *"have more support over actions ... from falling down"* is not the main reason for the robustness to wrong rewards.
>
> **Grid World Example**
>
> PEVI produces the same policy and has the same performance under all three wrong rewards (zero, random, and negative) in the grid world example. We will clarify this and add more details on the data generation process in the revision.
>
> **Positive Bias from Optimal Behavioral Policy**
>
> In Appendix C, we provide a proof (Prop. 3 and Appendix C.4.2) of positive bias due to optimal, as well as deterministic near-optimal, behavioral policies. Intuitively, if data is from an optimal policy, all the actions in the support are optimal and therefore by survival instinct the learned policy will be optimal. The point below also empirically validates our statement.
>
> **D4RL Expert Datasets**
>
> In Appendix B.6, we report the results on the expert datasets of hopper, walker, and halfcheetah from D4RL. In Fig. 9, we show that in most scenarios (13 out of 15), offline RL algorithms can consistently achieve expert-level performance using all three wrong rewards, which verifies our theoretical statement.
>
> **Positive Bias from Potential-based Reward Shaping**
>
> We first want to note that there is a typo in line 217: any *policy* $\to$ any *data distribution*.
>
> To answer the reviewer’s question, positive data bias is a property of a data distribution $\mu$ *with respect to a reward class* $\tilde{\mathcal{R}}$, per Definition 2. In line 217, we intended to convey that, if we consider a reward class $\tilde{\mathcal{R}}$ consists of potential-based reward shaping rewards, then any data distribution is $\infty$-positively biased with respect to $\tilde{\mathcal{R}}$. This follows from the fact that the ordering of policies under a potential-based shaped reward is the same as the ordering under the original reward. We will revise the wordings in line 271 to clarify this.
>
> **Levels of Survival Instinct**
>
> There are a few causes. First, different offline RL algorithms are designed based on different notions of pessimism, e.g. behavior regularization, pessimistic values, etc. These different notions may or may not be translatable to the R-admissibility condition (Definition 4 in Appendix C) that controls the survival instinct. Second, offline RL algorithms often have hyperparameters that adjust the level of pessimism, which, as the reviewer suggested, can affect the level of survival instinct. In Appendix E, we theoretically show how to set hyperparameters of several algorithms to be sufficiently pessimistic. In the experiments, we conducted a limited hyperparameter search, which might not always have yielded the same level of survival instinct as one another. Finally, as noted in the limitations, practical implementations may have slightly different properties than the theoretical algorithms due to numerical issues and implementation details.
>
> **Dataset with -1 Reward**
>
> The behavior of an offline RL agent is subject to how "falling" is interpreted. Consider the robot example in the review. Suppose that each trajectory ends immediately when the robot falls. If falling is interpreted as "entering an absorbing state of value 0", the offline RL agent *would* fall, as the reviewer suggested.
>
> But since we stop the trajectory upon falling, we find it more appropriate to consider falling as “entering a state that is unknown” (Footnote 4 & Appendix B.5). In this case, as the agent never sees what happens after falling, an offline RL algorithm with sufficient pessimism would imagine these unknown states have lower values than getting -1 rewards and hence avoid falling.
>
> We note that using -1 reward is actually similar to our experiments with the *negative* reward for hopper and walker. Fig. 7 shows offline RL algorithms are able to learn policies that avoid falling with the negative reward. We also observed that offline RL algorithms can learn good policies with -1 reward in our earlier preliminary experiments.
>
>
> **Assumption on Prop. 5**
>
> The reason why we do not explicitly make an assumption on the likeliness of long trajectories in Prop. 5 is that it is covered already by the third point in Assumption 3. Notice that time information is part of the state in a finite horizon problem, so Assumption 3 implies that there is a non-zero probability of having full-horizon data trajectories (otherwise, the concentrability coefficient would be $\infty$). When Assumption 3 holds, statistical errors due to finite dataset size are included in the $\iota$ term in Theorem 1 (line 132). We will clarify this in the revision.
>
> **Limitations**
>
> We agree with the reviewer on these limitations and will discuss them in the final version of the manuscript. In particular, we find it to be interesting future work to quantify the amount of positive bias of a data distribution, including the size of reward class $\tilde{\mathcal{R}}$ it has positive bias with respect to, without having to actually run offline RL algorithms.

---

> > ### Comment · Reviewer_NS2u · 2023-08-12
> >
> > Thank you for the thorough rebuttal! I appreciate the clarifications on my questions.
> >
> > I agree that there is an interesting correlation shown in Figure 4, however I am still not convinced length itself is a driving force of the survival instinct. I realize now that I did not express my thoughts correctly when I said "more support over good actions." Instead I meant to say concentration around good actions (and thus a lack of support over bad actions).
> >
> > Concretely, in the example that the author's give, there is 1 trajectory in Hopper medium and 44 trajectories in medium replay. It is therefore clear that the medium policy _only_ makes good decisions when it is put in a state where it is about to fall. On the other hand, medium-replay has bad actions when it is about to fall, and as such, the offline RL policy will not be as penalized for choosing these actions.
> >
> > Although I am still unconvinced about length bias, I still think the results are important for the offline RL community, and I will keep my score where it is.

---

> > > ### Author Response · Authors · 2023-08-12
> > > **Thank you for your quick response.**
> > >
> > > Thanks for getting back to us so quickly!
> > >
> > > From your response, we found our explanation regarding Fig. 3 in the rebuttal might have created a confusion about how to interpret Figure 4 and the length statistics. Please allow us to further clarify that.
> > >
> > > When we wrote
> > >
> > > > *hopper-medium-replay has 44 full-length trajectories and yet hopper-medium has only 1*
> > >
> > > we intended to mean that, *among all the trajectories*, only 44 and 1 of them have the full length of 1000 steps. We did not mean that "there is 1 trajectory in Hopper medium and 44 trajectories in medium replay". In D4RL, there are in total 2187 trajectories in hopper-medium dataset and 2041 trajectories in hopper-medium-replay dataset. We plotted each trajectory as a dot in Fig 4. We can see that most of them are incomplete trajectories; there are just a few dots (1 dot and 44, respectively, if we count) on the rightmost of Fig 4 having full length.
> > >
> > > Therefore, the medium policy does not always make good decisions when it's about to fall, because it actually falls in the majority of data trajectories, except one (the average trajectory length of hopper-medium is 457.25 steps, less than half the full length).
> > >
> > > The main difference between the two datasets is that: the behavior policy in hopper-medium always takes actions leading to high instantaneous reward until it falls. On the other hand, in hopper-medium-replay, some policies take actions that would continue to survive but have low instantaneous reward (see the dots on the lower right of the hopper-medium-replay subfigure of Fig 4).
> > >
> > > Such a difference in policy behaviors is what creates the length bias, i.e., the positive correlation between trajectory return and length, in hopper-medium. On the other hand, hopper-medium-replay does not have this property (due to the existence of surviving low-return long trajectories).
> > >
> > > Finally, the survival instinct in offline RL makes the agent favor long trajectories that can stay in the data before optimizing for the reward. Since all long trajectories have high return when the data has length bias (like hopper-medium), we see the robustness to wrong rewards in Fig 3.
> > >
> > > We are happy to answer any further questions.

---

> > > > ### Comment · Reviewer_NS2u · 2023-08-19
> > > >
> > > > Thank you for clearing up my misconceptions, and I am sorry for the delayed response. It is clear that there is something more interesting going on here than I originally thought. Since this was my biggest concern about the paper, I will raise my score from a 7 to an 8 since I believe this is an important paper for the offline RL community.

---

### Author Rebuttal · Authors · 2023-08-10

We thank all reviewers for providing feedback to our manuscript. It is encouraging to see that reviewers generally find our empirical findings and theoretical analysis relevant to the offline RL community. We are excited to see that the reviewers propose a few interesting future research directions, and we are happy to provide a discussion on these future directions in the revision. We also would like to thank reviewers for pointing out typos in our manuscript, which will be fixed in the final version. We will address specific questions and concerns from each reviewer in the individual responses.


In the attached PDF, we include a table which has additional experimental results on offline safety gymnasium [1] as per suggestions by reviewer k1TU. We observe that *offline RL with a naïve data filtering strategy can achieve comparable performance as the per-task best performing state-of-the-art offline safe RL algorithm*. We hope our new results can give reviewers more confidence on the safety property of offline RL.

[1] Liu et al. Datasets and Benchmarks for Offline Safe Reinforcement Learning. 2023.

---

### Author Response · Authors · 2023-08-21
**Thank you for reviewing and discussing with us**

Dear reviewers,

We would like to thank all reviewers again for their time and positive feedback on our work. We will incorporate the reviewers' suggestions in the final revision. It is glad to see that we all think that the findings presented in the paper bring out many new possibilities in RL. We are looking forward to them.

Best,
Authors

---

### Decision · Program_Chairs · 2023-09-21

**Decision:**

Accept (spotlight)

**Comment:**

This paper explores a phenomenon in offline reinforcement learning (RL) where agents learn effective policies even with incorrect reward signals. The authors argue that this resilience arises due to the inherent pessimism in offline RL algorithms and biases in data collection, introducing a 'survival instinct'. Empirical demonstrations and theory, demonstrate this observation across various environments. The paper has clear exposition, comprehensive experiments, and sound theoretical analysis. Some reviewers raised concerns about the clarity of some arguments, especially regarding data biases and the claim of intrinsic safety. Some also highlighted the need for specific clarifications around situations where this phenomenon manifests and several typographical errors scattered throughout the paper. The authors addressed most of the reviewers' concerns, and the reviewers found the work to be potentially impactful for future work.